# MMTS-Bench: A Comprehensive Benchmark for Multimodal Time Series Understanding and Reasoning

## Abstract

Time series data are central to domains such as finance, healthcare, and cloud computing, yet existing benchmarks for evaluating various large language models (LLMs) on temporal tasks remain scattered and unsystematic. To bridge this gap, we introduce MMTS-Bench, a comprehensive multimodal benchmark built upon a hierarchical taxonomy of time-series tasks, spanning feature analysis, temporal reasoning, and cross-modal alignment. MMTS-Bench comprises 2,424 time series question answering (TSQA) pairs across 4 subsets: **Base**, **InWild**, **Match**, and **Align**, generated through a progressive real-world QA framework and modular synthetic data construction. We conduct extensive evaluations on closed-source, open-source LLMs and existing time series adapted large language models (TS-LLMs), revealing that: (1) TS-LLMs significantly lag behind general-purpose LLMs in cross-domain generalization, (2) LLMs show weaknesses in local tasks compared to global tasks, and (3) chain-of-thought (CoT) reasoning and multimodal integration substantially improve performance. MMTS-Bench not only provides a rigorous evaluation framework but also offers clear directions for advancing LLMs toward robust, interpretable, and generalizable time-series reasoning.[1].

## 1 Introduction

Time-series data underpin critical systems in finance, healthcare, transportation, and cloud computing (Zeng et al., 2023; Zhou et al., 2021; Liu et al., 2024), capturing how processes evolve over time. Traditionally, tasks such as forecasting, classification, anomaly detection, and imputation (Nie et al., 2023; Zhang et al., 2020; 2024) rely on specialized statistical models and tooling, demanding substantial domain expertise. In recent years, with the rapid advancement of natural language processing (NLP), especially the breakthroughs in Large Language Models (LLMs) (OpenAI, 2023; Comanici et al., 2025; Anthropic, 2025a; Yang et al., 2024; Team et al., 2025b), new possibilities have emerged to overcome the professional barriers in time series analysis (Xie et al., 2024; Wang et al., 2025b;a; Jin et al., 2024b). Integrating time series data with LLMs to build end-to-end time series models has become a prominent research direction (Liu et al., 2023; Bai et al., 2025). Recently, a growing number of researchers have begun to explore the application of LLMs to time series analysis, giving rise to novel tasks such as time series description(Zhang et al., 2023), text-context-assisted forecasting (Jin et al., 2024a), simple time series question answering (QA) (Wang et al., 2025a), complex time series reasoning, and cross-variable QA (Xie et al., 2024).

The above works that combine LLMs with time-series data also require substantial training and testing datasets for support. Prior efforts either enrich classic time-series datasets with textual annotations (Liu et al., 2024; Yu et al., 2024) or build QA datasets for time series reasoning (Wang et al., 2025b; Kong et al., 2025). However, most current studies rely on a "flat" task taxonomy (Wang et al., 2025a; Cai et al., 2024) to define capabilities and synthesize QA data. Such taxonomies either have no hierarchical structure or are simple, making it difficult to comprehensively evaluate LLMs' abilities in time-series understanding and reasoning at a fine-grained level. Moreover, the construction of most time-series datasets and the fine-tuning of LLMs are limited to single or small-domain

---

[1]Code and data are available at `https://anonymous.4open.science/r/MMTS-BENCH-BEF7/`

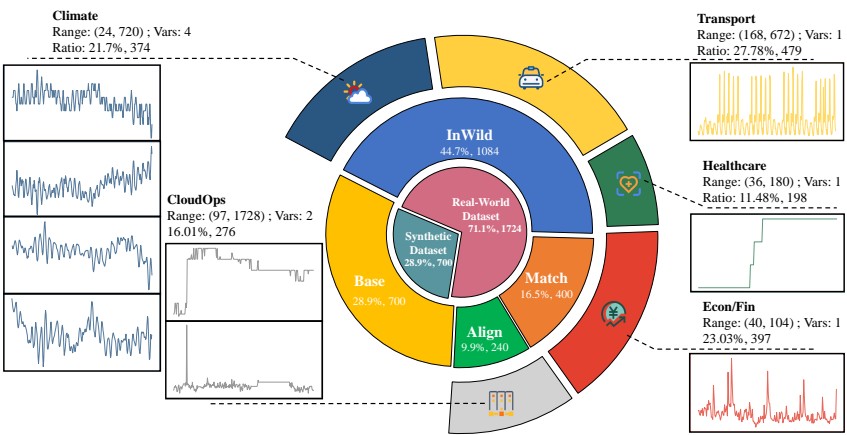

Figure 1: MMTS-Bench at a glance

data (Wang et al., 2025b; Dong et al., 2024), and there is still a lack of a comprehensive benchmark across multiple domains to evaluate LLMs' out-of-distribution (OOD) generalization.

In this work, we propose a hierarchical taxonomy of time series tasks that reorganizes analytical tasks into a multi-level, orthogonal structure spanning from basic perception to advanced reasoning, thereby clarifying and extending several key task areas that have previously been overlooked. Building on this taxonomy, we construct MMTS-Bench, a new multi-modal, multi-dimensional evaluation benchmark for time series tasks (see Figure 1), comprising 2,424 TSQA pairs across four subsets. One subset is built from synthetic time-series data: **(1) Base**, which assesses capabilities in structural awareness and feature analysis. The others are built from real-world time series spanning five domains (e.g., Transport; see Appendix A.2) in the LOTSA dataset (Woo et al., 2024b): **(2) In-Wild**, which targets feature analysis and temporal reasoning; **(3) Match**, which evaluates sequence-similarity matching and morphological correspondence; and **(4) Align**, which measures bidirectional conversion between time series and natural language as well as advanced cross-modal semantic understanding. We also conduct a comprehensive evaluation of multiple mainstream LLMs using MMTS-Bench, providing detailed capability rankings and deeper insights into current strengths and limitations, thereby offering concrete guidance for the development of future time-series foundation models and the construction of datasets. Our main contributions are as follows:

1. **MMTS-Bench.** We introduce a capability-oriented, hierarchical taxonomy of time-series tasks and instantiate it in MMTS-BENCH, a multimodal, multi-dimensional benchmark comprising 2,424 QA pairs across four subsets (Base, InWild, Match, Align) covering skills from feature analysis to temporal reasoning and cross-modal alignment; using this benchmark, we perform large-scale, fine-grained model assessments and derive practical recommendations for improvement.

2. **Progressive real-world TSQA generation.** We propose an innovative three-stage framework for real-world time series data QA generation, effectively addressing generation quality and reliability, and offering a new methodology for large-scale QA generation based on real-world time series data.

3. **Controllable synthetic data pipeline.** We develop a controllable synthetic data generation pipeline, where modular construction and templated generation are employed to systematically control data diversity and difficulty, leading to high-quality datasets for targeted evaluation of foundational abilities.

## 2 RELATED WORK

### 2.1 TIME-SERIES LLMS

Multimodal large language models (MLLMs) have demonstrated strong capabilities in natural language processing and cross-modal reasoning. In the domain of TS-LLMs, several implementation

paradigms have recently been explored. Time-MQA Kong et al. (2025) serialize time series as textual inputs and report early gains on the time-series question answering (TSQA) task. ChatTime (Wang et al., 2025a) quantize continuous values into a finite token space, enabling continuous pre-training within a unified LLM framework. Zhuang et al. (2024) use GPT-4o (OpenAI, 2024) in a two-stage, coarse-to-fine anomaly-detection pipeline over rendered time-series plots, while Insight-Miner (Zhang et al., 2023) and FinVis-GPT (Wang et al., 2023) adapt LLaVA (Liu et al., 2023) for time-series description and candlestick-chart analysis, respectively.

However, representing dense time series data as text or plots inflates sequence length and token budgets, with typically modest gains. Alignment-based methods mitigate these issues by retaining a dedicated time-series encoder and learning a lightweight projector into the LLM token-embedding space, enabling efficient TS–text interaction. Following this paradigm, Chow et al. (2024) and ChatTS (Xie et al., 2024) build TS-LLMs and report competitive results across classification, description, QA, and reasoning. Nevertheless, a unified and comprehensive benchmark for systematically evaluating the multi-dimensional capabilities of TS-LLMs is still lacking.

## 2.2 Time-Series QA Datasets

Although recent work has combined LLMs with time series and released several datasets, most of them remain confined to forecasting (Hu et al., 2025; Liu et al., 2025; 2024; Wang et al., 2024), while publicly available TSQA datasets are scarce. Moreover, existing TSQA datasets (Wang et al., 2025a; Kong et al., 2025; Wang et al., 2025b) suffer from domain inconsistencies, "flat" ability taxonomies, and rigid question formats, isolating different works and hindering meaningful cross-comparisons.

On the univariate side, ChatTime-TSQA (Wang et al., 2025a) is generated from fixed, simple templates and focuses on four basic properties—trend, volatility, seasonality, and outliers. Time-MQA (Kong et al., 2025) and Chat-TS (Quinlan et al., 2025) derive QA pairs from real-world domains via single-turn prompting; despite manual filtering, these datasets offer limited coverage for comprehensive, balanced evaluation. On the multivariate side, EngineMT-QA (Wang et al., 2025b) constructs QA pairs from aviation-engine data through a four-stage pipeline, but its narrow domain and template reliance constrain its generality as a benchmark. ChatTS (Xie et al., 2024) introduces TSEvol-Instruct, which generates QA pairs via iterative prompting over diverse time-series data (synthetic and real; univariate and multivariate); however, its "flat" taxonomy and rigid question design weaken its ability to evaluate distinct capability dimensions.

In contrast, our proposed MMTS-Bench is a multimodal, multi-dimensional benchmark for TSQA that covers varied difficulty, domains, and both synthetic and real data across univariate and multivariate cases. Through iterative expert curation and human validation, the benchmark offers a balanced and reliable basis for assessing the performance of models.

# 3 MMTS-Bench

## 3.1 Multi-dimensional Task Classification Framework

To systematically evaluate the comprehensive understanding and reasoning capabilities of LLMs in time series analysis, we propose a multi-dimensional task classification framework together with a corresponding dataset construction methodology. Existing time-series QA datasets for assessing LLMs often suffer from a lack of consistency and hierarchical structure. To address the limitations of such "flat" classification schemes, we decompose temporal understanding into five functionally orthogonal core dimensions (see Appendix A.3 for taxonomy details): structural awareness, feature analysis, temporal reasoning, sequence matching, and cross-modal understanding, which theoretically yield 286 fine-grained composite task types [2].

Based on this framework, we design four subsets under MMTS-Bench. **Base** provides a controlled synthetic environment focusing on fundamental abilities such as structural awareness and feature analysis, without involving complex reasoning. It includes multiple-choice, binary-choice, and numerical questions. For evaluation, we divide the subset into two splits: the Choice split and the

---

[2]By combining feature analysis and temporal reasoning, 35 composite sub-tasks are formed. With the addition of structural awareness, this extends to 280. Including 4 from sequence matching and 2 from cross-modal understanding, the total reaches 286.

Numerical split. **InWild** leverages real-world time series to further examine LLMs' capacity for feature analysis and temporal reasoning under complex and noisy conditions. It consists of multiple-choice and binary-choice questions. **Match** and **Align** focus on two less-studied dimensions in time series analysis, namely similarity matching and cross-modal understanding, and are composed of multiple-choice questions.

Table 1 summarizes the subtasks across the five orthogonal dimensions and their links to the subsets, with full definitions given in Table 5 (Appendix A.3). This multi-dimensional construction enables fine-grained profiling of LLM capabilities in time series analysis and helps identify the bottlenecks they face in complex analytical reasoning tasks.

Table 1: Overview of task dimensions, subtasks, and related subsets in MMTS-Bench.

| Dimensions | Subtask | Related Subsets |
|---|---|---|
| Structural Awareness | Non-Stationarity, Local-Global, Univariate-Multivariate | Base |
| Feature Analysis | Trend, Seasonality, Noise, Volatility, Basic Analysis | Base & InWild |
| Temporal Reasoning | Deductive, Inductive, Causal, Analogical, Counterfactual Reasoning | InWild |
| Sequence Matching | Isomorphic, Robust, Positioning, Reverse Matching | Match |
| Cross-Modal | Time-series to Semantic, Semantic to Time-series | Align |

## 3.2 SYNTHETIC DATASET

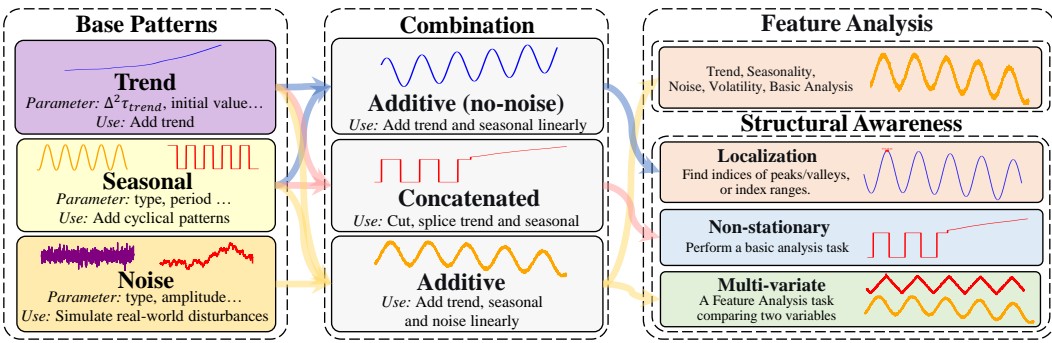

Figure 2: **Base** Construction Pipeline. Synthetic time series with controllable characteristics are generated by concatenating and adding basic components of trend, seasonality, and noise. The plotting style of this figure is adapted from Cai et al. (2024)

**Base** is designed to conduct fundamental evaluation experiments under controllable conditions using synthetic data. To achieve this, it employs 17 expert-designed templates, which use controllable parameters to automatically construct QA pairs. These parameters, including trend direction and strength, seasonal patterns, and noise types, generate synthetic time series through a modular framework (detailed mathematical formulation in Appendix A.4.1), consistent with established practices in time series construction (Cai et al., 2024; Das et al., 2024; Fu et al., 2024; Zhang et al., 2024). The framework consists of three parameterized units: trend, seasonality, and noise (as shown in Figure 2), which are superposed and concatenated to produce sequences that are interpretable and controllable.

## 3.3 REAL-WORLD DATASETS

**InWild** focuses on realistic and complex scenarios, enabling richer assessments of real-world generalization and the analytical and reasoning abilities of LLMs on time series, in contrast to **Base**, which emphasizes controllability and fundamental property evaluation. We construct **InWild** interactively with LLMs through a three-stage pipeline (as illustrated in Figure 3; see Appendix A.4.2 for details): (1) **Description Generation**, given multimodal inputs (raw sequences, visualizations, domain metadata, and pre-computed statistical features), where the domain metadata specifies the

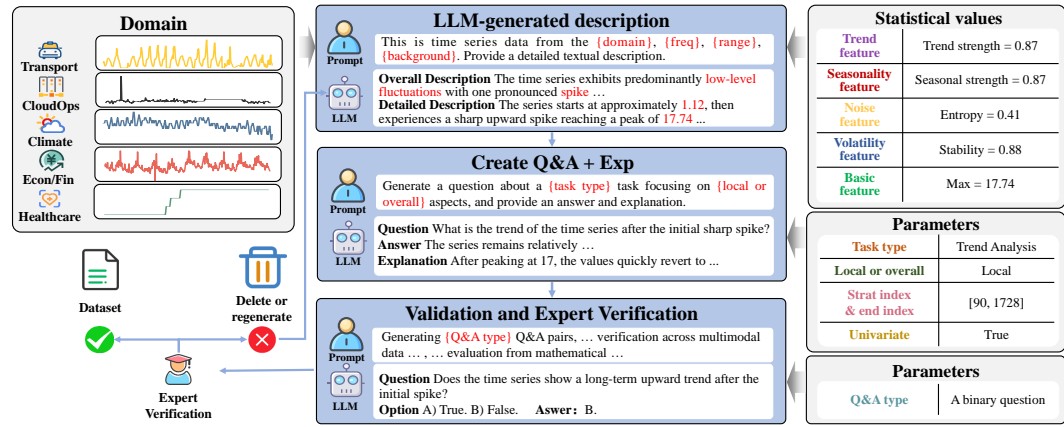

Figure 3: **InWild** Construction Pipeline. Flowchart illustrating the conversion of domain-specific time series via statistical analysis and multimodal input preparation for LLMs. It highlights the feedback loop between LLM generation and expert verification, showing how raw sequences from multiple domains (Economics/Finance, Transport, CloudOps, Climate, Healthcare) are enriched with features (trend strength, seasonal strength, entropy, stability) and structured as inputs for automated QA generation.

physical meaning of variables (e.g., hourly traffic volume, CPU and memory utilization, etc.), the LLM produces overall and detailed descriptions; (2) **QA Construction**, given initialization parameters (task type, global vs. local scope, uni- vs. multivariate setting, and the index range of the sequence), the descriptions are converted into Q-A-E (question-answer-explanation) triplets; and (3) **Validation and Expert Refinement**, the LLM performs logical and mathematical consistency checks to produce standardized QA pairs, followed by human expert review.

**Match** is designed to evaluate LLMs' ability to perform similarity matching on time series. It is constructed by extracting fragments from real-world time series and applying Dynamic Time Warping (DTW) to search the original series for four candidate sequences at different similarity levels. Each fragment and its candidate sequences are then combined into a multiple-choice question, where a fixed template asks the model to identify the candidate most similar to the fragment (Figure 4). In addition, we apply operations such as smoothing, extension, and reversal to the fragment samples, constructing four task paradigms of different difficulty levels (construction details in Appendix A.4.2): (1) Isomorphic, matching sequences with identical lengths, (2) Robust, matching sequences after smoothing, (3) Positioning, localizing target patterns within longer sequences, and (4) Reverse, matching sequences after reversal.

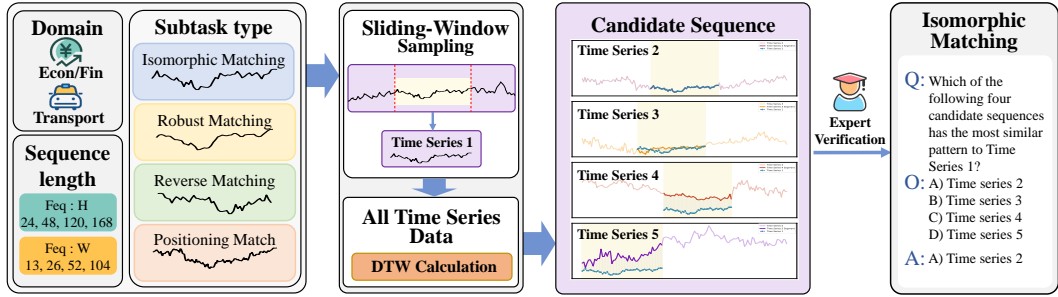

Figure 4: **Match** Construction Pipline. Fragments are extracted from real-world time series, and candidate series with different similarity levels are retrieved using DTW. QA pairs are then formed with fixed templates, while transformations such as smoothing, extension, and reversal create four task paradigms of varying difficulty.

**Align** is designed to evaluate LLMs' ability in cross-modal understanding between time series and natural language. It is constructed by extracting the descriptions generated in the first step of **InWild** and pairing them with the corresponding time series to form multiple-choice questions (construction details in Appendix A.4.2). Based on this process, two symmetric tasks are generated: (1) time-series to semantics, where a time series is provided and the LLM must select the matching description from several candidates, and (2) semantics to time-series, where a description is provided and the LLM must identify the corresponding time series among the options. To increase task difficulty, time series are sampled from three real-world domains with similar value ranges, ensuring that the incorrect options remain plausible.

### 3.4 HUMAN-IN-THE-LOOP CURATION

To ensure dataset quality, we adopt a human-in-the-loop process combining automated generation with expert review (Wu et al., 2022), executed by a fixed group of 10 time-series domain experts. In **InWild**, experts check question soundness and reasoning validity; in **Match**, experts confirm that the sequence most similar in shape indeed attains the minimum DTW distance; and in **Align**, experts verify textual descriptions accurately reflect temporal patterns.

We rigorously evaluated the reliability of this annotation process using Fleiss' $\kappa$ (Fleiss, 1971). An overall $\kappa$ score of 0.73 was achieved, falling into the "Substantial Agreement" category (0.61–0.80) (Landis & Koch, 1977). To ensure the validity of the final benchmark, ground truth labels were determined by majority vote, with any ambiguous cases undergoing a second round of adjudication. This high level of agreement provides strong statistical evidence for the dataset's reliability.

Furthermore, we evaluated one-step generation and our three-stage pipeline without human-in-the-loop in **InWild**. Results show that without time series inputs, LLMs achieve about 57% accuracy with one-step generation[3]; accuracy decreases to 44% with the three-stage pipeline, and further to 35% when expert review is applied. This demonstrates the necessity of the human-in-the-loop process in preventing question errors or answer leakage that may otherwise lead to inflated accuracy.

### 3.5 STATISTICAL RELIABILITY AND VALIDITY ANALYSIS

To complement human curation, we conducted rigorous statistical analyses (details in Appendix E) to verify the benchmark's quality.

**Evaluation Stability.** First, we assessed the evaluation stability through bootstrap confidence intervals estimation and iterative subsampling analysis. The results indicate that MMTS-Bench yields highly stable evaluation scores with narrow confidence intervals and a coefficient of variation down to $10^{-3}$ magnitude, providing a robust safety margin against sampling variance.

**Validity against Shortcut Learning.** Furthermore, to ensure performance reflects intrinsic reasoning rather than dataset artifacts, we analyzed the dependency of accuracy on explicit surface attributes (e.g., sequence length, dimensionality). Our analysis reveals negligible correlations (e.g., $|r| < 0.08$ for sequence length) and minimal performance gaps. These findings confirm that MMTS-Bench is robust against spurious correlations, serving as a reliable benchmark for assessing intrinsic time-series understanding capabilities.

## 4 EVALUATION RESULTS

Using the MMTS-Bench dataset, we conducted a systematic benchmarking and analysis of the latest open-source and closed-source LLMs alongside state-of-the-art (SOTA) TS-LLMs. We report *Accuracy*, *Accuracy@N%*, and *Relative Accuracy* (definitions in Appendix A.1) across multiple subsets spanning different task dimensions. To ensure statistical robustness and experimental reliability, all experiments were conducted five times independently at temperature 1.0, with results averaged across trials.

**Model Selection for Evaluation.** We conduct a comprehensive evaluation on MMTS-Bench using three representative categories of large language models to assess their performance on time

---

[3]In **InWild**, based on the distribution of multiple-choice and binary questions under no-input conditions, the expected accuracy of random guessing is 37.5%.

series QA tasks. Our selection includes: (1) Closed-source models: Claude 3.7 Sonnet (Anthropic, 2025a), Claude Sonnet 4 (Anthropic, 2025b), Gemini 2.5 Flash/Pro (Comanici et al., 2025), GPT 5 Minimal/High (OpenAI, 2025b), GPT 4.1/4.1 mini (OpenAI, 2025a), and GPT 4o (OpenAI, 2024); (2) Open-source models: DeepSeek V3 (DeepSeek-AI, 2024), Kimi K2 (Team et al., 2025a), and Qwen series (Yang et al., 2024; 2025) (including 2.5 and 3 variants with different parameter scales. Notably, for Qwen3 series models, we evaluate both the thinking and non-thinking modes; all Qwen2.5 models we evaluate are instruction-tuned ("Instruct"), we omit "Instruct" in later mentions for brevity); (3) TS-LLMs: ChatTS (Xie et al., 2024), ITFormer (Wang et al., 2025b), and ChatTime (Wang et al., 2025a), which are specifically designed for time series data analysis. For MLLMs (Qwen2.5 VL(Bai et al., 2025), Claude Sonnet series(Anthropic, 2025a;b), Gemini 2.5 Pro(Comanici et al., 2025), GPT 4 series(OpenAI, 2023)), we evaluate across text-only, vision-only, and vision-text combined inputs. To ensure a fair evaluation, we design a standardized time series input format (see Appendix B for details); for TS-LLMs, we carefully reuse the original inference scripts. This multi-dimensional evaluation framework aims to clarify the impact of different model architectures and input modalities on time series understanding and reasoning tasks.

Table 2: Performance comparison of typical LLMs across different categories on MMTS-Bench. 'TS' = Time Series modality. "–" = no testing. Best results are **underlined and bolded**.

| Category | Model | Modality | Average | MMTS-Bench Subsets | | | |
|---|---|---|---|---|---|---|---|
| | | | | Base | InWild | Match | Align |
| Closed-source | GPT-5-High | Text | **0.74** | **0.51** | **0.72** | **0.82** | **0.99** |
| | GPT-4o | Text | 0.61 | 0.42 | 0.62 | 0.50 | 0.97 |
| | Claude-Sonnet-4 | Text | 0.71 | 0.49 | 0.71 | 0.71 | 0.98 |
| | Gemini 2.5 Pro | Text | 0.70 | 0.48 | 0.68 | 0.79 | 0.98 |
| Open-source | Kimi-k2 | Text | **0.63** | **0.45** | **0.63** | 0.60 | **0.95** |
| | DeepSeek-v3 | Text | 0.62 | 0.41 | 0.61 | **0.65** | **0.95** |
| | Qwen2.5-14B | Text | 0.55 | 0.35 | 0.53 | 0.57 | 0.88 |
| | Qwen2.5-7B | Text | 0.45 | 0.33 | 0.44 | 0.40 | 0.69 |
| Time-series | ChatTS | TS | **0.49** | **0.39** | **0.50** | **0.37** | **0.80** |
| | ITFormer | TS | 0.31 | 0.31 | 0.33 | 0.24 | 0.29 |

## 4.1 VERTICAL COMPARISON OF LARGE LANGUAGE MODELS

**TS-LLMs Show Limited Generalization Capabilities.** From Table 2 and the more comprehensive experimental results in the Appendix D, we observe that both closed-source and open-source general-purpose LLMs consistently outperform TS-LLMs across diverse tasks in MMTS-Bench, including **InWild**, **Match**, **Align**, and the Choice split in **Base**. TS-LLMs show marked weaknesses in OOD generalization: ChatTS is comparable to, or slightly below, its base model Qwen2.5-14B; ITFormer lags substantially behind Qwen2.5-7B when applied outside the aero-engine domain; and ChatTime fails to produce valid outputs. These findings indicate that while TS-LLMs may perform adequately within narrow domains, their generalization remains severely limited. Moreover, comparison with human experts further underscores this gap: on **InWild**, humans achieve 67% accuracy, clearly surpassing TS-LLMs.

**Existing Multimodal Alignment in TS-LLMs Remains Inefficient.** To investigate the key factors affecting TS-LLM performance, we modified the training pipeline of the current SOTA model ChatTS and conducted a series of ablation studies on encoder architecture, scale, positional encoding, backbone LLM size, and prompt prefix. The results show that model performance is predominantly determined by the backbone LLM size, while being largely insensitive to encoder structure, scale, and positional encoding, suggesting that the encoder's contribution remains limited and underdeveloped. Notably, augmenting the prompt with simple statistical summaries leads to substantial improvements in reasoning accuracy, underscoring the importance of task-aware prompt design. Details are provided in Appendix C.

## 4.2 CROSS-DIMENSIONAL COMPARISON OF LARGE LANGUAGE MODEL PERFORMANCE

**LLMs Underperform on Temporal Reasoning Relative to Feature Analysis.** As shown in Table 15 (Appendix D), current LLMs reach an average accuracy of 62% on feature analysis tasks in **InWild**, notably higher than the 55% achieved on temporal reasoning tasks. A comparison under unified input modalities indicates that Claude-Sonnet-4 and Gemini-2.5-Pro are the strongest closed-source LLMs[4], while DeepSeek-V3 and Kimi-K2 lead among open-source LLMs (Figure 5). Table 15 further shows that weaker performance on feature analysis generally coincides with poor temporal reasoning. For instance, GPT-4.1-Mini among closed-source models and Qwen2.5-7B among open-source models both follow this trend, suggesting that insufficient feature analysis capability constrains temporal understanding.

Within the Feature Analysis dimension, seasonality tasks consistently yield the lowest accuracy across **Base** and **InWild**. This indicates that LLMs may struggle with capturing seasonal patterns, making seasonality a particularly challenging subtask. In the Temporal Reasoning dimension, results further reveal that causal and counterfactual reasoning are especially difficult compared to other reasoning tasks. As illustrated in Figure 8 (Appendix A.4.2), LLMs frequently fall into local reasoning traps in causal tasks, failing to capture causal relations from a global perspective. Similarly, counterfactual reasoning involves reconstructing dependencies under hypothetical conditions. When LLMs lack sufficient local and global awareness, they tend to make faulty conditional assumptions, which ultimately leads to reasoning failures.

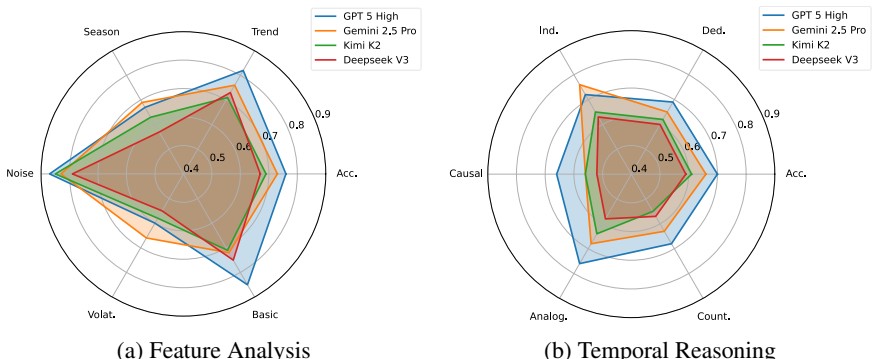

(a) Feature Analysis        (b) Temporal Reasoning

Figure 5: Accuracy of top-ranked LLMs on subtasks within the two dimensions of **InWild**. "Acc." denotes the average accuracy within each dimension. (a) Feature Analysis: Volat. = Volatility; (b) Temporal Reasoning: Ded. = Deductive, Ind. = Inductive, Analog. = Analogical, Count. = Counterfactual.

**LLMs Show Weaknesses in Local Tasks Compared to Global Tasks.** As shown in the **Base** results (Table 3), LLMs show their most pronounced weaknesses in Local subtasks under the Structural Awareness dimension. Across all model categories, accuracy on Local subtasks remains low, whereas performance on Global subtasks is considerably higher. This suggests that while LLMs can capture overall regularities in time-series structures, they generally lack precise localization and fine-grained discrimination across different input modalities—an issue that connects with the reasoning failures discussed above.

In contrast, performance gaps in non-stationarity and univariate–multivariate subtasks remain relatively modest. Non-stationary sequences are typically constructed by concatenating subsequences with distinct statistical properties, leading to global shifts that can be detected through overall structural and distributional cues. Similarly, because it is difficult in task design to generate questions that necessarily require leveraging cross-variable dependencies, multivariate tasks within the context length of LLMs often degenerate into multiple univariate ones.

Moreover, in the evaluations on the **Match** and **Align** subsets (see Table 16 and Table 18 in Appendix D), we observe two key findings: (1) among the four sequence matching subtasks, LLMs

---

[4]Claude-3.7-Sonnet ranked second but was excluded due to its substantial involvement in dataset construction.

Table 3: The *Accuracy@10%* metric (see Appendix A.1) of different models on the **Base** subset's numerical split. $^{cot}$ = *thinking* mode. 'TS' = Time Series modality. Abbreviations: Stat. = Stationary, Non-Stat. = Non-Stationary, Local = Localized, Uni. = Univariate, Multi. = Multivariate.

| Category | Model | Modality | Structural Awareness | | | | | |
|---|---|---|---|---|---|---|---|---|
| | | | Stat. | Non-Stat. | Local | Overall | Uni. | Multi. |
| Open-source | Kimi-K2 | Text | 0.56 | 0.56 | 0.19 | 0.45 | 0.56 | 0.54 |
| | Qwen3-32b$^{cot}$ | Text | 0.40 | 0.40 | 0.20 | 0.32 | 0.40 | 0.33 |
| | Qwen3-32b | Text | 0.47 | 0.50 | 0.07 | 0.38 | 0.47 | 0.42 |
| Closed-source | GPT-4o | Text | 0.53 | 0.58 | 0.13 | 0.43 | 0.53 | 0.51 |
| | GPT-4o | Vision | 0.34 | 0.29 | 0.31 | 0.31 | 0.34 | 0.34 |
| | GPT-4o | V+T | 0.59 | 0.57 | 0.31 | 0.56 | 0.59 | 0.56 |
| Time-series | ChatTS | TS | 0.41 | 0.42 | 0.12 | 0.40 | 0.41 | 0.40 |

perform significantly worse on Localization Matching and Reverse Matching than on the remaining ones. This may be due to the inherent limitations of attention mechanisms and autoregressive paradigms under temporal direction transformations, though further experiments are required to confirm this; (2) in the two subtasks under the Cross-Modal dimension, LLMs generally exhibit strong performance. This is because LLMs, by leveraging basic statistical cues (e.g., maxima, minima) and conducting logical reasoning, can align sequences with textual descriptions without the need to capture the complete temporal structure.

### 4.3 ANALYZING APPROACHES TO ENHANCE LLM PERFORMANCE ON TIME SERIES TASKS

**Multimodal Fusion Enhances LLMs' Ability in Time Series Analysis.** As shown in Table 15 (Appendix D), Gemini 2.5 Pro achieves 68% accuracy with text-only inputs, 72% with vision-only inputs, and reaches 76% when combining text and visual modalities. Similarly, GPT-4.1 exhibits similar improvements with multimodal fusion. Building on this observation, we further evaluated Qwen2.5-VL-7B across all MMTS-Bench datasets, which also showed consistent gains (see Figure 6a). However, such improvements are not universal; for example, GPT-4.1-Mini demonstrates shows reduced performance on **InWild**. We suspect this limitation arises from differences in the LLMs' inherent ability to integrate multiple modalities. Therefore, introducing additional modalities is an effective way to narrow the information gap of LLMs in time series analysis, but its effectiveness mainly depends on the fusion design and training methods of the LLMs.

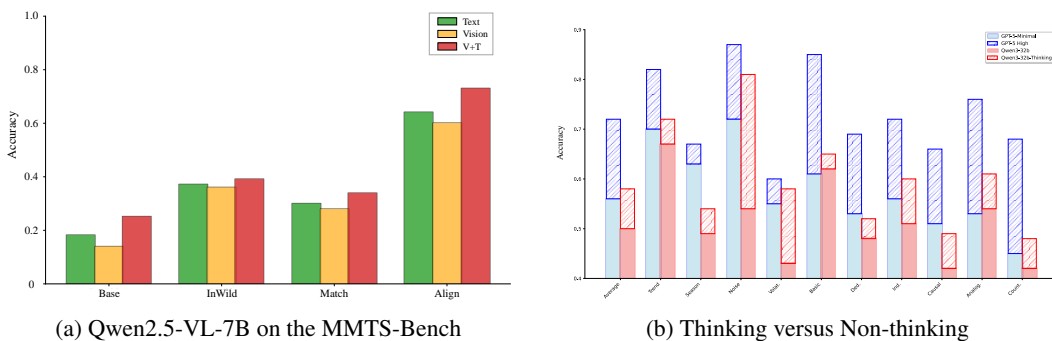

(a) Qwen2.5-VL-7B on the MMTS-Bench          (b) Thinking versus Non-thinking

Figure 6: (a) presents the evaluation results of Qwen2.5-VL-7B across four subsets. (b) illustrates the accuracy gains of GPT-5 and Qwen3-32B on the **InWild** across different subtasks, comparing performance with and without the thinking mode enabled.

**CoT Reasoning Enhances LLMs Beyond Parameter Scaling in Time Series Analysis.** Experiments with the Qwen2.5 series in Table 15 (Appendix D) demonstrate a scaling-law effect (Kaplan et al., 2020): scaling from 7B to 14B parameters yields clear performance gains, but further growth to 32B provides marginal improvements, indicating diminishing returns for temporal reasoning tasks in time series analysis. In contrast, enabling CoT reasoning produces substantial improvements

across all **InWild** subtasks for Qwen3 and GPT-5, as shown in Figure 6b. The benefits are especially pronounced in temporal reasoning, where the average improvement surpasses that observed in feature analysis tasks, and additional evaluations on the other subsets confirm this trend. These findings highlight that scaling laws impose inherent limits on parameter-based gains, whereas activating CoT reasoning enables models to capture temporal dependencies more effectively. Therefore, future work should focus on leveraging CoT reasoning to enhance LLMs' performance on time series analysis, rather than relying solely on scaling up LLM parameter size.

## 5 CONCLUSION

We presented MMTS-Bench, a comprehensive benchmark comprising 2,424 TSQA pairs across four specialized subsets for evaluating multi-modal time series understanding and reasoning abilities. Our extensive evaluations reveal that general-purpose LLMs outperform TS-LLMs in cross-domain generalization, and that current LLMs struggle with fine-grained localization and complex reasoning. We also find that simply scaling model size yields diminishing returns; instead, performance is more effectively enhanced through multi-modal inputs and explicit reasoning strategies like CoT, with the backbone LLM's capability being the dominant success factor.

**Limitations and Future Work.** MMTS-Bench currently does not cover several traditional time series evaluation tasks, such as forecasting, imputation, anomaly detection, and representation learning, which are also important components of time series capability. The benchmark also relies heavily on English annotations, potentially missing insights from multilingual temporal reasoning. Future work includes: (1) extending to longer-horizon sequences beyond current context limits, (2) adding complementary suites for traditional time series tasks (forecasting, imputation, anomaly detection, and representation learning) to provide a more complete picture of model capability, (3) developing more effective time series–text alignment paradigms beyond current encoder architectures, and (4) exploring prompt and tool-use strategies that leverage statistical features more systematically. We hope this evaluation protocol will help steer the development of time series LLMs toward robust, generalizable time series understanding.

## REPRODUCIBILITY STATEMENT

We elaborate on the implementation details of our benchmark construction and experimental setup in this paper and the Appendix. To facilitate end-to-end reproduction, we release an anonymized repository containing all data and code at `https://anonymous.4open.science/r/MMTS-BENCH-BEF7/`. We will maintain the anonymized repository for the duration of the review and, upon acceptance, migrate to a public repository and archive a snapshot to support long-term availability.

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

TABLE OF CONTENTS

## A  DETAILS ABOUT MMTS-BENCH

### A.1  METRICS

We adopt a stratified evaluation scheme with specialized metrics tailored to each answer type within our answer space $\mathcal{A}$, ensuring comprehensive and fair assessment across diverse question formats.

**Categorical Evaluation ($\mathcal{A}_{mc}$ and $\mathcal{A}_{bf}$).** For both multiple-choice and binary-choice tasks, we utilize **Accuracy** as the primary evaluation metric, measuring exact match performance through the indicator function:

$$\text{Accuracy} = \mathbb{I}(\hat{A} = A_{gt}) \tag{1}$$

where $\mathbb{I}(\hat{A} = A_{gt}) = 1$ if the predicted answer $\hat{A}$ matches ground-truth $A_{gt}$, and 0 otherwise.

**Numerical Evaluation ($\mathcal{A}_{num}$).** For numerical tasks, we employ two complementary metrics to capture both continuous proximity and threshold-based precision. Relative Accuracy quantifies the relative proximity between predicted and ground-truth values, yielding a normalized score in $[0, 1]$ where 1 indicates perfect prediction:

$$\text{Relative Accuracy} = \max\left(1.0 - \frac{|\hat{A} - A_{gt}|}{|A_{gt}|}, 0.0\right) \tag{2}$$

Accuracy@10% provides a stricter binary evaluation criterion, determining whether the prediction falls within a 10% relative error tolerance:

$$\text{Accuracy@10\%} = \mathbb{I}\left(\frac{|\hat{A} - A_{gt}|}{|A_{gt}|} \leq 0.1\right) \tag{3}$$

The scoring function $\mathcal{M}(\hat{A}, A_{gt})$ introduced in our protocol is instantiated using these type-specific metrics, ensuring appropriate assessment based on answer type while maintaining consistency across the evaluation framework.

### A.2  REAL-WORLD DATASET SOURCES

The real-world component of our benchmark is constructed from the LOTSA (Woo et al., 2024a; Godahewa et al., 2021; Nguyen et al., 2023; Woo et al., 2024b) dataset collection. To ensure broad domain coverage while maintaining representativeness and high quality, we selected five major domains: Transport, Cloud Operations, Climate, Economics, and Healthcare. Representative datasets from these domains include Traffic Hourly, Alibaba Cluster Trace 2018, ERA5 2018, M4 Weekly, Hospital, and COVID deaths, with statistical parameters summarized in Table 4. Their details are as follows.

**Transport** (Traffic Hourly) The dataset originates from the California Department of Transportation. It records hourly highway occupancy rates from multiple sensors in the San Francisco Bay Area over a 48-month period (2015–2016). It contains 862 time series, each with 17,376 points in the range [0,1]. Because of the long time span and the strong seasonal patterns, we applied a sliding window

with a maximum length of 672 points. To reduce token usage and irrelevant precision for LLMs, values were scaled by a factor of 100 and rounded to two decimal places.

**Cloud Operations** (Alibaba Cluster Trace 2018) This dataset describes CPU and memory utilization in a cluster of about 4,000 machines over eight days (from January 2 to January 8, 2018), sampled at five-minute intervals. It consists of 58,409 pairs of time series. Theoretical sequence length is 1,728 points, although some sequences are shorter due to missing samples (100–1,728 points). Values are within [0,100]. Because sequence lengths are moderate, no windowing was applied. Instead, we randomly sampled sequences and retained two decimal places.

**Climate** (ERA5 2018) The dataset comes from the European Centre for Medium-Range Weather Forecasts. It provides hourly global reanalysis data for 2018 at $2.8125°$ resolution ($64×128$ grid points), covering 45 variables across seven pressure levels (50, 250, 500, 600, 700, 850, and 925 hPa). Each time series pair has 8,736 points. To construct our benchmark subset, we selected relative humidity and temperature from the seven pressure levels, with values within [0,100]. To capture spatial diversity, we randomly sampled 50 locations worldwide and then applied sliding windows of length 720. All values were rounded to two decimal places.

**Economics** (M4 Weekly) The dataset is a subset of the M4 Competition (2018), which consists of 100,000 time series across different frequencies. The weekly subset includes 359 economic and business-related series, such as sales, demand, and index values. Sequence lengths range from 80 to 2,597 points. To preserve potential seasonalities and balance sequence lengths, we used a sliding window with a maximum length of 104 points, approximately two years in length. Shorter series were kept in full. Values were rounded to two decimal places.

**Healthcare** (Hospital & COVID deaths) The Hospital dataset records monthly patient counts related to medical products from January 2000 to December 2006. It contains 767 series of length 72. We applied sliding windows with common monthly cut lengths of 36, 60, and 72 points. All values were rounded to two decimal places.The COVID deaths dataset is sourced from the Johns Hopkins University repository. It contains cumulative daily death counts for countries and regions from January 22 to August 20, 2020. It consists of 266 daily series, each 182 points long. We applied a sliding window with a maximum length of 180 points. All values were rounded to two decimal places.

Table 4: Statistical parameters of subsets in LOTSA.

| Dataset | Domain | Frequency | #Time Series | #Obs. | #Vars |
|---|---|---|---|---|---|
| Traffic Hourly | Transport | H | 862 | 14,978,112 | 1 |
| Alibaba Cluster Trace 2018 | CloudOps | 5T | 58,409 | 95,192,530 | 2 |
| ERA5 2018 | Climate | H | 245,760 | 2,146,959,000 | 45 |
| M4 Weekly | Economics | W | 359 | 366,912 | 1 |
| Hospital | Healthcare | M | 767 | 55,224 | 1 |
| COVID Deaths | Healthcare | D | 266 | 48,412 | 1 |

A.3 DATASETS CLASSIFICATION

To systematically evaluate large models' capabilities across the proposed multi-dimensional framework, we construct four specialized subsets that collectively form the core of the MMTS-Bench. The design of these subsets is based on a core hypothesis: a model's understanding of time series is hierarchical and progressive, where deficiencies in foundational analytical abilities lead to systematic biases in higher-level reasoning tasks. To operationalize this hypothesis, we decompose temporal understanding into five orthogonal core dimensions.

**In this paper, the term "orthogonal" denotes functional distinctness rather than statistical independence.** We decompose time-series understanding into several information dimensions, each representing a unique and non-substitutable processing requirement. Since a single task typically relies on only a subset of these dimensions, we organize and annotate our tasks accordingly to prevent mixed or redundant definitions. The notion of hierarchy means that higher-level capabilities are the result of combining multiple dimensions, rather than introducing entirely new dimensions. Although

| Dimensions | Subtasks | Definition | Related Subsets |
|---|---|---|---|
| Structural Awareness | Non-Stationarity
Local-Global
Univariate-Multivariate | Analyzes statistical properties of concatenated subsequences.
Locates and analyzes specific sequence segments.
Processes and analyzes multiple time series data jointly. | Base |
| Feature Analysis | Trend Analysis
Seasonality Analysis
Noise Analysis
Volatility Analysis
Basic Analysis | Identifies long-term directional patterns and trend strength.
Captures seasonal patterns and seasonality strength.
Distinguishes random fluctuations from signal components.
Quantifies temporal variability and instability.
Computes fundamental statistics (mean, variance, range, etc.). | Base, InWild |
| Temporal Reasoning | Deductive Reasoning
Inductive Reasoning
Causal Reasoning
Analogical Reasoning
Counterfactual Reasoning | Applies general rules to infer properties of specific intervals.
Generalizes characteristics from observed sequences.
Identifies causal or lead-lag relationships between series.
Infers similarity by comparing temporal patterns.
Predicts outcomes under hypothetical changes. | InWild |
| Sequence Matching | Isomorphic Matching
Robust Matching
Positioning Match
Reverse Matching | Finds the most similar sequence under equal-length constraints.
Robustly matches patterns under preprocessing transformations.
Locates target patterns within longer sequences.
Recognizes similarity under temporal reversal. | Match |
| Cross-Modal Understanding | Time-series to Semantic
Semantic to Time-series | Converts time series patterns into textual descriptions.
Maps textual descriptions to corresponding time series data. | Align |

Table 5: Orthogonal dimensions of time series tasks described by MMTS-Bench, their covered subtask types, and related subsets.

the dimensions in the Base and InWild subsets are related, we deliberately assign them distinct roles. The Base subset is designed to isolate and "activate" individual dimensions (e.g., structural awareness) in controlled settings. In contrast, InWild tasks, drawn from realistic scenarios, necessitate the simultaneous integration of multiple dimensions. By explicitly annotating these dimensional dependencies, we enable fine-grained failure analysis: poor performance on an InWild task can be traced to deficiencies in specific dimensions (e.g., structural awareness), rather than being attributed to an undifferentiated notion of overall failure.

To this end, we employ a dual-tier evaluation architecture: **foundational capability assessment** using synthetic data, followed by **advanced capability assessment** using real-world data. This approach ensures both precise, controlled evaluation of core competencies and a realistic assessment of practical performance.

**Base.** This subset is constructed using precisely controlled synthetic data, eliminating confounding variables present in real-world data, to provide standardized, fine-grained evaluation of a model's foundational time series analysis capabilities in a controlled environment. It contains 700 QA pairs that encompass two core dimensions of Structural Awareness ($D_s$) and Feature Analysis ($D_f$).

The following three subsets are constructed from real-world data in the LOTSA benchmark to evaluate advanced capabilities in complex scenarios.

**InWild.** This subset is constructed from across five specialized domains (Transport, CloudOps, Climate, Econ/Fin, Healthcare) to evaluate a model's capabilities in advanced time series understanding and reasoning. Through combinatorial arrangements of three core dimensions—Structural Awareness ($D_s$), Feature Analysis ($D_f$), and Temporal Reasoning ($D_r$)—it generates 1,084 QA pairs covering 140 subtask types.

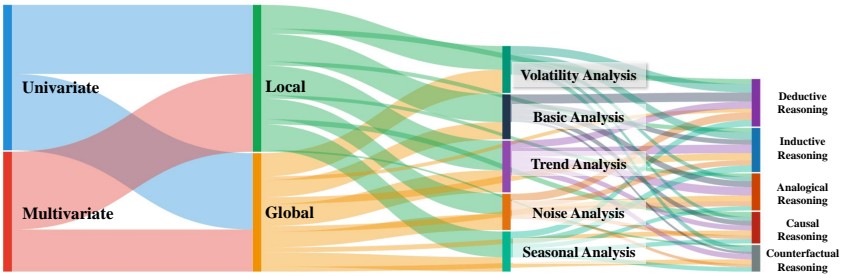

Figure 7: Sankey Diagram of Subtask Labels in the InWild Subset. This diagram illustrates the relationships and transitions between subtask labels in the InWild subset, highlighting their interdependencies in a clear and intuitive way.

**Match.** This subset is constructed from Transport and Econ/Fin domains to evaluate a model's performance in time series similarity matching and morphological correspondence. We generated four categories of sub-tasks by varying the Sequence Matching ($D_m$) dimensionality while holding Structural Awareness ($D_s$) dimensionality—stationarity, global scope, and univariate series—constant. The subset contains 400 QA pairs, with 100 pairs in each category.

**Align.** This subset is constructed from across five specialized domains (Transport, CloudOps, Climate) to systematically evaluate models' capabilities in bidirectional understanding and cross-modal conversion between numerical time series and natural language. From the perspective of the Cross-Modal Understanding dimension ($D_c$), we constructed 240 bidirectional QA pairs based on homologous time series sequences.

This hierarchical dataset architecture enables MMTS-Bench to provide comprehensive evaluation spanning from foundational analytical skills to advanced analytical capabilities, providing detailed diagnostic capability profiles that identify specific strengths and weaknesses of models in the context of time series understanding and reasoning tasks.

---

[5]For the detailed comparison table, please refer to https://anonymous.4open.science/r/MMTS-BENCH-BEF7/comparison.md

Table 6: This table presents a horizontal comparison of existing datasets for time series understanding and reasoning. The **TS Type** column denotes the source of time series data, where "R" refers to real-world data and "S" refers to synthetic data. The **Domain** column indicates the application domain of the time series. **Question Type** specifies the types of questions included in the dataset, such as Choice or Numerical. **Taxonomy** shows how each dataset categorizes different capability dimensions. **Input Method** describes how time series are processed in these works, including TS-as-text, TS-encoded and TS tokens. **Cross-Var.** indicate whether the dataset contains cross-variable analysis tasks, respectively. Finally, the **Generation** column outlines the construction methods of different datasets.[5]

| Dataset | TS Type | Domain | Size | Question Type | Taxonomy | Reasoning | Input Method | Cross-Var. | Generation |
|---------|---------|--------|------|---------------|----------|-----------|--------------|------------|------------|
| EngineMT-QA | R | Single | 11K/– | Choice | Flat | ✓ | TS Encoded | ✗ | Templates + Polishment |
| Time-MQA-TSQA | R | Multiple | 200K/1.4K | Choice+Num. | Flat | ✓ | TS-as-Text | ✗ | Single-round prompting |
| ChatTime-TSQA | S | – | 48K/0 | Choice | Flat | ✗ | TS tokens | ✗ | Templates |
| TimeSeriesExam | S | – | 746/– | Choice | Flat | ✓ | – | ✓ | Templates |
| ChatTS | R+S | Multiple | 11K/525 | Choice+Num. | Flat | ✓ | TS Encoded | ✓ | TS Self-Evol |
| Chat-TS | R | Multiple | 3741/100 | Choice | Flat | ✓ | TS tokens | ✗ | Single-round prompting |
| MMTS-Bench | R+S | Multiple | 2524/1724 | Choice+Num. | Multi-level | ✓ | – | ✓ | 3-stage prompting + Templates |

## A.4 DATASET CONSTRUCTION METHOD

To comprehensively evaluate multimodal time series understanding capabilities, we develop a dual-pathway construction methodology combining synthetic and real-world data sources. The synthetic pathway employs modular component synthesis with systematic parameter control to enable controlled evaluation of fundamental time series properties. The real-world pathway leverages progressive conversational frameworks, similarity matching algorithms, and cross-modal conversion techniques to construct comprehensive evaluation benchmarks. This approach yields four specialized subsets: **Base** for controlled synthetic evaluation, and **InWild** for multi-dimensional reasoning, **Match** for similarity matching, and **Align** for cross-modal understanding using authentic LOSTA data.

### A.4.1 SYNTHETIC DATASET

The time series data within the **Base** subset is generated through a modular synthesis approach. This process begins with the creation of three fundamental primitive components: (1) **Trend Components**, which can be configured with specific directions and magnitudes; (2) **Seasonal Components**, a diverse range of periodicities and waveforms are supported, encompassing both standard patterns (e.g., sine waves) and complex composite waveforms representative of real-world scenarios; and (3) **Noise Components**, where various noise types are included.

The final time series is formed by the superposition of these components. This principle is formally expressed through an additive model where the generated time series $y_t$ is the sum of three weighted components:

$$y^{(t)} = \tau_{\text{trend}}^{(t)} + \tau_{\text{seasonal}}^{(t)} + \tau_{\text{noise}}^{(t)}, \quad t = 1, 2, \ldots, T. \tag{4}$$

Each component $\tau^{(t)}$ is synthesized by scaling a corresponding **base signal** $S(t)$ with a randomly sampled weight $w$, such that $\tau_{\text{trend}}^{(t)} = w_{\text{trend}} \cdot S_{\text{trend}}(t)$, and similarly for the other components. The generation of these base signals is governed by a set of configurable parameters to ensure diversity, as detailed below.

First, a global **Sequence Length** ($T$) for all base signals is determined by sampling from the interval $[128, 2048]$ (corresponding to min_length and max_length). Then, each base signal is constructed as follows:

- Trend Base Signal ($S_{\text{trend}}^{(t)}$) is designed to capture long-term, non-stationary behavior. It is modeled as an ARIMA(0,2,0) process, representing a second-order random walk:

$$S_{\text{trend}}^{(t)} = \sum_{n=1}^{t} \sum_{m=1}^{n} X_m, \quad \text{where} \quad X_t \sim \mathcal{N}(0, \sigma^2). \tag{5}$$

  For dataset generation, the smoothness of this trend is controlled by the second difference parameter, $\delta_s$ (delta_s), which corresponds to the variance $\sigma^2$ and is sampled from $[0.01, 0.1]$. The overall amplitude of the base signal is sampled from $[0.1, 1000]$.

- Seasonal Base Signal ($S_{\text{seasonal}}^{(t)}$) provides the series' periodic structure. It is constructed from standard periodic waveforms (e.g., sine, square, triangular):

$$S_{\text{seasonal}}^{(t)} = S_0\big(\text{mod}(t + \phi, T_0)\big), \tag{6}$$

where $S_0(\cdot)$ is the standard waveform, $\phi$ is a phase shift, and $T_0$ is the period. In our dataset, the period $T_0$ is set to one-fifth of the total sequence length (i.e., $T_0 = T/5$), and the amplitude is sampled from $[0.1, 1000]$.

- Noise Base Signal ($S_{\text{noise}}^{(t)}$) introduces various types of random fluctuations. It can be drawn from several distinct statistical distributions to simulate different scenarios:

$$S_{\text{noise}}^{(t)} = X_t, \quad X_t = \begin{cases} \mathcal{N}(\mu, \sigma^2), & \text{Gaussian white noise} \\ U(a, b), & \text{Uniform noise} \\ \text{round}(V_t/q) \cdot q, \ V_t \sim U(a, b), & \text{Quantization noise} \\ X_{t-1} + V_t, \ V_t \sim U(a, b), & \text{Random walk noise} \end{cases} \tag{7}$$

The specific parameters for each noise type (e.g., $\mu, \sigma^2$ for Gaussian, $a, b$ for Uniform) are configured to generate a variety of noise profiles. For the Quantization noise, the step size $q$ is specifically defined as one-tenth of the signal's amplitude.

- Component Weights ($\boldsymbol{w}$) After generating the three base signals, their respective contributions to the final time series are determined by a set of weights. The weights $\boldsymbol{w} = (w_{\text{trend}}, w_{\text{seasonal}}, w_{\text{noise}})$ are sampled from a symmetric Dirichlet distribution, $\boldsymbol{w} \sim \text{Dir}(\boldsymbol{\alpha})$ with $\boldsymbol{\alpha} = (1, 1, 1)$. This ensures an unbiased combination where the weights are positive and sum to one.

During the construction of the subset, the ground-truth labels for the four types of subtasks in the Feature Analysis ($D_f$) dimension are systematically generated based on the parameter system of the underlying primitive components. For the sub-tasks in the Structural Awareness ($D_s$) dimension, a differentiated construction strategy is adopted:

- **Univariate vs. Multivariate:** Multivariate series are generated to evaluate a model's differential capabilities in analyzing univariate versus multivariate statistical properties.

- **Local vs. Global:** Descriptive and localization-based tasks are created by defining sub-intervals within the synthetic time series, thereby testing a model's local and global perceptual abilities.

- **Stationarity vs. Non-stationarity:** Composite series are constructed by concatenating two sub-series with distinct statistical properties. These are then used in conjunction with feature analysis tasks to assess a model's proficiency in identifying non-stationarity.

QA pairs are systematically generated based on a set of 17 distinct templates. These templates cover a spectrum of tasks, ranging from qualitative feature analysis to quantitative numerical computation. Except for a subset of interval localization problems that require manual annotation, the vast majority of QA pairs are automatically generated. This template-based automation serves as the execution layer for our pipeline, the distinct advantages of which—specifically in terms of diversity and alignment precision compared to prior works—are detailed below.

**Advancements in Generation Pipeline.** While modular synthesis is a shared paradigm in recent works like TimeSeriesExam (Cai et al., 2024) and ChatTS (Xie et al., 2024), our pipeline introduces critical enhancements in diversity, precision, and alignment.

- **Generation Diversity:** Unlike TimeSeriesExam, which relies on simple base patterns (e.g., linear, exponential) and restricted sub-options, we adopt an STL-inspired decomposition into Trend, Seasonal, and Noise modules with rich control parameters. For instance, our trends are generated via second-order random walks (controlled by $\delta^2$ and initial values) rather than simple functions, and we incorporate diverse noise types (e.g., quantized, random-walk) and flexible combination strategies (Additive, Concatenated) that surpass the fixed combination schemes seen in ChatTS.

- **Precision and Alignment:** We log precise quantitative parameters (e.g., specific waveform configurations) rather than qualitative labels. This granular logging allows the QA generation engine to leverage exact numerical values, enabling the construction of nuanced evaluation items—such as differentiating between weak, medium, and strong seasonality based on the `seasonal_strength` parameter—rather than limiting assessment to binary presence/absence questions.

### A.4.2 REAL-WORLD DATASET

To comprehensively evaluate multimodal time series understanding across diverse analytical dimensions, we construct three specialized subsets from real-world LOSTA data targeting distinct aspects of time series analysis: comprehensive QA-based evaluation (**InWild**), sequential similarity matching (**Match**), and cross-modal language understanding (**Align**). Each subset employs systematic construction pipelines with domain-specific sampling strategies and automated generation frameworks built upon authentic temporal data from five key domains.

INWILD SUBSET

This subset evaluates comprehensive understanding and reasoning capabilities in time series analysis through an innovative progressive, multi-turn conversational approach. The subset employs open-ended question templates to ensure rich diversity while avoiding rigid patterns. The generation process synergistically integrates three analytical dimensions: structural awareness, feature analysis, and temporal reasoning, creating 140 unique dimensional combinations that compel models to perform deep inference on underlying patterns and dynamic relationships.

**Three-Stage Progressive Generation Pipeline** Our dataset construction follows a systematic three-stage pipeline where each stage serves a distinct function in creating high-quality question-answering pairs.

- **Stage 1: Initialization and Context Generation** The process begins with random sampling to define core parameters including question type (multiple-choice or true/false), task type (combining feature analysis and temporal reasoning capabilities), number of variables (univariate or multivariate), and analysis scope (local sub-sequence or global series). The system provides Claude 3.7 Sonnet with comprehensive multimodal context: time series visualizations, raw numerical sequences, domain metadata, and pre-computed statistical features from specialized libraries. The model generates a structured textual description tailored to the specified task type.

- **Stage 2: Task Specification and Reasoning Construction** Building upon the initial description, this stage further specifies the task according to predefined variables and analysis scope. For local-scope questions, the system provides indices for key time points. The model abstracts the structured information to construct specific questions, ground-truth answers, and detailed reasoning explanations.

- **Stage 3: Formatting and Quality Verification** The model formats outputs into standardized QA pairs according to designated question types while performing automated consistency verification across three dimensions: mathematical soundness of logic and calculations, descriptive accuracy between textual elements and data, and logical interpretability of reasoning coherence.

MATCH SUBSET

The **Match** subset is specifically designed to evaluate models' capabilities in time series similarity matching and morphological correspondence analysis. The subset employs standardized question-answer templates with fixed structural awareness parameters (stationarity, univariate, and global sequence) while systematically varying sequence matching dimensions to create four distinct matching paradigms.

**Four Matching Tasks**: (1) **Isomorphic Matching** evaluates models' fundamental ability to identify sequential similarity under identical temporal scales and data distributions, primarily assessing recognition accuracy of statistical characteristics and dynamic patterns; (2) **Robust Matching** evaluates models' resilience to maintain matching accuracy under data preprocessing transformations

such as moving average smoothing, testing adaptability to data quality variations; (3) **Localization Matching** assesses models' precision in identifying target temporal patterns within extended time windows, focusing on temporal pattern retrieval and spatial localization performance; (4) **Reverse Matching** evaluates models' adaptability to temporal direction transformations, testing sequence correspondence recognition under time-reversed conditions.

These four progressive difficulty gradients comprehensively examine models' integrated capabilities in time series similarity measurement, pattern alignment, and morphological recognition under various constraints and challenging scenarios.

**Dataset Construction Pipeline** The construction process begins with segmenting sequences based on their typical periodicity patterns. Using a sliding window approach, we calculate Dynamic Time Warping (DTW) distances to identify four segments: minimum DTW distance, median distance, maximum distance, and second-maximum distance. We then randomly shuffle the subsequences of these four segments to serve as the four options in question–answer pairs. We construct the final dataset using standardized QA templates, covering four sequence matching tasks across two domains.

ALIGN SUBSET

The **Align** subset evaluates models' bidirectional conversion capabilities between time series data and natural language. It is intentionally designed as an alignment calibration subtask, aiming to assess whether a model can accurately match time series with natural language descriptions under conditions of low linguistic ambiguity. Consequently, the descriptions are intentionally specific regarding trend ranges and magnitude changes, ensuring the evaluation focuses on precise cross-modal alignment.

**Bidirectional Cross-Modal Tasks** (1) **Time-series to Semantic Conversion** requires models to identify which textual description best matches a given time series' statistical characteristics and dynamic patterns. (2) **Semantic to Time-series Conversion** requires models to select the temporal sequence that best corresponds to a given natural language description of trends, fluctuations, and periodicity.

**Dataset Construction Pipeline** The subset employs symmetric construction ensuring task consistency. The data originates from three real-world domains: Traffic, CloudOps, and Climate. To prevent shortcut learning based on domain or value range, we implemented two strategies:

- **Value Scaling:** CloudOps and Climate series are scaled to similar ranges to mitigate magnitude-based biases.
- **Controlled Sampling:** For each correct sample, we select three distractors. We ensure that at least one distractor comes from the same domain and shares a similar statistical distribution as the correct answer. This forces models to rely on fine-grained patterns and numeric details rather than domain priors.

For time-series to semantic tasks, Claude 3.7 Sonnet generates feature descriptions for four temporal samples, which serve as answer options. For semantic to time-series tasks, the model generates structured descriptions of correct samples, which become question prompts with temporal sequences as options.

HUMAN-IN-THE-LOOP CURATION

All real-world datasets undergo rigorous verification by a panel of ten time series analysis experts, each employing specialized quality control procedures. For TSQA, experts review generated QA pairs for academic rigor, retaining high-quality samples while flagging substandard pairs for regeneration under identical parameter configurations. For **Match**, verification focuses on eliminating samples with similar DTW distances but fundamentally different morphological patterns through combined manual assessment and statistical metric analysis. For **Align**, experts evaluate the accuracy and coherence of LLM-generated textual descriptions, correcting or removing content that fails to accurately represent temporal characteristics. This comprehensive human-in-the-loop curation ensures dataset quality and practical applicability across all evaluation benchmarks.

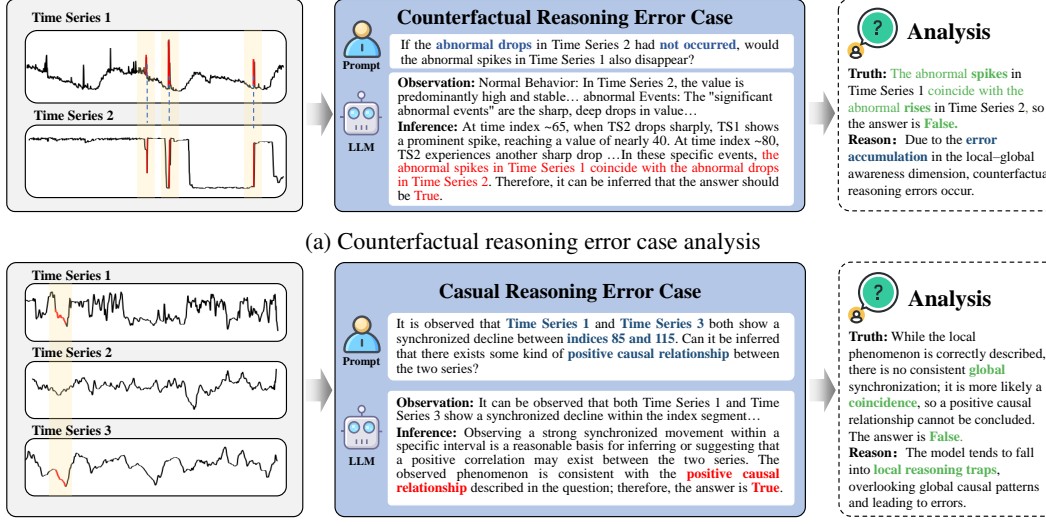

(a) Counterfactual reasoning error case analysis

(b) Casual reasoning error case analysis

Figure 8: The top subfigure illustrates a counterfactual reasoning error case, whereas the bottom subfigure illustrates a causal reasoning error case. In both subfigures, the left panel visualizes the time series data, the central panel presents the questions together with the LLM responses, and the right panel reports the ground-truth answers accompanied by error analysis. Red text highlights the erroneous reasoning points, green text denotes the correct reasoning points, and blue text marks the keywords from the questions.

## B STANDARDIZED TS INPUT FORMAT

### B.1 EVALUATION PROMPT

We use the following prompts to standardize multiple-choice QA and numerical QA evaluation. The system prompt mandates the answer format and ambiguity policy; the user prompt injects per-item content. The model's output is scored by extracting the single letter inside the `<final_answer>` tag.

### B.1.1 SYSTEM PROMPT FOR MULTIPLE-CHOICE QA

```
1  You are an expert AI assistant specialized in answering numerical
       questions with high accuracy and consistency. Your task is to analyze
        questions carefully and provide precise numerical answers.
2
3  IMPORTANT INSTRUCTIONS:
4  1. You must provide a numerical answer (integer, decimal, or scientific
       notation)
5  2. Your final answer must be a single number only
6  3. If the question seems ambiguous, choose the most reasonable
       interpretation
7  4. Do not include units unless specifically requested in the question
8
9  RESPONSE FORMAT:
10 You must structure your final answer exactly as follows:
11
12 <final_answer>
13 [State your numerical answer as a single number only]
14 </final_answer>
15
16 Remember: Your final answer should contain ONLY the numerical value,
       nothing else.
```

Listing 1: System Prompt for multiple-choice QA

### B.2 SYSTEM PROMPT FOR NUMERICAL QA

```
1  You are an expert AI assistant specialized in answering multiple-choice
       questions with high accuracy and consistency. Your task is to analyze
        questions carefully and provide clear, well-reasoned answers.
2
3  IMPORTANT INSTRUCTIONS:
4  1. You must select your answer from the given options only
5  2. Your final answer must be a single letter (A, B, C, D, etc.)
6  3. If the question seems ambiguous, choose the most reasonable
       interpretation
7  4. Do not make up information not provided in the question
8
9  RESPONSE FORMAT:
10 You must structure your final answer exactly as follows:
11
12 <final_answer>
13 [State your chosen option as a single letter: A, B, C, or D]
14 </final_answer>
15
16 Remember: Your final answer should contain ONLY the letter of your chosen
       option, nothing else.
```

Listing 2: System Prompt for numerical QA

### B.2.1 USER PROMPT

```
Please answer the following multiple-choice/numerical question based on
    the given information:

Question: {question}

{given_values_str}

Available Options: {option}

Please analyze this question carefully, consider the given value and all
    available options, then provide your answer following the exact
    format specified in the system instructions.
```

Listing 3: User Prompt used for Evaluation

### B.3 TIME SERIES TO IMAGE CONVERSION

We follow the plotting style of Zhuang et al. Zhuang et al. (2024) and adapt it for multi-channel time series. Specifically, we preserve the single-channel resolution ($1500 \times 320$ at 100 dpi) and scale the figure height linearly with the number of channels by stacking channel-wise subplots with a fixed per-channel height (320 px at 100 dpi). This keeps a consistent time axis across channels while maintaining comparable vertical resolution per channel.

Listing 4: Python code for converting time series data into images

```python
def plot_time_series_as_image(value_list):
    if len(value_list) > 8:  # single-channel time series
        num_channels = 1
    else:  # multi-channel time series
        num_channels = len(value_list)

    # Figure parameters: base width and per-channel height
    # Single channel: 1500x320; increase height by 320 for each
        additional channel
    width_inches = 15.0  # 1500 pixels / 100 dpi = 15 inches
    height_per_channel = 3.2  # 320 pixels / 100 dpi = 3.2 inches
    total_height = height_per_channel * num_channels

    plt.figure(figsize=(width_inches, total_height), dpi=100)

    if num_channels == 1:  # single-channel time series
        plt.plot(range(len(value_list)), value_list, 'b-', linewidth=1.5)
        plt.title('Time Series', fontsize=12)
        plt.xlabel('Time Index', fontsize=10)
        plt.ylabel('Value', fontsize=10)
        plt.grid(True, alpha=0.3)
        plt.xlim(0, len(value_list) - 1)
    else:  # multi-channel time series
        for i, channel_data in enumerate(value_list):
            plt.subplot(num_channels, 1, i + 1)
            plt.plot(range(len(channel_data)), channel_data, 'b-',
                linewidth=1.5)
            plt.title(f'Time Series {i+1}', fontsize=10)
            plt.xlabel('Time Index', fontsize=8)
            plt.ylabel('Value', fontsize=8)
            plt.grid(True, alpha=0.3)
            plt.xlim(0, len(channel_data) - 1)
    plt.tight_layout()
```

# C ABLATION STUDY

Across all tasks in the MMTS-Benchmark, *ChatTS* demonstrates the best performance within the open-source TS-LLM category, showcasing robust time series analysis and reasoning capabilities. To further investigate the key factors that influence TS-LLM performance and provide insights for future research, we modified the official ChatTS training pipeline [6], adopting their released training data and recommended training strategy. We conducted controlled ablations on the **encoder architecture and size, positional encoding strategies, LLM backbone size, and prompt prefix design**.

## C.1 EXPERIMENTAL SETUP

In its original implementation, ChatTS employs Qwen2.5-14B-Instruct as the backbone LLM, with a 5-layer MLP serving as the time series encoder. During training, textual embeddings are aligned with time series embeddings to equip the model with time series reasoning capabilities. To examine the role of the encoder, we replaced the MLP with alternative architectures, including CNN and Transformer encoders with variable depth. We further tested the effect of introducing learnable positional embeddings or index-based positional features into the time series input.

Due to computational constraints, our experiments use Qwen2.5-3B-Instruct as the backbone, and we also report its text-only baseline performance on MMTS-Benchmark. For comparison, we include performance of Qwen2.5-14B-Instruct, allowing us to isolate the effect of LLM backbone size. Finally, since ChatTS incorporates a prompt prefix that contains statistical information (e.g., offset, scale factor, length, max/min values, left/right boundary values), we tested models trained with and without this prefix to measure its contribution.

All models were evaluated on **InWild**, **Match**, and **Align**. While we closely followed the ChatTS training methodology, inevitable differences arise due to random training data mixing, limited compute budgets, and variations in model size and hyperparameters. Nonetheless, the relative comparisons across ablations yield consistent and reliable conclusions.

## C.2 EVALUATION RESULTS

We categorize the factors related to the time series encoder into three dimensions: **(i) encoder architecture**, **(ii) encoder size**, and **(iii) positional encoding**. For the architecture study, we compared a 5-layer MLP (17.1M parameters), a CNN (50.3M), and a Transformer (6.3M) as the TS Encoders of our TS-LLMs. As shown in Table 7, the results indicate that model performance is largely insensitive to encoder architecture, with only marginal differences across tasks. Relative to the Qwen2.5-3B baseline, trained models exhibit no significant improvements on **InWild** and **Match**, but achieve clear gains on Sem→TS while degrading on TS→Sem. This suggests that the encoder introduces a directional bias in learning, which may be related to the distributional characteristics of the training data.

Table 7: Performance of TS-LLMs with different time series encoder architectures. We compare a 5-layer MLP, CNN, and Transformer as encoders, with their parameter sizes indicated in parentheses. The baseline is Qwen2.5-3B-Instruct, which treats the time series as plain text input.

| Dataset | Baseline | MLP(17.1M) | CNN(50.4M) | Transformer(6.3M) |
|---------|----------|------------|------------|-------------------|
| InWild  | 38.75    | 38.25      | 37.36      | 38.84             |
| Match   | 27.45    | 30.67      | 28.42      | 30.00             |
| Sem→TS  | 49.17    | 59.44      | 60.83      | 60.56             |
| TS→Sem  | 64.17    | 45.56      | 44.44      | 47.50             |

To further examine scaling effects within a fixed architecture, we tested MLP encoders of varying depths (1, 3, 5, and 7 layers), as reported in Table 8. For reference, we also include the Qwen2.5-3B baseline and the original ChatTS(14B) checkpoint released on Hugging Face.[7] Results show

---

[6] https://github.com/xiezhe-24/ChatTS-Training
[7] https://huggingface.co/bytedance-research/ChatTS-14B

that increasing the number of MLP layers does not yield a monotonic improvement, indicating that simply enlarging the encoder does not directly translate into better performance. In contrast, comparing models with 3B and 14B backbones reveals consistent improvements of 10%–30% across tasks for the larger ones. This highlights the dominant role of the backbone's intrinsic reasoning capacity in determining TS-LLM performance. Besides, a comparison with the unaligned Qwen2.5 backbones yields results consistent with the above trend: while InWild and Match remain largely unchanged(except ChatTS in Match), aligned models achieve clear improvements on Sem→TS but show noticeable degradation on TS→Sem, thereby further validating our observation.

Table 8: Ablation study on the number of layers in time series encoders. We evaluate different layer counts for the MLP encoder, comparing performance with the baseline models Qwen2.5-3B-Instruct and Qwen2.5-14B-Instruct, and the ChatTS model. For the ChatTS model, we use the weights released by the original authors on Hugging Face, with Qwen2.5-14B-Instruct as the backbone and a 5-layer MLP as the time series encoder.

| Dataset | Qwen2.5-3B | 1 Layer(0.3M) | 3 Layers(8.7M) | 5 Layers(17.1M) | 7 Layers(25.5M) | Qwen2.5-14B | ChatTS |
|---|---|---|---|---|---|---|---|
| InWild | 38.75 | 38.90 | 38.44 | 38.25 | 37.45 | 52.84 | 50.28 |
| Match | 27.45 | 30.67 | 30.42 | 30.67 | 33.00 | 56.55 | 36.80 |
| Sem→TS | 49.17 | 60.83 | 51.67 | 59.44 | 61.39 | 86.83 | 91.17 |
| TS→Sem | 64.17 | 45.28 | 44.44 | 45.56 | 45.28 | 88.50 | 68.33 |

We also evaluated the effect of positional encoding strategies, following the three configurations in the official training code: no positional encoding, learnable embeddings appended to the input series, and normalized index values concatenated with the input series. Using a 3-layer MLP encoder, the results are reported in Table 9. These findings suggest that positional encoding design also has only a limited impact on performance compared with the backbone scale.

Table 9: Performance of models with different positional encoding strategies. `no_emb` denotes no positional encoding, `pos_emb` denotes learnable embeddings, and `pos_idx` denotes normalized index values used as positional encoding.

| Dataset | no_emb | pos_emb | pos_idx |
|---|---|---|---|
| InWild | 39.11 | 38.44 | 39.42 |
| Match | 31.00 | 30.42 | 28.67 |
| Sem→TS | 58.83 | 51.67 | 52.22 |
| TS→Sem | 42.22 | 44.44 | 45.00 |

Finally, we investigated the role of the prompt prefix introduced in ChatTS, which encodes statistical descriptors of the time series (e.g., offset, scale factor, length, min/max, and boundary values). We compared models trained with and without the prefix using a 5-layer MLP encoder, and additionally tested a model trained without the prefix but provided with the prefix at inference time. Results (Table 10) demonstrate that the statistical prompt prefix has a significant impact on model performance, especially in Align tasks, likely because it provides auxiliary information that enhances both interpretability and reasoning efficiency.

Table 10: Performance of models trained with and without the prompt prefix. **ON** indicates that the prefix is used during both training and testing; **OFF** indicates that it is used in neither; and **OFF**$^*$ denotes models trained without the prefix but evaluated with it.

| Dataset | ON | OFF | OFF$^*$ |
|---|---|---|---|
| InWild | 38.25 | 36.53 | 37.01 |
| Match | 30.67 | 25.50 | 33.41 |
| Sem→TS | 59.44 | 24.17 | 59.72 |
| TS→Sem | 45.56 | 21.39 | 45.83 |

## C.3 CONCLUSION ON ABLATIONS

Our ablation findings can be summarized as follows:

- **Limited Encoder Contribution.** Under the current alignment paradigm, encoder architecture, scale, and positional encoding have only marginal effects. More effective paradigms for time series–text alignment remain an open challenge.

- **Backbone Dominance.** The LLM backbone size is the primary determinant of performance; scaling the backbone directly boosts temporal reasoning ability.

- **Prompt Engineering Effectiveness.** Incorporating statistical information into prompts substantially enhances model inference, suggesting that prompt engineering is a promising direction for strengthening TS-LLM reasoning. Future work should explore alternative prompt formats and auxiliary signals.

# D    FULL RESULTS

Table 11: The *Accuracy* metric of different models on the **Base** subset's Choice split. $^{cot}$ denotes *thinking* mode. [1] denotes models evaluated without any time-series input. [2] denotes ChatTS without built-in statistical computation module. $-\text{VL}$ = Vision-Language. 'TS' stands for Time Series modality, as time-series-specific models introduce a TS encoder. '–' indicates that the model failed to respond correctly. **Bold underlined** values indicate the best performance within each category for each metric, and **bold** values indicate the second-best performance. Stat. and Non-Stat. columns represent the questions with staionary and non-stationary time series, respectively.

| Category | Model Name | Modality | Total | Trend | Seasonality | Noise | Local | Overall |
|---|---|---|---|---|---|---|---|---|
| Open-source | DeepSeek-V3 | Text | 0.41 | 0.49 | 0.37 | 0.33 | **0.53** | 0.49 |
| | Kimi-K2 | Text | **0.45** | **0.50** | 0.40 | **0.39** | **0.55** | **0.50** |
| | Qwen3-32b$^{cot}$ | Text | **0.42** | **0.53** | **0.45** | 0.37 | 0.32 | **0.53** |
| | Qwen3-32b | Text | 0.40 | 0.46 | 0.37 | 0.35 | 0.47 | 0.46 |
| | Qwen3-8b$^{cot}$ | Text | 0.35 | 0.38 | 0.40 | 0.33 | 0.26 | 0.38 |
| | Qwen3-8b | Text | 0.32 | 0.33 | 0.27 | 0.26 | 0.50 | 0.33 |
| | Qwen2.5-32b | Text | 0.35 | 0.39 | 0.28 | 0.30 | 0.48 | 0.39 |
| | Qwen2.5-14b | Text | 0.35 | 0.39 | 0.31 | 0.27 | 0.46 | 0.39 |
| | Qwen2.5-7b | Text | 0.33 | 0.44 | 0.22 | 0.28 | 0.46 | 0.44 |
| | Qwen2.5-32b[1] | Text | 0.30 | 0.39 | 0.23 | 0.19 | 0.44 | 0.39 |
| | Qwen2.5-7b-VL | Text | 0.29 | 0.35 | 0.23 | 0.24 | 0.36 | 0.35 |
| | Qwen2.5-7b-VL | Vision | 0.32 | 0.32 | **0.41** | 0.28 | 0.29 | 0.32 |
| | Qwen2.5-7b-VL | V+T | 0.34 | 0.41 | 0.34 | 0.27 | 0.34 | 0.41 |
| Closed-source | Claude-3.7-Sonnet | Text | 0.49 | **0.59** | 0.54 | 0.35 | 0.52 | **0.59** |
| | Claude-Sonnet-4 | Text | 0.49 | 0.57 | 0.49 | 0.35 | **0.63** | 0.57 |
| | Gemini-2.5-Pro | Text | 0.48 | 0.50 | 0.51 | 0.39 | 0.59 | 0.50 |
| | Gemini-2.5-Flash | Text | 0.39 | 0.49 | 0.36 | 0.24 | 0.56 | 0.49 |
| | GPT-5-Minimal | Text | 0.45 | 0.45 | 0.40 | **0.42** | 0.56 | 0.45 |
| | GPT-5-High | Text | **0.51** | 0.53 | 0.51 | **0.45** | **0.60** | 0.53 |
| | GPT-4o | Text | 0.42 | 0.51 | 0.36 | 0.34 | 0.52 | 0.51 |
| | GPT-4o | Vision | **0.55** | **0.60** | **0.64** | **0.42** | 0.56 | **0.60** |
| | GPT-4o | V+T | **0.51** | 0.53 | **0.60** | 0.39 | 0.56 | 0.53 |
| | GPT-4o-mini | Text | 0.36 | 0.43 | 0.34 | 0.28 | 0.44 | 0.43 |
| Time-series | ChatTS | TS | 0.39 | 0.42 | 0.39 | 0.31 | 0.49 | 0.37 |
| | ChatTS[2] | TS | 0.39 | 0.37 | 0.39 | 0.34 | 0.53 | 0.36 |
| | ITFormer | TS | 0.31 | 0.30 | 0.29 | 0.28 | 0.42 | 0.29 |
| | ChatTime | TS | – | – | – | – | – | – |

Table 12: The *Accuracy@10%* metric of different models on the **Base** subset's numerical split. [cot] denotes *thinking* mode. [2] denotes ChatTS without built-in statistical computation module. -VL = Vision-Language. 'TS' stands for Time Series modality, as time-series-specific models introduce a TS encoder. '–' indicates that the model failed to respond correctly. **Bold underlined** values indicate the best performance within each category for each metric, and **bold** values indicate the second-best performance. Stat. and Non-Stat. columns represent the questions with staionary and non-stationary time series, respectively.

| Category | Model Name | Modality | Total | Trend | Seasonality | Basic | Stat. | Non-Stat. | Local | Overall | Uni-Var. | Multi-Var. |
|---|---|---|---|---|---|---|---|---|---|---|---|---|
| Open-source | DeepSeek-V3 | Text | **0.41** | **0.08** | 0.02 | **0.57** | **0.57** | 0.52 | 0.21 | **0.47** | **0.57** | **0.56** |
| | Kimi-K2 | Text | **0.42** | 0.04 | **0.13** | **0.56** | **0.56** | **0.56** | 0.19 | **0.45** | **0.56** | **0.54** |
| | Qwen3-32b[cot] | Text | 0.31 | 0.00 | **0.10** | 0.40 | 0.40 | 0.40 | 0.20 | 0.32 | 0.40 | 0.33 |
| | Qwen3-32b | Text | 0.35 | 0.02 | 0.03 | 0.47 | 0.47 | 0.50 | 0.07 | 0.38 | 0.47 | 0.42 |
| | Qwen3-8b[cot] | Text | 0.27 | 0.00 | 0.00 | 0.34 | 0.34 | 0.38 | 0.13 | 0.27 | 0.34 | 0.29 |
| | Qwen3-8b | Text | 0.26 | 0.02 | 0.03 | 0.33 | 0.33 | 0.39 | 0.07 | 0.27 | 0.33 | 0.26 |
| | Qwen2.5-32b | Text | 0.34 | 0.03 | 0.00 | 0.45 | 0.45 | 0.50 | 0.11 | 0.36 | 0.45 | 0.44 |
| | Qwen2.5-14b | Text | 0.25 | 0.05 | 0.00 | 0.29 | 0.29 | 0.41 | 0.04 | 0.24 | 0.29 | 0.34 |
| | Qwen2.5-7b | Text | 0.18 | **0.06** | 0.00 | 0.19 | 0.19 | 0.31 | 0.01 | 0.16 | 0.19 | 0.31 |
| | Qwen2.5-7b-VL | Text | 0.14 | 0.03 | 0.03 | 0.14 | 0.14 | 0.23 | 0.04 | 0.12 | 0.14 | 0.21 |
| | Qwen2.5-7b-VL | Vision | 0.25 | 0.05 | 0.09 | 0.28 | 0.28 | 0.27 | **0.27** | 0.24 | 0.28 | 0.35 |
| | Qwen2.5-7b-VL | V+T | 0.19 | 0.05 | 0.04 | 0.21 | 0.21 | 0.27 | 0.09 | 0.18 | 0.21 | 0.31 |
| Closed-source | Claude-3.7-Sonnet | Text | **0.54** | 0.13 | 0.20 | **0.66** | **0.66** | 0.65 | 0.44 | **0.55** | **0.66** | **0.66** |
| | Claude-Sonnet-4 | Text | 0.53 | 0.04 | 0.33 | 0.62 | 0.62 | 0.61 | **0.53** | 0.50 | 0.62 | 0.66 |
| | Gemini-2.5-Pro | Text | **0.63** | 0.14 | **0.41** | **0.72** | **0.72** | **0.69** | **0.71** | **0.60** | **0.72** | **0.72** |
| | Gemini-2.5-Flash | Text | 0.43 | 0.04 | 0.20 | 0.56 | 0.56 | 0.55 | 0.20 | 0.45 | 0.56 | 0.40 |
| | GPT-5-Minimal | Text | 0.42 | 0.10 | 0.09 | 0.58 | 0.58 | 0.53 | 0.16 | 0.48 | 0.58 | 0.57 |
| | GPT-4o | Text | 0.41 | 0.06 | 0.09 | 0.53 | 0.53 | 0.58 | 0.13 | 0.43 | 0.53 | 0.51 |
| | GPT-4o | Vision | 0.28 | **0.17** | 0.10 | 0.34 | 0.34 | 0.29 | 0.31 | 0.31 | 0.34 | 0.34 |
| | GPT-4o | V+T | 0.51 | **0.42** | **0.37** | 0.59 | 0.59 | 0.57 | 0.31 | 0.56 | 0.59 | 0.56 |
| | GPT-4o-mini | Text | 0.33 | 0.02 | 0.00 | 0.40 | 0.40 | 0.51 | 0.09 | 0.32 | 0.40 | 0.43 |
| Time-series | ChatTS | TS | 0.37 | 0.36 | 0.43 | 0.41 | 0.41 | 0.42 | 0.12 | 0.40 | 0.41 | 0.40 |
| | ChatTS[2] | TS | 0.01 | 0.00 | 0.45 | 0.01 | 0.01 | 0.01 | 0.11 | 0.01 | 0.01 | 0.02 |
| | ITFormer | TS | 0.00 | 0.00 | 0.00 | 0.00 | 0.00 | 0.01 | 0.02 | 0.00 | 0.00 | 0.00 |
| | ChatTime | TS | – | – | – | – | – | – | – | – | – | – |

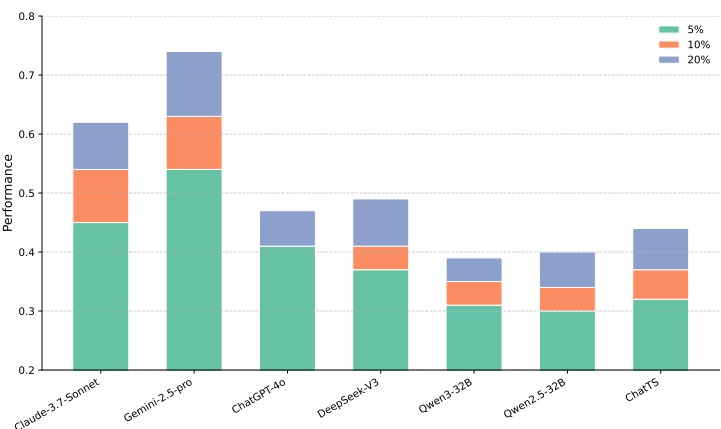

Figure 9: This stacked bar chart illustrates the *Accuracy@N%* performance of several representative models (closed-source, open-source, and TS-LLMs) on the numerical split of the **Base** subset. To explore the hierarchical distribution of numerical reasoning ability, we report results for N = 5, 10, 20.

Table 13: The *Relative Accuracy* metric of different models on the **Base** subset's numerical split. $^{cot}$ denotes *thinking* mode. $^2$ denotes ChatTS without built-in statistical computation module. $-\text{VL}$ = Vision-Language. 'TS' stands for Time Series modality, as time-series-specific models introduce a TS encoder. '–' indicates that the model failed to respond correctly. **Bold underlined** values indicate the best performance within each category for each metric, and **bold** values indicate the second-best performance. Stat. and Non-Stat. columns represent the questions with staionary and non-stationary time series, respectively.

| Category | Model Name | Modality | Total | Trend | Seasonality | Basic | Stat. | Non-Stat. | Local | Overall | Uni-Var. | Multi-Var. |
|---|---|---|---|---|---|---|---|---|---|---|---|---|
| Open-source | DeepSeek-V3 | Text | **0.59** | **0.27** | 0.32 | **0.73** | **0.73** | **0.66** | **0.41** | **0.63** | **0.73** | **0.76** |
| | Kimi-K2 | Text | **0.65** | **0.37** | **0.49** | **0.74** | **0.74** | **0.71** | **0.56** | **0.66** | **0.74** | **0.72** |
| | Qwen3-32b$^{cot}$ | Text | 0.42 | 0.00 | 0.30 | 0.51 | 0.51 | 0.52 | 0.27 | 0.41 | 0.51 | 0.47 |
| | Qwen3-32b | Text | 0.49 | 0.07 | 0.25 | 0.61 | 0.61 | 0.64 | 0.29 | 0.50 | 0.61 | 0.61 |
| | Qwen3-8b$^{cot}$ | Text | 0.37 | 0.00 | 0.19 | 0.47 | 0.47 | 0.49 | 0.19 | 0.37 | 0.47 | 0.43 |
| | Qwen3-8b | Text | 0.40 | 0.13 | 0.19 | 0.49 | 0.49 | 0.54 | 0.16 | 0.42 | 0.49 | 0.46 |
| | Qwen2.5-32b | Text | 0.51 | 0.16 | 0.13 | 0.66 | 0.66 | 0.63 | 0.30 | 0.56 | 0.66 | 0.64 |
| | Qwen2.5-14b | Text | 0.44 | 0.14 | 0.10 | 0.51 | 0.51 | 0.59 | 0.29 | 0.43 | 0.51 | 0.55 |
| | Qwen2.5-7b | Text | 0.35 | 0.20 | 0.12 | 0.40 | 0.40 | 0.49 | 0.14 | 0.36 | 0.40 | 0.50 |
| | Qwen2.5-7b-VL | Text | 0.32 | 0.16 | 0.15 | 0.37 | 0.37 | 0.43 | 0.12 | 0.33 | 0.37 | 0.42 |
| | Qwen2.5-7b-VL | Vision | 0.40 | 0.26 | **0.36** | 0.44 | 0.44 | 0.48 | 0.22 | 0.4 | 0.44 | 0.50 |
| | Qwen2.5-7b-VL | V+T | 0.19 | 0.05 | 0.04 | 0.21 | 0.21 | 0.27 | 0.09 | 0.18 | 0.21 | 0.31 |
| Closed-source | Claude-3.7-Sonnet | Text | **0.74** | 0.51 | 0.59 | **0.80** | **0.80** | **0.79** | 0.72 | **0.74** | **0.80** | **0.80** |
| | Claude-Sonnet-4 | Text | 0.73 | 0.36 | 0.61 | 0.78 | 0.78 | 0.79 | **0.79** | 0.69 | 0.78 | **0.80** |
| | Gemini-2.5-Pro | Text | **0.82** | **0.56** | **0.78** | **0.85** | **0.85** | **0.83** | **0.88** | **0.79** | **0.85** | **0.83** |
| | Gemini-2.5-Flash | Text | 0.57 | 0.10 | 0.55 | 0.69 | 0.69 | 0.66 | 0.34 | 0.57 | 0.69 | 0.53 |
| | GPT-5-Minimal | Text | 0.61 | 0.24 | 0.33 | 0.77 | 0.77 | 0.68 | 0.45 | 0.66 | 0.77 | 0.78 |
| | GPT-4o | Text | 0.60 | 0.32 | 0.35 | 0.73 | 0.73 | 0.7 | 0.38 | 0.64 | 0.73 | 0.67 |
| | GPT-4o | Vision | 0.58 | 0.30 | 0.71 | 0.60 | 0.60 | 0.55 | 0.63 | 0.55 | 0.60 | 0.57 |
| | GPT-4o | V+T | 0.70 | **0.59** | **0.75** | 0.76 | 0.76 | 0.70 | 0.60 | 0.73 | 0.76 | 0.74 |
| | GPT-4o-mini | Text | 0.50 | 0.17 | 0.14 | 0.60 | 0.60 | 0.66 | 0.31 | 0.51 | 0.60 | 0.61 |
| Time-series | ChatTS | TS | 0.55 | 0.46 | 0.79 | 0.59 | 0.59 | 0.56 | 0.34 | 0.56 | 0.59 | 0.56 |
| | ChatTS$^2$ | TS | 0.19 | 0.06 | 0.76 | 0.10 | 0.10 | 0.10 | 0.36 | 0.09 | 0.10 | 0.10 |
| | ITFormer | TS | 0.15 | 0.02 | 0.06 | 0.15 | 0.15 | 0.19 | 0.23 | 0.12 | 0.15 | 0.18 |
| | ChatTime | TS | – | – | – | – | – | – | – | – | – | – |

Table 14: The *Average Offset* metric of different models on the **Base** subset's numerical split. $^{cot}$ denotes *thinking* mode. $^2$ denotes ChatTS without built-in statistical computation module. $-VL$ = Vision-Language. 'TS' stands for Time Series modality, as time-series-specific models introduce a TS encoder. '–' indicates that the model failed to respond correctly. **Bold underlined** values indicate the best performance within each category for each metric, and **bold** values indicate the second-best performance. Stat. and Non-Stat. columns represent the questions with staionary and non-stationary time series, respectively. **H** remark means the average offset value is higher than 1e5.

| Category | Model Name | Modality | Total | Trend | Seasonality | Basic | Stat. | Non-Stat. | Local | Overall | Uni-Var. | Multi-Var. |
|---|---|---|---|---|---|---|---|---|---|---|---|---|
| Open-source | DeepSeek-V3 | Text | **1.06** | **1.17** | 0.68 | **0.69** | **0.69** | 1.64 | **0.69** | **0.79** | **0.69** | 1.94 |
| | Kimi-K2 | Text | **1.24** | **1.80** | **0.57** | **1.14** | **1.14** | 1.68 | **0.49** | 1.27 | 1.14 | **1.59** |
| | Qwen3-32b$^{cot}$ | Text | 49.19 | 512.64 | 1.07 | 2.32 | 2.32 | 3.73 | 8.45 | 107.63 | 2.32 | 4.94 |
| | Qwen3-32b | Text | 4.57 | 34.94 | 0.75 | 1.43 | 1.43 | 1.48 | 2.72 | 8.34 | 1.43 | 3.17 |
| | Qwen3-8b$^{cot}$ | Text | 54.04 | 568.67 | 1.04 | 2.45 | 2.45 | 3.47 | 7.73 | 119.29 | 2.45 | 5.25 |
| | Qwen3-8b | Text | 6.01 | 39.83 | 0.81 | 3.05 | 3.05 | 1.76 | 4.96 | 10.64 | 3.05 | 4.15 |
| | Qwen2.5-32b | Text | 1.65 | 5.91 | 0.92 | 1.49 | 1.49 | **1.05** | 1.22 | 2.40 | 1.49 | 3.77 |
| | Qwen2.5-14b | Text | 16.56 | 170.23 | 0.90 | 2.06 | 2.06 | **1.12** | 1.33 | 36.76 | 2.06 | 5.37 |
| | Qwen2.5-7b | Text | 3.02 | 3.49 | 0.88 | 3.20 | 3.20 | 1.97 | 6.00 | 3.26 | 3.20 | 4.81 |
| | Qwen2.5-7b-VL | Text | H | 7.06 | 1.82 | 5.55 | 5.55 | H | 98.73 | 5.86 | 5.55 | 15.49 |
| | Qwen2.5-7b-VL | Vision | H | 13.24 | **0.61** | 31.51 | 31.51 | 1.32 | 1.17 | 28.08 | 31.51 | H |
| | Qwen2.5-7b-VL | V+T | H | 34.87 | 1.01 | H | H | H | 17.56 | H | H | 4.15 |
| Closed-source | Claude-3.7-Sonnet | Text | **0.48** | **1.24** | 0.46 | **0.32** | **0.32** | **0.39** | 0.62 | **0.51** | **0.32** | 0.50 |
| | Claude-Sonnet-4 | Text | **0.72** | **1.26** | 0.56 | 0.62 | 0.62 | 0.78 | **0.58** | 0.75 | 0.62 | **0.44** |
| | Gemini-2.5-Pro | Text | 1.13 | 9.62 | **0.27** | 0.36 | 0.36 | **0.30** | **0.35** | 2.21 | 0.36 | **0.27** |
| | Gemini-2.5-Flash | Text | 34.04 | 350.9 | 0.62 | 2.49 | 2.49 | 1.48 | 8.87 | 74.38 | 2.49 | 5.57 |
| | GPT-5-Minimal | Text | 0.91 | 1.85 | 0.69 | 0.70 | 0.70 | 0.97 | 0.83 | 0.94 | 0.70 | 0.67 |
| | GPT-4o | Text | 2.27 | 16.22 | 0.71 | 0.45 | 0.45 | 1.18 | 1.40 | 3.70 | 0.45 | 2.57 |
| | GPT-4o | Vision | 1.47 | 2.70 | 0.30 | 2.46 | 2.46 | 1.10 | 0.63 | 2.50 | 2.46 | 4.18 |
| | GPT-4o | V+T | **0.72** | 1.66 | **0.25** | **0.31** | **0.31** | 0.73 | 1.35 | **0.59** | **0.31** | 0.48 |
| | GPT-4o-mini | Text | 1.24 | 2.58 | 0.86 | 0.94 | 0.94 | 1.30 | 1.21 | 1.28 | 0.94 | 1.63 |
| Time-series | ChatTS | TS | 1.79 | 6.75 | 0.23 | 1.68 | 1.68 | 1.34 | 0.90 | 2.72 | 1.68 | 2.26 |
| | ChatTS$^2$ | TS | 21.89 | 130.25 | 0.24 | 20.54 | 20.54 | 8.88 | 0.88 | 43.18 | 20.54 | 7.21 |
| | ITFormer | TS | H | 7.08 | 0.94 | 10.89 | 10.89 | 1.59 | 0.79 | 10.10 | 10.89 | H |
| | ChatTime | TS | – | – | – | – | – | – | – | – | – | – |

Table 15: Performance of different models on the **InWild** subset. $^{cot}$ denotes *thinking* mode. [1] denotes models evaluated without any time-series input. [2] denotes ChatTS without built-in statistical computation module. $-VL$ = Vision-Language. 'TS' stands for Time Series modality, as time-series-specific models introduce a TS encoder. '−' indicates that the model failed to respond correctly. In reasoning tasks, abbreviations are: Ded. (Deductive), Ind. (Inductive), Analog. (Analogical), and Count. (Counterfactual). Best and second-best results within each category are underlined and **bolded**, respectively.

| Category | Model Name | Modality | Average | Feature Analysis | | | | | | Temporal Reasoning | | | | | |
|---|---|---|---|---|---|---|---|---|---|---|---|---|---|---|---|
| | | | | Acc. | Trend | Season | Noise | Volat. | Basic | Acc. | Ded. | Ind. | Causal | Analog. | Count. |
| Open-source | DeepSeek-V3 | Text | **0.61** | **0.67** | 0.73 | **0.57** | 0.79 | 0.55 | 0.75 | **0.59** | **0.60** | **0.63** | **0.52** | 0.58 | 0.57 |
| | Kimi-K2 | Text | 0.63 | 0.69 | 0.71 | 0.63 | **0.85** | 0.58 | 0.71 | 0.61 | 0.62 | 0.65 | 0.56 | 0.64 | 0.55 |
| | Qwen3-32b$^{cot}$ | Text | 0.58 | 0.65 | **0.72** | 0.54 | **0.81** | **0.58** | 0.65 | 0.55 | 0.52 | 0.60 | 0.49 | **0.61** | 0.48 |
| | Qwen3-32b | Text | 0.50 | 0.56 | 0.67 | 0.49 | 0.54 | 0.43 | 0.62 | 0.48 | 0.48 | 0.51 | 0.42 | 0.54 | 0.42 |
| | Qwen3-8b$^{cot}$ | Text | 0.50 | 0.57 | 0.54 | 0.51 | 0.74 | 0.54 | 0.55 | 0.47 | 0.41 | 0.52 | 0.43 | 0.49 | 0.48 |
| | Qwen3-8b | Text | 0.45 | 0.48 | 0.45 | 0.43 | 0.65 | 0.34 | 0.58 | 0.44 | 0.48 | 0.46 | 0.42 | 0.45 | 0.37 |
| | Qwen2.5-32b | Text | 0.53 | 0.62 | 0.66 | 0.53 | 0.77 | 0.52 | 0.62 | 0.49 | 0.47 | 0.53 | 0.51 | 0.50 | 0.44 |
| | Qwen2.5-14b | Text | 0.53 | 0.61 | 0.60 | 0.55 | 0.73 | 0.58 | 0.63 | 0.49 | 0.48 | 0.53 | 0.48 | 0.53 | 0.41 |
| | Qwen2.5-7b | Text | 0.44 | 0.45 | 0.61 | 0.48 | 0.35 | 0.35 | 0.42 | 0.44 | 0.41 | 0.47 | 0.43 | 0.45 | 0.44 |
| | Qwen2.5-7b-VL | Text | 0.37 | 0.37 | 0.41 | 0.32 | 0.44 | 0.28 | 0.38 | 0.37 | 0.38 | 0.35 | 0.40 | 0.35 | 0.36 |
| | Qwen2.5-7b-VL | Vision | 0.36 | 0.37 | 0.39 | 0.44 | 0.37 | 0.30 | 0.33 | 0.35 | 0.38 | 0.36 | 0.36 | 0.33 | 0.33 |
| | Qwen2.5-7b-VL | V+T | 0.39 | 0.41 | 0.48 | 0.41 | 0.45 | 0.30 | 0.40 | 0.38 | 0.39 | 0.37 | 0.43 | 0.36 | 0.35 |
| | Qwen2.5-32b$^{1}$ | Text | 0.34 | 0.34 | 0.40 | 0.40 | 0.24 | 0.33 | 0.31 | 0.34 | 0.35 | 0.35 | 0.38 | 0.31 | 0.30 |
| Closed-source | Claude-3.7-Sonnet | Text | 0.69 | 0.78 | 0.81 | 0.72 | 0.85 | 0.73 | 0.81 | 0.65 | 0.66 | 0.67 | 0.58 | 0.71 | 0.60 |
| | Claude-3.7-Sonnet | Vision | 0.69 | 0.75 | 0.82 | 0.67 | 0.82 | 0.70 | 0.75 | 0.66 | 0.64 | 0.72 | 0.62 | 0.71 | 0.60 |
| | Claude-3.7-Sonnet | V+T | **0.73** | 0.79 | **0.84** | **0.71** | 0.87 | 0.70 | 0.85 | **0.71** | 0.66 | **0.76** | 0.70 | 0.74 | 0.66 |
| | Claude-Sonnet-4 | Text | 0.71 | 0.79 | 0.78 | 0.67 | 0.91 | 0.70 | **0.89** | 0.68 | 0.67 | 0.75 | 0.60 | 0.69 | 0.63 |
| | Claude-Sonnet-4 | Vision | 0.67 | 0.74 | 0.84 | 0.59 | 0.87 | 0.63 | 0.79 | 0.64 | 0.59 | 0.75 | 0.55 | 0.66 | 0.60 |
| | Claude-Sonnet-4 | V+T | 0.71 | 0.78 | 0.81 | 0.64 | **0.90** | 0.72 | 0.86 | 0.68 | 0.65 | **0.76** | 0.62 | 0.72 | 0.66 |
| | Gemini-2.5-Pro | Text | 0.68 | 0.73 | 0.76 | 0.69 | 0.83 | 0.66 | 0.72 | 0.66 | 0.65 | **0.76** | 0.56 | 0.68 | 0.63 |
| | Gemini-2.5-Pro | Vision | 0.72 | **0.80** | **0.84** | 0.63 | 0.89 | 0.77 | 0.85 | 0.69 | **0.69** | **0.76** | 0.60 | 0.71 | 0.64 |
| | Gemini-2.5-Pro | V+T | 0.76 | 0.83 | 0.86 | **0.71** | 0.87 | **0.76** | 0.92 | 0.73 | 0.75 | 0.78 | 0.62 | 0.77 | 0.69 |
| | Gemini-2.5-Flash | Text | 0.50 | 0.50 | 0.53 | 0.40 | 0.59 | 0.45 | 0.55 | 0.51 | 0.47 | 0.57 | 0.45 | 0.56 | 0.45 |
| | GPT-5-High | Text | 0.72 | 0.76 | 0.82 | 0.67 | 0.87 | 0.60 | 0.85 | 0.70 | 0.69 | 0.72 | **0.66** | **0.76** | **0.68** |
| | GPT-5-Minimal | Text | 0.56 | 0.64 | 0.70 | 0.63 | 0.72 | 0.55 | 0.61 | 0.52 | 0.53 | 0.56 | 0.51 | 0.53 | 0.45 |
| | GPT-4.1 | Text | 0.58 | 0.64 | 0.63 | 0.57 | 0.76 | 0.58 | 0.68 | 0.56 | 0.58 | 0.61 | 0.52 | 0.52 | 0.55 |
| | GPT-4.1 | Vision | 0.58 | 0.61 | 0.70 | 0.46 | 0.70 | 0.51 | 0.67 | 0.57 | 0.57 | 0.61 | 0.57 | 0.55 | 0.56 |
| | GPT-4.1 | V+T | 0.63 | 0.67 | 0.73 | 0.55 | 0.85 | 0.60 | 0.68 | 0.62 | 0.62 | 0.69 | 0.59 | 0.59 | 0.56 |
| | GPT-4.1-Mini | Text | 0.59 | 0.66 | 0.69 | 0.49 | 0.88 | 0.55 | 0.73 | 0.56 | 0.56 | 0.60 | 0.54 | 0.56 | 0.49 |
| | GPT-4.1-Mini | Vision | 0.49 | 0.51 | 0.56 | 0.37 | 0.54 | 0.51 | 0.55 | 0.49 | 0.51 | 0.45 | 0.52 | 0.50 | 0.48 |
| | GPT-4.1-Mini | V+T | 0.58 | 0.64 | 0.68 | 0.54 | 0.75 | 0.53 | 0.73 | 0.56 | 0.59 | 0.61 | 0.55 | 0.54 | 0.44 |
| | GPT-4o | Text | 0.62 | 0.70 | 0.74 | 0.61 | 0.79 | 0.61 | 0.75 | 0.58 | 0.59 | 0.62 | 0.50 | 0.66 | 0.50 |
| | GPT-4o | Vision | 0.59 | 0.67 | 0.76 | 0.56 | 0.74 | 0.64 | 0.66 | 0.56 | 0.55 | 0.62 | 0.51 | 0.57 | 0.50 |
| | GPT-4o | V+T | 0.63 | 0.71 | 0.75 | 0.59 | 0.85 | 0.61 | 0.77 | 0.59 | 0.59 | 0.65 | 0.55 | 0.63 | 0.53 |
| Time-series | ChatTS | TS | 0.50 | 0.55 | 0.50 | 0.61 | 0.53 | 0.54 | 0.58 | 0.48 | 0.51 | 0.50 | 0.49 | 0.44 | 0.45 |
| | ChatTS$^{2}$ | TS | 0.48 | 0.51 | 0.53 | 0.55 | 0.48 | 0.52 | 0.50 | 0.48 | 0.50 | 0.50 | 0.52 | 0.42 | 0.43 |
| | ITFormer | TS | 0.33 | 0.37 | 0.29 | 0.31 | 0.37 | 0.29 | 0.36 | 0.36 | 0.37 | 0.40 | 0.35 | 0.35 | 0.35 |
| | ChatTime | TS | – | – | – | – | – | – | – | – | – | – | – | – | – |
| Human | Experts | – | 0.67 | 0.71 | 0.59 | 0.67 | 0.75 | 0.75 | 0.80 | 0.66 | 0.66 | 0.72 | 0.63 | 0.67 | 0.58 |

Table 16: Performance of different models on the **Match** subset. $^{cot}$ denotes *thinking* mode. [1] denotes models evaluated without any time-series input. [2] denotes ChatTS without built-in statistical computation module. $-VL$ = Vision-Language. 'TS' stands for Time Series modality, as time-series-specific models introduce a TS encoder. '–' indicates that the model failed to respond correctly. **Bold underlined** values indicate the best performance within each category for each metric, and **bold** values indicate the second-best performance.

| Category | Model Name | Modality | Average | Isomorphic | Robust | Localization | Reverse |
|---|---|---|---|---|---|---|---|
| Open-source | DeepSeek-V3 | Text | **0.65** | **0.90** | **0.79** | **0.57** | 0.35 |
| | Kimi-K2 | Text | 0.60 | 0.78 | 0.66 | 0.50 | 0.44 |
| | Qwen3-32b$^{cot}$ | Text | 0.60 | 0.80 | 0.64 | 0.42 | **0.52** |
| | Qwen3-32b | Text | 0.50 | 0.72 | 0.58 | 0.36 | 0.35 |
| | Qwen3-8b$^{cot}$ | Text | 0.50 | 0.62 | 0.56 | 0.36 | **0.46** |
| | Qwen3-8b | Text | 0.42 | 0.56 | 0.48 | 0.31 | 0.32 |
| | Qwen2.5-32b | Text | **0.62** | **0.84** | **0.72** | **0.53** | 0.41 |
| | Qwen2.5-14b | Text | 0.57 | 0.78 | 0.61 | 0.42 | 0.45 |
| | Qwen2.5-7b | Text | 0.40 | 0.45 | 0.49 | 0.35 | 0.29 |
| | Qwen2.5-7b-VL | Text | 0.30 | 0.31 | 0.36 | 0.27 | 0.28 |
| | Qwen2.5-7b-VL | Vision | 0.28 | 0.30 | 0.31 | 0.28 | 0.25 |
| | Qwen2.5-7b-VL | V+T | 0.34 | 0.40 | 0.41 | 0.27 | 0.30 |
| | Qwen2.5-32b[1] | Text | 0.25 | 0.25 | 0.25 | 0.25 | 0.25 |
| Closed-source | Claude-3.7-Sonnet | Text | 0.74 | 0.93 | **0.81** | **0.67** | 0.54 |
| | Claude-Sonnet-4 | Text | 0.71 | 0.85 | 0.79 | **0.61** | 0.60 |
| | Gemini-2.5-Pro | Text | **0.79** | **0.96** | 0.80 | 0.59 | **0.80** |
| | Gemini-2.5-Flash | Text | 0.44 | 0.63 | 0.57 | 0.28 | 0.26 |
| | GPT-5-High | Text | **0.81** | **0.98** | **0.81** | 0.60 | **0.86** |
| | GPT-5-Minimal | Text | 0.57 | 0.84 | 0.67 | 0.53 | 0.26 |
| | GPT-4.1 | Text | 0.67 | 0.89 | **0.82** | 0.55 | 0.40 |
| | GPT-4.1-Mini | Text | 0.63 | 0.90 | 0.78 | 0.44 | 0.40 |
| | GPT-4o | Text | 0.50 | 0.68 | 0.60 | 0.41 | 0.34 |
| | GPT-4o | Vision | 0.45 | 0.56 | 0.50 | 0.34 | 0.38 |
| | GPT-4o | V+T | 0.55 | 0.79 | 0.64 | 0.38 | 0.38 |
| Time-series | ChatTS | TS | 0.37 | 0.47 | 0.40 | 0.24 | 0.36 |
| | ChatTS[2] | TS | 0.32 | 0.46 | 0.41 | 0.22 | 0.20 |
| | ITFormer | TS | 0.24 | 0.16 | 0.25 | 0.25 | 0.29 |
| | ChatTime | TS | – | – | – | – | – |

Table 17: Performance of the ChatTS model on the **Match** subset across different time series length ranges. *Total* corresponds to the range [13,504], and "–" indicates that no questions fall into the given length range for that task.

| Length Range | Isomorphic | Robust | Localization | Reverse |
|---|---|---|---|---|
| Total | 0.47 | 0.40 | 0.24 | 0.36 |
| [64, 1024] | **0.53** | **0.57** | 0.23 | **0.44** |
| [256, 512] | – | – | **0.36** | – |

Table 18: Performance of different models on the **Align** subset. $^{cot}$ denotes *thinking* mode. [1] denotes models evaluated without any time-series input. [2] denotes ChatTS without built-in statistical computation module. $-\text{VL}$ = Vision-Language. 'TS' stands for Time Series modality, as time-series-specific models introduce a TS encoder. '–' indicates that the model failed to respond correctly. **Bold underlined** values indicate the best performance within each category for each metric, and **bold** values indicate the second-best performance.

| Category | Model Name | Modality | Average | TS→Sem | Sem→TS |
|---|---|---|---|---|---|
| Open-source | DeepSeek-V3 | Text | **0.94** | **0.95** | 0.94 |
| | Kimi-K2 | Text | **0.95** | **0.94** | **0.96** |
| | Qwen2.5-32b | Text | 0.93 | 0.92 | **0.94** |
| | Qwen2.5-14b | Text | 0.88 | 0.87 | 0.88 |
| | Qwen2.5-7b | Text | 0.69 | 0.68 | 0.71 |
| | Qwen3-32b$^{cot}$ | Text | 0.89 | 0.92 | 0.86 |
| | Qwen3-32b | Text | 0.87 | 0.86 | 0.88 |
| | Qwen3-8b$^{cot}$ | Text | 0.86 | 0.88 | 0.83 |
| | Qwen3-8b | Text | 0.79 | 0.74 | 0.84 |
| | Qwen2.5-7b-VL | Text | 0.64 | 0.68 | 0.59 |
| | Qwen2.5-7b-VL | Vision | 0.60 | 0.61 | 0.60 |
| | Qwen2.5-7b-VL | V+T | 0.73 | 0.78 | 0.67 |
| | Qwen2.5-32b[1] | Text | 0.27 | 0.29 | 0.26 |
| Closed-source | Claude-3.7-Sonnet | Text | 0.97 | 0.97 | **0.98** |
| | Claude-Sonnet-4 | Text | **0.98** | **0.98** | **0.99** |
| | Gemini-2.5-Pro | Text | 0.97 | 0.97 | **0.99** |
| | Gemini-2.5-Flash | Text | 0.94 | 0.94 | 0.95 |
| | GPT-5-High | Text | **0.99** | **0.99** | **0.99** |
| | GPT-5-Minimal | Text | 0.97 | 0.97 | **0.98** |
| | GPT-4o | Text | 0.96 | 0.96 | 0.97 |
| | GPT-4o-Mini | Text | 0.86 | 0.82 | 0.90 |
| Time-series | ChatTS | TS | 0.80 | 0.68 | 0.91 |
| | ChatTS[2] | TS | 0.45 | 0.49 | 0.42 |
| | ITFormer | TS | 0.29 | 0.32 | 0.26 |
| | ChatTime | TS | – | – | – |

Table 19: Token cost for a single evaluation on the InWild subset (1,084 samples).

| Model | Input/Output Tokens | Price Cost |
|---|---|---|
| Qwen2.5-32B | ≈6M / ≈20k | 0 |
| DeepSeek-V3 | ≈5M / ≈250k | ≈$1.62 |
| GPT-4o | ≈5M / ≈400k | ≈$16.50 |
| Claude-Sonnet-4 | ≈5M / ≈600k | ≈$24.00 |
| Gemini-2.5-Pro | ≈6M / ≈500k | ≈$12.50 |

# E   STATISTICAL ROBUSTNESS ANALYSIS

To assess the testing stability, robustness, and validity of MMTS-Bench, we conducted comprehensive statistical evaluations and bias analyses: **Bootstrap Confidence Interval**, **Iterative Subsampling Analysis**, and **Assessment of Dataset Artifacts**. These experiments evaluate the dataset's testing stability, the adequacy of its scale, and its resistance to spurious correlations, respectively.

## E.1   BOOTSTRAP CONFIDENCE INTERVAL

To evaluate the reliability of model performance and quantify its uncertainty, we adopted the non-parametric bootstrap method. Specifically, the experimental setup is as follows: Given a test set of size $D$, we generated $N = 1000$ bootstrap samples, also of size $D$, through sampling with replacement.

We selected a few representative models (covering close-source, open-source LLMs and TS-LLMs) for the MMTS-InWild subset, the choice split of the MMTS-Base subset, and the entire MMTS-Bench dataset. We then calculated their respective accuracy scores to obtain an empirical distribution for this metric. Based on this distribution, we report the mean accuracy, standard deviation (std), and the 95% confidence interval (CI).

Table 20: Bootstrap confidence interval of different models on the MMTS-InWild dataset.

| Models | Mean | Std | CI Low | CI High |
|---|---|---|---|---|
| Gemini-1.5-Pro (text) | 0.6827 | 0.0142 | 0.6541 | 0.7103 |
| Gemini-2.5-Pro (vision) | 0.7262 | 0.0138 | 0.6983 | 0.7537 |
| GPT-4o (text) | 0.6128 | 0.0148 | 0.5830 | 0.6421 |
| DeepSeekV3 (text) | 0.6218 | 0.0150 | 0.5932 | 0.6504 |
| Qwen2.5-32b (text) | 0.5379 | 0.0156 | 0.5083 | 0.5683 |

Table 21: Bootstrap confidence interval of different models on the MMTS-Base's choice split.

| Models | Mean | Std | CI Low | CI High |
|---|---|---|---|---|
| Gemini-1.5-Pro (text) | 0.6340 | 0.0205 | 0.5933 | 0.6726 |
| Claude-3.7-Sonnet (text) | 0.5967 | 0.0207 | 0.5581 | 0.6391 |
| GPT-4o (text) | 0.5609 | 0.0210 | 0.5211 | 0.6021 |
| DeepSeekV3 (text) | 0.5433 | 0.0205 | 0.5053 | 0.5845 |
| Qwen2.5-32b (text) | 0.4727 | 0.0218 | 0.4331 | 0.5158 |

Table 22: Bootstrap confidence interval of different models on the complete MMTS-Bench dataset.

| Models | Mean | Std | CI Low | CI High |
|---|---|---|---|---|
| Gemini-1.5-Pro (text) | 0.7235 | 0.0094 | 0.7059 | 0.7422 |
| GPT-4o (text) | 0.6059 | 0.0103 | 0.5864 | 0.6265 |
| DeepSeekV3 (text) | 0.6366 | 0.0100 | 0.6165 | 0.6562 |
| Qwen2.5-32b (text) | 0.5789 | 0.0104 | 0.5580 | 0.5986 |

The experimental results (Table 20, 21, 22 and Figure 10) indicate that the performance evaluations across all models exhibit low statistical dispersion, both on the full dataset and the subsets. Specifically, the bootstrapping experiment shows that the standard deviation (std) ranges only between $0.01 \sim 0.02$, and the width of the 95% CI remains within a narrow range (approximately $0.5 \sim 0.8$ percentage points).

These tight error bounds strongly confirm the statistical robustness of the MMTS-Benchmark. It demonstrates that the benchmark is insensitive to data sampling variance and can provide stable and reproducible evaluation results for cross-model and cross-capability comparisons. Furthermore, to further suppress the stochastic noise from model generations, we also introduced a mechanism

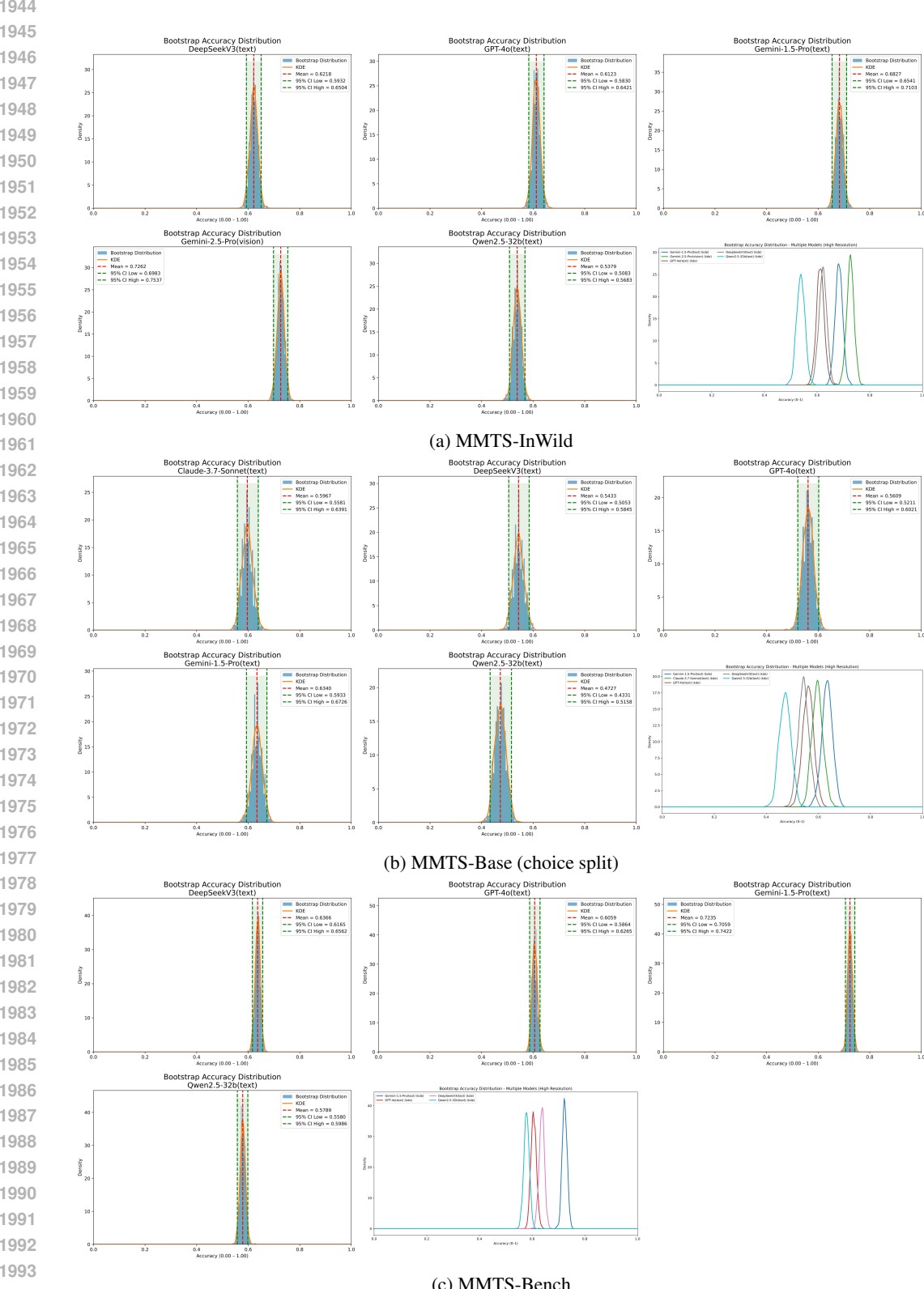

Figure 10: Visualization of the accuracy distribution and confidence intervals for different representative models on the MMTS-InWild, MMTS-Base (choice), and MMTS-Bench datasets.

of multiple sampling and majority voting during the evaluation, thereby establishing a more robust performance baseline.

### E.2 ITERATIVE SUBSAMPLING ANALYSIS

To investigate the relationship between evaluation stability and dataset scale, and to estimate the minimal sample size required to yield robust results, we conducted an Iterative Subsampling Analysis focusing on the representative model, `Gemini-2.5-Pro`, with text input.

The specific experimental setup is as follows: For a given dataset size $D$, we set the subsampling size $S$ as a variable that progressively increases from an initial value up to $D$, with an increment step of $T = 20$. At each fixed size $S$, we perform $N = 50$ independent repetitions of sampling, and calculate the mean, standard deviation (std), and coefficient of variation (CV) of the model's performance.

We use the coefficient of variation ($CV = \sigma/\mu$) as the core metric to measure evaluation stability. The dataset size $S$ is deemed to possess sufficient statistical stability when the CV curve, as $S$ increases, shows a descending trend and falls below a pre-set convergence threshold of $\tau = 0.02$. [8] This experiment covered the entire MMTS-Benchmark and its four subsets.

Table 23: Comparison of the minimum required sample size for stable assessment versus the actual sample size in the subsampling analysis experiment, along with the model's mean, standard deviation, and coefficient of variation under the actual sample size across four data subsets and the entire MMTS-Bench dataset.

| Dataset | Full Sample | Min Sample | Mean | Std | CV |
|---|---|---|---|---|---|
| Align | 240 | 60 | 0.9813 | 0.0100 | 0.0041 |
| Base (choice) | 568 | 400 | 0.6357 | 0.0203 | 0.0038 |
| Match | 400 | 260 | 0.7795 | 0.0199 | 0.0047 |
| InWild | 1084 | 600 | 0.6811 | 0.0130 | 0.0011 |
| Full | 2292 | 600 | 0.7199 | 0.0093 | 0.0010 |

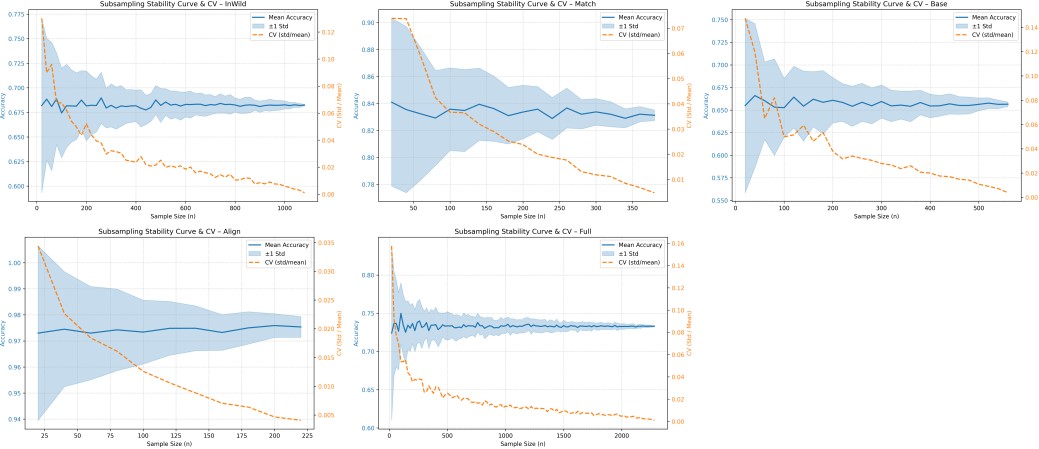

Figure 11: Trends of model accuracy metrics (mean, standard deviation, and coefficient of variation) with varying subsampling size on four subsets and the entire MMTS-Bench.

---

[8]We empirically set the convergence threshold to $\tau = 0.02$, which requires the standard deviation of the evaluation score to be controlled within 2% of the mean. For a typical model accuracy range ($50\% \sim 80\%$), this means the measurement error is limited to an absolute range of approximately $1\% \sim 1.6\%$. This strict stability constraint is crucial for suppressing "ranking flips" caused by sampling variance, ensuring the benchmark can reliably distinguish between models with slight performance differences.

The experimental results (Table 23 and Figure 11) demonstrate that the current data scale of MMTS-Bench provides an ample safety margin for evaluation stability. Specifically, the actual sample size of each subset significantly exceeds the minimum number of samples required to reach the convergence threshold (approximately $1.42 \sim 4$ times the required minimum), and the lowest CV has dropped to the $10^{-3}$ magnitude. This outcome confirms that we have not only ensured high confidence in the evaluation results at the current scale but have also achieved a good balance between statistical robustness and evaluation efficiency (computational and time costs).

### E.3  ASSESSMENT OF DATASET ARTIFACTS AND SHORTCUT LEARNING

A substantial body of research warns against benchmark performance driven by spurious correlations or explicit features rather than genuine reasoning (Geirhos et al., 2020; Gururangan et al., 2018). To ensure MMTS-Bench evaluates robust time-series reasoning rather than relying on dataset artifacts, we analyzed the dependency of model performance on explicit surface-level attributes.

Specifically, we examined the correlation between TSQA accuracy and three explicit factors: sequence length ($L$), variable count ($V$), and question text length ($T$) on the InWild subset. We evaluated three representative models: GPT-4o, Qwen2.5-32B, and ChatTS(Xie et al., 2024). We introduce three metrics to quantify these dependencies:

- **Correlation ($r_L, r_T$):** The Pearson correlation coefficient between accuracy and the logarithm of sequence length ($r_L$) or question text length ($r_T$). A value close to 0 indicates no linear dependency.
- **Length Sensitivity ($\Delta_{\text{long}}$):** The difference in mean accuracy between the samples in the longest quartile ($\geq$ 75th percentile) and the shortest quartile ($\leq$ 25th percentile).
- **Dimensionality Gap ($\Delta_{\text{dim}}$):** The difference in mean accuracy between multivariate and univariate samples.

As presented in Table 24, the results reveal minimal dependence on these artifacts. The correlation with sequence length is negligible across all models ($|r_L| < 0.08$), and the accuracy gap between extreme lengths ($\Delta_{\text{long}}$) remains within a narrow range (approx. $0.05 \sim 0.08$), showing no consistent bias towards short or long sequences. Similarly, the performance gap between univariate and multivariate series is marginal ($|\Delta_{\text{dim}}| < 0.04$), and question length shows only a weak effect ($|r_T| \approx 0.15$). These findings confirm that MMTS-Bench performance is not trivially predictable by simple metadata features.

Table 24: Analysis of potential dataset artifacts and shortcut learning on the InWild subset. Small absolute values for all metrics indicate that model performance is not dominated by simple features like length or dimensionality.

| Model Name | $r_L$ | $\Delta_{\text{long}}$ | $\Delta_{\text{dim}}$ | $r_T$ |
|---|---|---|---|---|
| GPT-4o | -0.0693 | 0.0824 | -0.0101 | -0.1451 |
| Qwen2.5-32B | 0.0547 | -0.0525 | 0.0353 | -0.0264 |
| ChatTS | 0.0727 | -0.0502 | -0.0163 | -0.0862 |

### E.4  CONCLUSION

We conducted three systematic analyses—Bootstrap Confidence Interval, Iterative Subsampling Analysis, and Assessment of Dataset Artifacts—which collectively demonstrate that MMTS-Bench possesses high statistical robustness, an efficient scale, and validity against shortcut learning. The results confirm that model performance on MMTS-Bench is driven by genuine time-series understanding rather than explicit surface-level features (e.g., sequence length or dimensionality), ensuring that the evaluation results are stable, reliable, and trustworthy.

## F    USE OF LLMS

During the preparation of this paper, we employed large language models (LLMs) to polish paragraphs and assist with grammar checking, aiming to reduce the gap with native English writing and to improve readability for reviewers and readers. LLMs were also used extensively in constructing the benchmark, with specific details already mentioned earlier.

