# OpenReview forum: "MMTS-Bench: A Comprehensive Benchmark for Multimodal Time Series Understanding and Reasoning"
_ICLR.cc/2026/Conference — Submitted to ICLR 2026_

### Official Review · Reviewer_Ay4A · 2025-10-22

**Soundness:** 3
**Presentation:** 4
**Contribution:** 3
**Rating:** 6
**Confidence:** 5

**Summary:**

In this paper, the authors proposed a new time series dataset for benchmarking LLM/MLLM's ability to understand and reason about time series input. 2424 TSQA pairs are generated across four subsets of tasks and five different domains. Human-in-the-loop curation is used together with LLM to generate the QA pairs for real-world time series data and further being validated by experts. Comprehensive experiments are conducted including both opensource and close source LLM models, showing that current SOTA LLMs still fall back on understanding and reasoning for time series for certain tasks.

**Strengths:**

S1: The authors proposed a new multi-modal, multi-dimensional benchmark covering different domains and tasks.
S2: Human-in-the-loop curation is used to ensure the dataset quality.
S3: Extensive experiments are conducted and show interesting results.

**Weaknesses:**

W1: The category of the datasets could be enhanced. For instance, the hardness of the question (easy or hard) and the length (100 points or 10000 points) and the number of the time series in the questions.
W2: Experiments could be improved. For instance, there is no token cost / latency experiments.

**Questions:**

Q1: The authors listed "controllable synthetic data pipeline" as one of the main contributions. However, this part is not significantly different from previous work, as the author wrote in the paper. Why do the authors treat it as one of the key contribution.
Q2: The authors concluded that TS-LLMs significantly lag behind general-purpose LLMs in cross-domain generalization. Are there any tasks TS-LLMs perform better? I guess they would perform better when the input time series contains very long context and multiple input time series. However, there are no such experiments.
Q3: The authors wrote "on InWild, humans achieve 67% accuracy", why do humans achieve lower accuracy than GPT-5, when they label the datasets?
Q4: It seems that all models perform worse in Base QA. Why is this the case? All models perform worse on synthetic datasets.
Q5: The description of a time series is generated by LLM. Then it is used to evaluate ability of LLM to judge alignment. Is alignment task evaluation valid?
Q6: The authors use prompt to evoke CoT. Did the author try thinking mode?

---

> ### Author Response · Authors · 2025-11-19
> **Part 1 of the rebuttal**
>
> Thank you for your positive and constructive feedback. We are encouraged that you recognized the central contributions of our work in (1) introducing a novel multi-modal, multi-dimensional time-series benchmark and (2) the comprehensive empirical evaluation revealing the current capabilities and limitations of LLMs on time-series reasoning. We also appreciate your thoughtful questions and suggestions, which have helped us identify key areas for clarification and future work. Below, we provide a detailed response to each of your points.
>
> # Response to Weaknesses
>
> ## W1：
>
> > The category of the datasets could be enhanced. For instance, the hardness of the question (easy or hard) and the length (100 points or 10000 points) and the number of the time series in the questions.
>
> We will incorporate your suggestion by augmenting the dataset with additional metadata such as sequence length and variable dimensionality. However, regarding the suggestion to assign explicit “difficulty levels” to questions, we cautiously believe that it is hard to define a consistent and objective metric for difficulty. The notion of difficulty as implicitly judged by a LLM based on its internal mechanisms can differ from human, intuition-based judgments; moreover, if we rely on a third-party model to define difficulty, we would merely transfer the bias to that model’s own strengths and weaknesses, thereby introducing new biases.
>
> Nevertheless, we performed an exploratory analysis to test whether our dataset could be difficulty-stratified from meta information such as sequence length, variable count, and text length. For each model \\(m\\) and each QA pair \\(i\\), we define a binary label \\(y_i^{(m)} = 1\\) if all sampled predictions of model \\(m\\) on QA pair \\(i\\) match the gold answer (i.e., that QA pair is “easy” for that model), and \\(y_i^{(m)} = 0\\) otherwise. As features, we use the sequence length \\(L_i\\), the variable count \\(d_i\\), and the text length \\(T_i\\), and form \\(\\mathbf{x}_i = (\\log(1 + L_i), \\log(1 + d_i), \\log(1 + T_i))\\). We then fit a per-model logistic regression
>
> \\[
> P(y_i^{(m)} = 1 \\mid \\mathbf{x}_i) = \\sigma\\big(\\beta_0 + \\beta_1 \\log(1 + L_i) + \\beta_2 \\log(1 + d_i) + \\beta_3 \\log(1 + T_i)\\big),
> \\]
>
> and evaluate it using ROC–AUC on the InWild subset. The resulting AUCs are 0.6099 for GPT-4o, 0.5682 for ChatTS (time-series adapted LLMs), and 0.4757 for Qwen 2.5 32b. According to established guidelines [1], an AUC of 0.5 corresponds to random guessing, and values between 0.5 and 0.7 are typically interpreted as providing only poor to weak discrimination. These results indicate that, on our dataset, sequence length, variable count, and text length provide at best limited predictive signal for difficulty classification. In other words, relying solely on such low-level metadata does not yield stable or reliable difficulty stratification; the effective difficulty of our questions is largely driven by higher-level semantic and reasoning properties instead. Overall, we believe that the definition and calibration of difficulty levels on such datasets still require further in-depth investigation.
>
> ## W2：
>
> > Experiments could be improved. For instance, there is no token cost / latency experiments.
>
> We thank the reviewer for pointing out the absence of token cost / latency experiments and fully agree that resource overhead is an important dimension in benchmark design. To address this concern, we additionally measured inference cost on the InWild subset: for Qwen 2.5 32B, DeepSeek V3, GPT-4o, Claude Sonnet 4, and Gemini 2.5 Pro, we recorded all input/output tokens on the InWild samples and converted them into total monetary cost using the official API prices of each model. As shown in Table 1, different models use about 5M–6M input tokens and 20k–600k output tokens, and the cost of a single full evaluation run over all 1,084 InWild samples is below \$25 for all models.
>
> **Table 1: Token cost for a single evaluation on the InWild subset (1,084 samples).**
>
> | Model           | Input/Output Tokens | Price Cost |
> |-----------------|---------------------|------------|
> | Qwen 2.5 32B    | ≈6M/≈20k            | 0          |
> | DeepSeek V3     | ≈5M/≈250k           | ≈$1.62     |
> | GPT-4o          | ≈5M/≈400k           | ≈$16.50    |
> | Claude Sonnet 4 | ≈5M/≈600k           | ≈$24.00    |
> | Gemini 2.5 Pro  | ≈6M/≈500k           | ≈$12.50    |
>
> Regarding latency, commercial LLM APIs differ substantially in their server implementation, and network conditions, making direct comparison of end-to-end wall-clock time difficult to conduct in a fair and reproducible way.
>
> ---
>
> [1] Hosmer Jr, David W., Stanley Lemeshow, and Rodney X. Sturdivant. Applied logistic regression. John Wiley & Sons, 2013.

---

> ### Author Response · Authors · 2025-11-20
> **Part 2 of the rebuttal**
>
> # Response to Questions
>
> ## Q1:
>
> > The authors listed "controllable synthetic data pipeline" as one of the main contributions. However, this part is not significantly different from previous work, as the author wrote in the paper. Why do the authors treat it as one of the key contribution.
>
> To synthesize rich time-series data in a controllable manner and accurately annotate the corresponding attributes, many recent works have adopted composable and modular time-series generation pipelines. Prior work such as **TimeSeriesExam** [1] and **ChatTS** [2] also follow modular composition–based synthesis. Our data generation pipeline also builds upon this modular method while improving diversity, attribute-recording precision, and the alignment precision between synthesized time series and their corresponding textual contents across different capability dimensions.
>
> **Generation Diversity.** TimeSeriesExam adopts a “base patterns + composition” strategy. However, its base patterns are relatively simple, being restricted to only three categories (non-periodic, periodic, and random processes), each with a limited set of sub-options (e.g., linear, exponential, sinusoids, square waves). This results in a constrained diversity in the final generated time series. In contrast, inspired by STL [3]  decomposition, we decompose the base pattern into Trend, Seasonal, and Noise modules, each equipped with rich, quantifiable control parameters.
>
> - Trend is generated via a second-order random walk, allowing richer trend dynamics through parameters such as $\delta^2 \tau_{\text{trend}}$ and initial value.
>
> - Seasonal includes multiple waveform patterns (such as sinusoids, square waves, and simulated traffic flow waveforms), testing the model's ability to perceive repeated periodic structures.
>
> - Noise includes four carefully designed types (Gaussian white noise, quantized noise, random-walk noise, etc.), providing more realistic and diverse noise characteristics.
>
> ChatTS, on the other hand, samples from an Attribute Pool with a fixed combination of four components (Trend, Periodicity, Local Fluctuation, Noise) using a fixed “Multiple” combination. Compared to our richer combination strategies (Additive / Concatenated / Additive without Noise), this may limit the diversity of the synthesized time series.
>
> **Attribute-Recording Precision.** During synthesis, we recorded detailed routing information and all quantitative parameters involved in generation, including the precise configuration of each base pattern and the applied composition methods. This is significantly more granular than the mostly qualitative attribute annotations used in TimeSeriesExam, and it enables our QA-generation stage to leverage exact numerical values and attributes when constructing questions.
>
> **Time Series–Text Alignment Accuracy.** Given the diversity and granularity of our synthesized series, enabling LLMs to generate factually aligned and high-precision QA pairs is a challenging task. Thanks to our proposed capability-dimension framework and precise attribute logging, we can feed rich auxiliary information into QA generation and tightly couple it with specific capability dimensions. This leads to significantly improved alignment between the time series and the textual QA pairs.
> Moreover, the fine-grained attribute records allow us to construct more challenging and discriminative questions. For example, instead of the binary “presence or absence of seasonality” questions seen in TimeSeriesExam and ChatTS, we can leverage the numerical seasonal_strength parameter to differentiate between weak, medium, and strong seasonality, resulting in more nuanced and informative evaluation items.
>
> ---
>
> [1] Cai, Yifu, et al. "Timeseriesexam: A time series understanding exam." arXiv preprint arXiv:2410.14752 (2024).
>
> [2] Xie, Zhe, et al. "Chatts: Aligning time series with llms via synthetic data for enhanced understanding and reasoning." arXiv preprint arXiv:2412.03104 (2024).
>
> [3] Cleveland, Robert B., et al. "STL: A seasonal-trend decomposition." J. off. Stat 6.1 (1990): 3-73.

---

> ### Author Response · Authors · 2025-11-20
> **Part 3 of the rebuttal**
>
> ## Q2:
>
> > The authors concluded that TS-LLMs significantly lag behind general-purpose LLMs in cross-domain generalization. Are there any tasks TS-LLMs perform better? I guess they would perform better when the input time series contains very long context and multiple input time series. However, there are no such experiments.
>
> Indeed, our unified evaluation across different MMTS-Bench subsets reveals that although current TS-LLMs are specifically designed for time series, they still lag significantly behind the strongest general-purpose LLMs in various TSQA settings. However, on certain tasks (e.g., Base and InWild), the leading TS-LLM (e.g., ChatTS) can match or even surpass some open-source LLMs of comparable scale, demonstrating its effectiveness as a specialized model.
>
> Regarding the multiple-input time series experiment you mentioned, both the InWild and Base subsets of MMTS-Bench contain a rich set of multivariate questions. We summarize model performance on multivariate questions as follows:
>
> ---
>
> **Table 1**:  Accuracy evaluation of representative models on the **InWild** subset. 'Average' represents the overall average accuracy, and 'MultiVar' represents the accuracy on multivariate questions.
>
> | **Models**        | **Average** | **MultiVar** |
> | ----------------- | ----------- | ------------ |
> | DeepSeek-V3       | 0.61        | 0.60         |
> | Qwen2.5-14b       | 0.53        | 0.56         |
> | Qwen2.5-7b        | 0.44        | 0.47         |
> | Claude-4-Sonnet   | 0.71        | 0.70         |
> | Gemini-2.5-Pro    | 0.68        | 0.61         |
> | GPT-4o         | 0.62        | 0.60         |
> | ChatTS            | 0.50        | 0.49         |
> | ITFormer          | 0.33        | 0.36         |
>
> ---
>
> **Table 2**: The performance evaluation of representative models on the numerical split of the **Base** subset.
>
> | **Models**        | **Accuracy@10% ↑** | **Relative Accuracy ↑** | **Offset ↓** |
> | ----------------- | ------------------ | ----------------------- | ------------ |
> | DeepSeek-V3       | 0.56               | 0.76                    | 1.94         |
> | Qwen2.5-14b       | 0.34               | 0.55                    | 5.37         |
> | Qwen2.5-7b        | 0.31               | 0.50                    | 4.81         |
> | Claude-4-Sonnet   | 0.66               | 0.80                    | 0.44         |
> | Gemini-2.5-Pro    | 0.72               | 0.83                    | 0.27         |
> | GPT-4o         | 0.51               | 0.67                    | 2.57         |
> | ChatTS            | 0.40               | 0.56                    | 2.26         |
> | ITFormer          | 0.00               | 0.18                    | High         |
>
> ---
>
> From these results, we can see that the accuracies of different models on the whole InWild dataset and its multivariate questions are highly consistent, with overall variation within **3%** (except Gemini-Pro, which fluctuates up to **7%**). Existing TS-LLMs do not outperform the strongest general LLMs on typical multivariate questions; however, on **numerical split** in the Base subset, ChatTS shows clear improvements over base models of the same or even larger scale.
>
> Regarding very long contexts and very large numbers of input time series, we would like to clarify the design considerations behind MMTS-Bench. During dataset construction, we indeed attempted to build a “multivariate long-sequence” subset by synthesizing ~300 time series of length 10k and constructing QA pairs for them. However, preliminary evaluations showed that the average input length per sample reached tens of thousands of tokens, which far exceeds the context limits of many models. This introduced two major issues:   **(1)Severe unfairness**—different models varied in whether they could fully ingest the input, making the evaluation reflect context-window limits rather than the actual reasoning ability we aimed to measure. **(2)Extremely high inference cost**—very long contexts caused inference time and computation cost to grow sharply, making large-scale repeated experiments and community reproduction infeasible.
>
> In addition, the current leading TS-LLM model ChatTS explicitly states that “The length should be between 16 and 1024, with at most 15 time series.” Given these practical constraints, we ultimately restricted the dataset to sequence lengths 13–2048 and 1–4 variables. Compared with existing benchmarks, our length range is already the most extensive, while still containing a sufficient number of multivariate questions, which is enough to support multi-dimensional capability evaluation of current TS-LLMs.
>
> In summary, current TS-LLMs still have substantial room for improvement across multiple task dimensions. We hope that MMTS-Bench, as a balanced, interpretable, and high-quality dataset, can serve as an effective benchmark for evaluating models across diverse capability dimensions, and further advance the development of TS-LLMs.

---

> ### Author Response · Authors · 2025-11-20
> **Part 4 of the rebuttal**
>
> ## Q3:
> > The authors wrote "on InWild, humans achieve 67% accuracy", why do humans achieve lower accuracy than GPT-5, when they label the datasets?
>
> We would like to clarify a potential source of confusion: the reported 67% “human accuracy” refers to the human participants in our evaluation, not to the annotators who created the dataset labels; these two groups are disjoint.
>
> ## Q4:
>
> > It seems that all models perform worse in Base QA. Why is this the case? All models perform worse on synthetic datasets.
>
> The primary objective of the Base subset is to provide a highly controlled and simplified analytical environment for decoupling and analyzing specific types of time-series reasoning abilities. Since each sequence in Base is generated by a known mechanism, it enables the isolated evaluation of fine-grained capabilities. For instance, as shown in Table 3 and Section 4.2, most models exhibit particular difficulty with local and seasonality-related subtasks within Base, revealing limitations in their analytical precision along these dimensions.
>
> Regarding the lower absolute performance on Base, we attribute it to two main factors:
>
> - **Increased intrinsic difficulty due to high-precision controlled tasks.** Base includes tasks designed under strictly controlled conditions. These tasks require more precise numerical computation and accurate sequence-segment localization (see Appendix Tables 12–14). Such designs have lower tolerance for approximation errors relative to the other data subsets, thereby amplifying the current limitations of large models in fine-grained quantitative analysis.
>
> - **Absence of latent domain priors.** Base is constructed using synthetic data and therefore does not contain the latent domain characteristics present in real-world data. This design choice shifts the focus purely towards structural awareness and feature analysis of time series. In contrast, the other three subsets (InWild, Match, Align) are derived from real-world domains, whose distributions may more closely align with the data encountered during large-scale pre-training. This alignment potentially grants models an implicit "prior advantage" on those subsets.
>
> For these reasons, we consider the model’s lower absolute accuracy on Base to be aligned with the design expectations.
>
> ## Q5:
>
> > The description of a time series is generated by LLM. Then it is used to evaluate ability of LLM to judge alignment. Is alignment task evaluation valid?
>
> We fully understand the reviewer's concern regarding the potential for circular dependency when using LLM-generated content for evaluation. To address this, our Align benchmark's evaluation is grounded in objective human expert judgment—not the style or bias of any particular model—through a rigorous **Human-in-the-Loop Curation** process (Section 3.4 and Appendix A.4.2). Specifically, the LLM is utilized solely to generate initial drafts to ensure linguistic fluency; however, every description has been verified and refined by domain experts to guarantee that it objectively and accurately reflects the true patterns of the time series. Furthermore, our **distractor design (Section 3.3)** uses incorrect options that are challenging to distinguish, including time series from the same domain and those from different domains but with carefully matched value ranges. This forces the model to perform fine-grained feature recognition rather than relying on simple numerical matching or superficial patterns. In summary, **human verification combined with carefully designed distractors** ensures the validity of the evaluation, aiming to test the model's genuine cross-modal understanding capabilities.
>
> ## Q6:
>
> > The authors use prompt to evoke CoT. Did the author try thinking mode?
>
> We thank the reviewer for the question about thinking mode. In addition to using CoT prompts in our main evaluation, we also evaluate the LLMs' thinking modes. Specifically, for Qwen3-8B/32B we compare thinking mode against non-thinking mode, and for GPT-5 we compare Minimal Thinking and High Thinking configurations. The corresponding results are reported in Fig. 6(b) and Tables 11–16 and 18. For example, on the InWild dataset, the average accuracy of Qwen3-32B improves from **0.50** in non-thinking mode to **0.58** in thinking mode, and GPT-5 improves from **0.56** under Minimal Thinking to **0.72** under High Thinking.
> It is worth noting that enabling thinking mode for LLMs substantially increases the number of output tokens and the overall inference cost. Therefore, in our overall evaluation framework we adopt a unified CoT-prompting strategy as the default setting for all models, in order to ensure comparability of computational cost across models, and we report thinking-mode results only for a representative subset of models (Qwen3 and GPT-5) as complementary ablation studies.
> Overall, these results strongly support our conclusion that explicit CoT reasoning is more effective than merely scaling up model size.

---

> > ### Comment · Reviewer_Ay4A · 2025-11-26
> >
> > Thanks for the rebuttal, which addressed most of my concerns. I would like to keep my score, which is already positive.

---

### Official Review · Reviewer_w7Fz · 2025-10-23

**Soundness:** 2
**Presentation:** 2
**Contribution:** 2
**Rating:** 2
**Confidence:** 4

**Summary:**

This paper introduces a new dataset for multi-modal time-series and text understanding, addressing an important gap in existing datasets by combining temporal data with natural language question-answering tasks. The dataset is organized into several categories, with a combination of both synthetic and real datasets

However, the dataset has quality and methodological concerns. The Base category relies heavily on synthetic data generation methods that closely follow existing work. More importantly, the InWild and Align categories depend on LLM-generated descriptions. With this in mind, examination of the provided anonymous repo reveals that descriptions for multi-variate datasets often fail to capture their multi-dimensional nature, hence contradicting the paper's claims about handling such complexity. Furthermore, the Align task descriptions are also overly specific and distinguishable, hence explaining the near-perfect model performance and suggesting that the task may be too easy.

**Strengths:**

- A dataset focusing on multimodal time-series and text is highly valuable, as significant gaps still exist in current datasets addressing this area.

**Weaknesses:**

- **Base Category:** This category of the dataset relies on synthetic time-series data generated using preset trends, noise, and seasonality components to synthesize QnAs. The process closely follows the approach found in [Ref] (see Fig. 4 in [Ref]). Additionally, for this dataset category, the authors mention 17 expert-designed templates to create diverse, well-structured QnAs. However, there is no reference or detailed description of these templates provided anywhere in the paper.

[Ref]: Zhe Xie, Zeyan Li, Xiao He, Longlong Xu, Xidao Wen, Tieying Zhang, Jianjun Chen, Rui Shi, and Dan Pei. 2025. *ChatTS: Aligning Time Series with LLMs via Synthetic Data for Enhanced Understanding and Reasoning.* Proc. VLDB Endow. 18, 8 (April 2025), 2385–2398. https://doi.org/10.14778/3742728.3742735

- **InWild Category:** This category instead considers realistic datasets, which is important for capturing real-world behavior. The QnAs in this category rely on LLMs to generate descriptions. However, since LLMs are not yet strong at understanding time-series data (hence the motivation for multimodal models), the accuracy of these QnAs depends heavily on expert validation. It would be helpful to discuss what types of issues experts identified during validation and how frequently these occurred.

- **Quality of the dataset:** The InWild and Align categories depend heavily on LLM-generated descriptions. However, closer inspection of the anonymous GitHub repository reveals several issues.
  - First, for multi-variate datasets (e.g., the climate dataset in Appendix A.2), the descriptions fail to capture the multi-variate aspect that the paper claims to emphasize. For example:
    > "This series demonstrates irregular cyclical behavior with multiple peaks and valleys, showing significant amplitude variations across different time intervals."

    Such a description does not reflect the multi-variate nature of the data, raising questions about how well the dataset captures this complexity as claimed by the authors on multi-dimensionality.
  - Similarly, in the Align section (e.g., `ts2caption_qa_120.csv`), many descriptions focus on very specific events, such as:
    > "This time series shows an overall upward trend from around 55 to 72, with notable volatility and several significant peaks and valleys throughout the sequence."

    These descriptions make the Align task relatively easy, as the differences between descriptions are clear and unambiguous. This could explain the near-perfect model performance on this task. This suggests potential quality issues in the Align category of the dataset.

- **Hierarchy categorization:** The authors describe their dataset as having a hierarchical taxonomy of time-series tasks with an orthogonal structure, spanning from basic perception to advanced reasoning (page 2). However, it is unclear why these are considered orthogonal. For instance, structural awareness (the Base category) seems conceptually related to Temporal Reasoning (the InWild category). Thus, the rationale for this hierarchical and orthogonal categorization is not well justified.

**Minor comments:**

- Typo: “a innovative” → “an innovative.”
- Figure 3: The arrow is tilted and should be fixed.
- Presentation: The paper relies heavily on the reader to navigate appendices. It would improve readability to include explicit references to the relevant appendices throughout the main text.

**Questions:**

Please refer to the Weaknesses section

---

> ### Author Response · Authors · 2025-11-17
> **Part 1 of the rebuttal**
>
> Thank you very much for your careful reading of our submission paper and thoughtful comments.
>
> We truly appreciate your detailed observations and constructive feedback regarding the Base, InWild, Align subset, the overall dataset quality, and our hierarchical task taxonomy. These comments are extremely valuable and help us further clarify the motivation and design considerations of MMTS-Bench.
>
> We provide our point-by-point responses to **W1–W4** below.
>
> ---
>
> ## W1: Expert-Designed Templates of Base Subset
>
> In the current version of the paper, these 17 expert-designed templates are only briefly mentioned, which may make it difficult for readers to fully understand their design and usage.
>
> To address this and improve clarity and reproducibility, we have updated the repository to make these templates and their implementation details fully available, including:
>
> - The complete content of all 17 templates in `template.json`;
>
> - A `README.md` file in the same directory describing the template categories (trend/seasonality/noise/basic) and their roles;
>
> - A generation script `generate_question.py`, which can be used to reproduce all synthetic QA pairs in the Base subset.
>
> You can directly inspect these files at the following link:
> https://anonymous.4open.science/r/MMTS-BENCH-BEF7/Generate_QA/Base/templates/
>
> Due to space limitations, we do not enumerate all templates in the main paper. In the revised version, we will explicitly refer to this repository to further improve clarity and reproducibility.
>
> ---
>
> ## W2: Expert Validation in Constructing InWild Subset
> When constructing MMTS-InWild, we applied an expert validation protocol to all generated QA pairs, mainly targeting three types of issues:
>
> **(1) data–description misalignment**, such as incorrect descriptions of trend, seasonality, or volatility, and misplaced temporal indices;
>
> **(2) question–answer correctness**, including ambiguously worded questions, answers contradicting the time-series patterns, or cases where there is no unique correct answer;
>
> **(3) answer leakage**, such as question wording that hints at the correct option or candidate options that are clearly unbalanced.
> For each subset, approximately 20–30% of the generated QA pairs were flagged by experts and subsequently revised or discarded, with temporal indices misalignment being the most frequent error type.

---

> ### Author Response · Authors · 2025-11-17
> **Part 2 of the rebuttal**
>
> ## W3: Quality of the dataset
>
> ### (1) On how multi-variate complexity is captured
>
> The two examples the reviewer notes both come from our Align subset. The goal of Align is to evaluate **semantic alignment between time series and textual descriptions** (“time series ↔ semantics”), so many descriptions are intentionally focused on the overall pattern of a single time series (e.g., trend, seasonality, volatility) and do not involve complex analysis across multiple time series.
>
> The QA instances that directly correspond to the “multi-dimensional / multi-variate complexity” you are concerned about **mainly appear in the InWild subset**, especially in the climate and cloudops domains. InWild contains complex questions that require the model to perform feature analysis and temporal reasoning over multiple variables, and this kind of multi-variate complexity mainly manifests in **two forms**:
>
> > For your convenience, all of the examples mentioned below can be directly located in the file `tsqa_1084.csv`  in our repository (https://anonymous.4open.science/r/MMTS-BENCH-BEF7/Benchmark/InWild/tsqa_1084.csv), by using the num column to find the corresponding question (e.g., num=45).
>
> - **Analysis and reasoning over relationships between multiple variables**
>
>   In this type of question, the model must simultaneously observe two or more time series and perform analytical reasoning about their relationships, contrasts, or hypothetical dependencies, rather than describing or analyzing a single series in isolation. In the climate domain, we have typical examples such as:
>
>   - Compare the statistical and randomness characteristics of multiple variables to draw a conclusion (num=45):
>    > "...by analyzing the temporal characteristics of two variables, Time Series 1 and Time Series 2, which variable exhibits stronger randomness?"
>   - Compute and compare the noise characteristics of different variables (num=102):
>   > “When examining the time series data between indices 68 to 148, which of the following statements best represents an analogical inference about the observed noise patterns in Time Series 1 and Time Series 2?”
>   - Infer whether different time series exhibit opposite trends over the same time period (num=387):
>   > “...can we reasonably infer that when Time Series 1 reaches a peak, the lower Time Series 2 tends to exhibit a downward trend?”
>   - Analyze the patterns of multiple time series and perform analogical reasoning (num=860):
>   > “...determine whether the statement is true or false: The data pattern of Time Series 1 is close to a system with rapid state transitions, while the data pattern of Time Series 3 is close to a system with gradual changes.”
>   - Reason about potential causal or correlational relationships between different series within the same interval (num=882):
>   > “...determine whether the following statement is true or false: The sharp decline in the values of Time Series 1 between index 48 and 62 may be the potential cause of the corresponding drop in the values of Time Series 2 during the same period.”
>
>   In the cloudops domain, we have typical examples such as:
>
>   - Perform counterfactual reasoning based on the original dependency between variables (num=83):
>   > “Suppose constraints were set so that Time Series 1 would likely not exceed 5% during the entire monitoring period. Which of the following statements best reflects the expected Time Series 2 values for most of the time series?”
>   - Conduct inductive reasoning over multiple variables, distinguishing them and identifying their changing patterns (num=776):
>   > “...which of the following inductive conclusions about the relationship between Time Series 1 and Time Series 2 is best supported by the observed patterns?”
>   - Analyze the relationship between the volatility patterns of different variables (num=967):
>   > “...which conclusion is most accurate regarding the relationship between Time Series 1 and Time Series 2 patterns when analyzing their volatility characteristics?”
>   - Capture the relationship in raw multivariate data and predict outcomes under hypothetical changes (num=981):
>   > “...if Time Series 1 does not show the original trend within the range of index 175–250, then which of the following statements can best describe the possible relationship between Time Series 1 and Time Series 2?”
>   - Reasoning about system behavior and interpreting hypothetical scenarios via bivariate pattern analysis (num=1020):
>   > “An optimization pattern is defined as follows: when Time Series 2 drops by at least 10 percentage points from its stable state (approximately 99%), an efficient system should reallocate resources to increase Time Series 1. Based on the provided time series data, which logical conclusion can be deduced about the system's resource optimization behavior?”

---

> ### Author Response · Authors · 2025-11-17
> **Part 3 of the rebuttal**
>
> - **Analysis and reasoning about a single variable under multi-variate input**
>
>   Another type of question focuses on a single target variable, but this time series is embedded in a multi-variate dataset, and the model must correctly identify the target variable and carry out analysis and reasoning according to the question requirement. For example, in the climate domain we ask the model to:
>
>   - Compare local fluctuation patterns of the same variable across different intervals (num=887):
>   > “Observing the three provided time series datasets, determine whether the following statement is true or false: The fluctuation pattern of Time Series 1 between index 45 and 65 is completely different from the fluctuation pattern between index 80 and 100.”
>   - Conduct local causal reasoning about a particular segment of the given series (num=894):
>   > “Examining the noise patterns in Time Series 3 between indices 450–500, what is the most likely causal explanation for the sustained negative values in this segment?”
>   - Evaluate basic trend and perform deductive reasoning for a specific series under a multi-variate setting (num=917):
>   > “Based on the trend patterns observed throughout the entire time period of Time Series 1, if the value of Time Series 1 falls below 90 again in the future, is it possible for it to continue declining to below 75?”
>
> ### (2) On the “simplicity” of the Align subset and the near-perfect scores
>
> We appreciate the reviewer's concern that the specific descriptions in the Align subset might reduce task complexity. We clarify this from two perspectives:  the intended design objectives of Align and the empirical results across different models.
>
> - **Design perspective**. Align is intentionally designed as an alignment calibration subtask. Its primary goal is to assess whether a model can accurately match time series with natural language descriptions under relatively low linguistic ambiguity, rather than to deliberately introduce vagueness or difficulty in the text itself. For this reason, the descriptions are intentionally specific about aspects such as trend ranges and magnitude changes. This should not be interpreted as a lack of dataset quality, but as a reflection of the particular role that Align plays in the overall benchmark.
> As described in Appendix A.2 and A.3, the original time series in Align come from three real-world domains: traffic (value ranging [0, 1]), cloudops, and climate (value ranging [0, 100]). To prevent models from making decisions solely based on domain or value range, we implemented two design strategies: (1) cloudops and climate are scaled to similar ranges, which reduces obvious cross-domain magnitude differences; and (2) in each QA instance with four options, we always include at least one distractor that comes from the same domain as the correct answer and has a similar statistical distribution. This forces models to rely on fine-grained patterns and numeric details.
>
> - **Empirical perspective**. The results in Table 18 show that Align retains strong discriminative power across models: most closed-source models are typically around 0.97–0.99, many open-source models fall in the 0.7–0.9 range, and specialized time-series models or smaller models often score below 0.8. This indicates that, despite its relatively simple form, Align still provides a meaningful ranking and diagnostic signal for alignment ability.
> From a benchmark design perspective, we argue that it is valuable to include multiple subtasks encompassing a range of difficulty levels. The Align subset serves for alignment calibration and upper-bound checking, whereas InWild, particularly its reasoning-oriented questions, provides a more challenging evaluation.

---

> ### Author Response · Authors · 2025-11-17
> **Part 4 of the rebuttal**
>
> ## W4: Clarifying Orthogonality and Hierarchy
>
> In this paper, the term "orthogonal" denotes functional distinctness rather than statistical independence. We decompose time-series understanding into several information dimensions, each representing a unique and non-substitutable processing requirement. Since a single task typically relies on only a subset of these dimensions, we organize and annotate our tasks accordingly to prevent mixed or redundant definitions.
> The notion of hierarchy means that higher-level capabilities are the result of combining multiple dimensions, rather than introducing entirely new dimensions.  Although the dimensions in the Base and InWild subsets are related, we deliberately assign them distinct roles. The Base subset is designed to isolate and "activate" individual dimensions (e.g., structural awareness) in controlled settings. In contrast, InWild tasks, drawn from realistic scenarios, necessitate the simultaneous integration of multiple dimensions. By explicitly annotating these dimensional dependencies, we enable fine-grained failure analysis: poor performance on an InWild task can be traced to deficiencies in specific dimensions (e.g., structural awareness), rather than being attributed to an undifferentiated notion of overall failure.
>
> ## Minor Revisions
>
> We sincerely thank the reviewer for the careful attention to detail; in the revised version we will correct the noted typo, fix the arrow in Figure 3, add clearer pointers from the main text to the relevant appendices, and further tighten our checks to ensure that the overall presentation is smooth and rigorous.

---

> ### Author Response · Authors · 2025-11-26
> **Follow-up on our responses**
>
> Dear Reviewers,
>
> We would like to kindly follow up on our submitted rebuttal. We have carefully addressed your comments through detailed point-by-point responses.
>
> We are eager to engage in further discussion to clarify any remaining uncertainties. We would greatly appreciate your feedback on whether our responses have resolved your concerns.
>
> Thank you for your time and insightful reviews.
>
> Best regards, Authors

---

> > ### Comment · Reviewer_w7Fz · 2025-11-28
> > **Response to Rebuttal**
> >
> > I want to thank the authors for the rebuttal. Nevertheless, my concerns regarding the dataset quality remain. Specifically:
> >
> > **Differences from prior work:** After examining the added templates used in Base category, it is difficult to see how this dataset materially differs from prior work that follows similar approaches such as [Ref] Zhe Xie, Zeyan Li, Xiao He, Longlong Xu, Xidao Wen, Tieying Zhang, Jianjun Chen, Rui Shi, and Dan Pei. 2025. ChatTS: Aligning Time Series with LLMs via Synthetic Data for Enhanced Understanding and Reasoning. Proc. VLDB Endow. 18, 8 (April 2025), 2385–2398. https://doi.org/10.14778/3742728.3742735
> >
> > **Dataset quality:** Given the above, the contributions' burden lies on the other parts of the dataset besides the Base category. However, as mentioned earlier, concerns about their quality remain unresolved. For example, regarding the align category, the issue with focusing on such generic forms of descriptions is that the resulting QnA fail to meaningfully challenge models in terms of time-series understanding. For instance, consider ts_id 6633 from the dataset:
> >
> > A) This highly volatile time series shows periodic spikes reaching above 30, with baseline values typically ranging from 1 to 15, exhibiting cyclical patterns of sharp increases followed by gradual declines.
> >
> > B) This volatile time series fluctuates between approximately 55 and 97, showing irregular oscillations with several notable peaks and valleys, displaying high variability throughout the entire sequence without clear trends.
> >
> > C) This time series exhibits a U-shaped pattern, starting around 0.8, declining to minimum values near 0.4 around index 120, then steadily rising to peak values above 1.4 towards the end.
> >
> > D) This time series maintains relatively stable values around 60-65 for most of the sequence, then shows a gradual decline to the 57-59 range, ending with a dramatic drop to approximately 30.
> >
> >
> > As shown above, the distinctions between the options are straightforward and unambiguous. This makes it very easy for models to answer correctly, which as I previously mentioned, could explain the near-perfect performance reported on this task. This raises concerns regarding the validity of the Align QnAs subcategory as a challenging evaluation dimension for LLM understanding. Finally, regarding the multivariate component, I thank the authors for the clarification; however, the paper gives the impression of stronger claims about multivariate coverage than what is reflected in the rebuttal, specifically that multivariate data is only included within certain subsets of the inwild category.

---

> ### Author Response · Authors · 2025-11-28
> **Second Rebuttal – Part I**
>
> We thank the reviewer for the continued feedback. We appreciate the opportunity to further clarify the technical distinctions of our work and the rigorousness of the dataset construction. Below, we address the remaining concerns regarding the **novelty of the Base category**, **the validity of the Align task**, and **the scope of multivariate coverage**.
>
> **1. Differences from Prior Work**
>
> The reviewer notes that the Base category appears similar to ChatTS (Xie et al., 2025) due to the shared use of modular synthesis. We clarify that while both works are inspired by the broad idea of controllable synthesis, they differ substantially in **generation mechanisms, parameterization, and evaluation depth**. These distinctions, which may not be fully visible from the JSON templates alone, are summarized below:
>
> - **Generation Mechanics (STL-inspired decomposition vs. Fixed compositional pipeline):** ChatTS relies on sampling from a predefined attribute pool with a relatively fixed structure (Trend + Periodicity + Fluctuation + Noise). In contrast, MMTS-Bench adopts a flexible **STL-inspired decomposition** strategy supporting multiple composition modes (Additive, Concatenated, and Additive-without-Noise). Crucially, our trend component is generated via a **second-order random walk** controlled by variance parameters and initial values—distinct from the simple linear/exponential formulations in earlier benchmarks. We also incorporate unique noise types (e.g., quantized and random-walk noise) not present in ChatTS. These mechanisms enable temporal behaviors far beyond fixed functional combinations.
>
> - **Evaluation Granularity (Quantitative graded attributes vs. Qualitative labels):** ChatTS primarily annotates series using qualitative, binary properties (e.g., presence/absence of seasonality). MMTS-Bench, however, logs the **exact numerical parameters** used during synthesis (e.g., seasonal_strength). This allows us to construct graded questions that distinguish between **weak, medium, and strong** signal intensities. This supports fine-grained sensitivity testing—evaluating perceptibility thresholds relevant to LLM reasoning—which is impossible with qualitative labels alone.
>
> - **Structural Diversity (Non-stationarity):** Our "Concatenated" strategy specifically stitches together segments with differing trends or seasonal properties, creating sequences where statistical characteristics shift over time. This explicitly supports tasks evaluating **stationarity versus non-stationarity**, a structural dimension not explicitly targeted by the fixed combination logic of ChatTS.
>
> - **Unified Contribution:** It is precisely these advancements in generation mechanisms and parameter granularity that enable the richer task design in the Base dataset. Our contribution lies in the **co-evolution** of an improved generative framework and the diagnostically meaningful dataset it produces. The method and dataset are an integrated whole: the refined framework makes nuanced tasks possible, and the tasks highlight why such methodological refinement is necessary.
>
> **2. Dataset Quality and Validity of the Align Category**
>
> Regarding the distinct options in ts_id 6633, we respectfully suggest distinguishing between **"task difficulty"** and **"dataset quality."**
>
> - **Quality vs. Difficulty:** "Low quality" typically implies factual errors, misalignment, or ambiguity. The examples cited (A vs. B vs. C vs. D) are indeed distinct, but they are **factually accurate and unambiguous**. This is a deliberate design choice. As stated in the paper, Align is explicitly designed as an **"alignment calibration"** subtask. Its purpose is to verify whether a model possesses the fundamental **Cross-Modal Understanding** capability to map statistical features (e.g., "U-shaped," "periodic spikes") to natural language without the interference of linguistic ambiguity.
>
> - **Validity as a Discriminator:** While the task appears simple for SOTA closed-source models (e.g., GPT-4o achieves 96%), empirical results prove it is **not** a solved problem for all LLMs. There is a significant performance gap that validates its use for calibration:
>
>     - **SOTA Models** (e.g., GPT-4o) demonstrate strong Cross-Modal Understanding.
>
>     - **Smaller/Open Models struggle significantly.** For instance, Qwen2.5-7B achieves only **69%** accuracy, and Qwen2.5-7B-VL (Vision) scores roughly **60-73%**. This **~30%** performance gap demonstrates that Align effectively differentiates models based on foundational alignment capabilities. Failures here signal a deficit in processing time-series tokens, explaining subsequent failures in complex reasoning (InWild).
>
> - **Design Mechanism:** To ensure robustness, we implemented **Controlled Sampling**: for every question, at least one distractor is drawn from the **same domain** with a **similar statistical distribution**. This forces the model to perform actual feature verification rather than relying on domain shortcuts.

---

> ### Author Response · Authors · 2025-11-28
> **Second Rebuttal – Part II**
>
> **3. Multivariate Coverage Claims**
>
> We apologize if our initial presentation created ambiguity regarding the extent of multivariate coverage. We do not claim universal coverage; rather, multivariate data is strategically deployed. We will **explicitly clarify** the exact scope in the final manuscript:
> **Base Subset:** Multivariate synthetic series are specifically generated to test the **"Univariate-Multivariate"** subtask (distinguishing statistical properties across variables).
>
> **InWild Subset:** Multivariate analysis is a core component of our **Feature Analysis** and **Temporal Reasoning** dimensions. As illustrated in the Sankey diagram (Figure 7 in Appendix A.3), "Multivariate" tasks flow specifically into "Temporal Reasoning" and "Feature Analysis."
>
> We will refine the manuscript to explicitly delineate this scope, ensuring that the descriptions of multivariate coverage are precise and strictly aligned with the dataset composition.

---

### Official Review · Reviewer_cxqH · 2025-10-30

**Soundness:** 3
**Presentation:** 3
**Contribution:** 3
**Rating:** 6
**Confidence:** 4

**Summary:**

This paper presents MMTS-Bench, a comprehensive and multi-dimensional benchmark designed to evaluate the capabilities of Large Language Models (LLMs) in understanding and reasoning over time series data. The proposed benchmark comprises 2,424 question-answer pairs, systematically organized into four subsets (Base, InWild, Match, and Align). These subsets are designed to assess five core task dimensions: structural awareness, feature analysis, temporal reasoning, sequence matching, and cross-modal understanding. The data is sourced from both synthetically generated series and real-world domains (e.g.,transport, climate, economics, healthcare). The authors  further  conduct an extensive evaluation of a wide range of open-source, closed-source, and time-series-specialized LLMs.

**Strengths:**

1. The proposed five-dimensional taxonomy (structural awareness, feature analysis, temporal reasoning, sequence matching, and cross-modal understanding) provides a systematic and hierarchical framework for evaluating different aspects of reasoning over time-series data.
2. Another strength is the use of real-world time series from five diverse domains: Transport, Cloud Operations, Climate, Economics, and Healthcare. This diversity is crucial for rigorously assessing model generalization and robustness across different data distributions and real-world scenarios.
3. The evaluation of models across different categories (closed-source, open-source, TS-LLMs) and modalities (text, vision, vision+text) provides a holistic view of the landscape.

**Weaknesses:**

1. Since all tasks are multiple-choice, the evaluation mainly reflects recognition accuracy rather than deeper reasoning or explanatory ability. It would be helpful if future versions could include a small number of open-ended or explanation-based questions to capture richer reasoning behaviors.
2. Some tasks, particularly in the *InWild* and *Align* subsets, appear to rely on textual descriptions derived from the same data as the QA pairs.  It might be worth clarifying how potential textual cues are controlled to avoid trivial answer hints or information leakage.
3. Current metrics (mostly multiple-choice accuracy) capture correctness but not reasoning depth or interpretability. Additional analyses—such as reasoning trace evaluation or error categorization—would better support the claim of “understanding and reasoning.”
4. The paper includes a CoT ablation showing a 10–20% improvement, which is encouraging. It might be helpful if some qualitative examples or error analyses (e.g., distinguishing logical vs. perceptual errors) were added, as these could provide deeper insight into how models reason through the tasks.

**Questions:**

1. Can MMTS-Bench be extended for generative reasoning tasks (open-ended answers) rather than multiple-choice only?

2. How do you ensure that QA pairs in InWild and Align subsets do not leak cues from textual descriptions that trivially hint at the answer?

3. Section 3.4 mentions human verification, but it might be useful to provide a quantitative measure of annotation reliability—such as agreement rates or inter-annotator consistency—to strengthen confidence in the dataset’s quality.

---

> ### Author Response · Authors · 2025-11-19
> **Part 1 of the rebuttal**
>
> Thank you for the thoughtful assessment and constructive feedback. We are encouraged by your positive comments on our benchmark's taxonomy and evaluation breadth. Our point-by-point responses are detailed below.
>
> ## 1.Task Design and Evaluation Metrics
>
> > W1: Since all tasks are multiple-choice, the evaluation mainly reflects recognition accuracy rather than deeper reasoning or explanatory ability. It would be helpful if future versions could include a small number of open-ended or explanation-based questions to capture richer reasoning behaviors.
>
> > W3: Current metrics (mostly multiple-choice accuracy) capture correctness but not reasoning depth or interpretability. Additional analyses—such as reasoning trace evaluation or error categorization—would better support the claim of “understanding and reasoning.”
>
> > Q1: Can MMTS-Bench be extended for generative reasoning tasks (open-ended answers) rather than multiple-choice only?
>
> We thank the reviewer for their thoughtful feedback on task design and evaluation metrics. In the current version, all tasks are designed as multiple-choice questions with a single correct answer and automatic verification. This design is primarily motivated by the need for objectivity, reproducibility, and fairness in large-scale benchmarking. This commitment to deterministic evaluation is a principle shared by many mainstream LLM benchmarks: for instance, MMLU and MMMU adopt multiple-choice formats with unambiguous answers; GSM8K and AIME-style math benchmarks, although open-ended in form, ultimately require unique numerical or symbolic answers and are evaluated via exact matching; and code benchmarks such as SWE-bench and HumanEval rely on pre-constructed test suites for automated verification.
>
> Although MMTS-Bench adopts a multiple-choice format, our hierarchical taxonomy ensures that models must perform time-series reasoning—such as analyzing trends and seasonality—before arriving at the correct answer. Therefore, multiple-choice accuracy remains a meaningful indicator of a model’s temporal understanding and reasoning ability.
>
> We fully acknowledge the value of open-ended reasoning tasks and the evaluation of reasoning traces. However, we would like to highlight the trade-offs involved: reliably evaluating free-form natural language explanations at the scale of MMTS-Bench typically requires either a specially trained reward model or an automated grader based on strong baseline models, which could itself introduce new sources of bias. For these reasons, we have prioritized deterministic multiple-choice accuracy as the core metric in this version.
>
> It is worth emphasizing that MMTS-Bench is naturally extensible to generative reasoning scenarios. Each sample retains rich structured metadata, allowing the same data and problem settings to be reformatted into open-ended responses or "explain-your-reasoning" tasks. This also paves the way for future work on fine-grained error analysis and reasoning trace evaluation, with the aid of a dedicated reward model. We regard the systematic integration of such tasks and reasoning-path evaluation as a key direction for future versions.

---

> ### Author Response · Authors · 2025-11-19
> **Part 2 of the rebuttal**
>
> ## 2. Measures Against Textual Bias and Annotation Reliability Assessment
>
> > W2: Some tasks, particularly in the InWild and Align subsets, appear to rely on textual descriptions derived from the same data as the QA pairs. It might be worth clarifying how potential textual cues are controlled to avoid trivial answer hints or information leakage.
>
> > Q2: How do you ensure that QA pairs in InWild and Align subsets do not leak cues from textual descriptions that trivially hint at the answer?
>
> > Q3: Section 3.4 mentions human verification, but it might be useful to provide a quantitative measure of annotation reliability—such as agreement rates or inter-annotator consistency—to strengthen confidence in the dataset’s quality.
>
> To address the concerns regarding potential textual cues and information leakage, we implemented a rigorous, two-step strategy encompassing both proactive filtering and post-hoc evaluation.
>
> First, during the QA construction phase for the InWild and Align subsets, we proactively controlled for textual bias through a two-step filtering process:
>
> - We removed any background sentences and domain-specific phrases that were not essential for answering the question.
>
> - We either discarded or rewrote any QA items where the question text or options alone could trivially reveal the answer without requiring analysis of the time-series data (e.g., by directly echoing the descriptive text).
>
> These measures ensure that solving the retained questions necessitates genuine analysis and reasoning over the temporal patterns, rather than relying on superficial textual hints.
>
> Second, to quantitatively validate the effectiveness of our filtering process, we conducted an ablation experiment. We evaluated an open-source model (Qwen 2.5 32B) in a no-input setting, where it only received the question and options without any time-series data. The model's accuracy across subsets was consistently close to **random guessing** (Base: 0.30 vs. 0.27; InWild: 0.34 vs. 0.38; Match: 0.25 vs. 0.25; Align: 0.27 vs. 0.25; all values are accuracy, with the second number being the random-guess baseline). More importantly, these results are markedly lower than the accuracies achieved when the model has access to the time-series inputs (see Appendix D). This provides strong empirical evidence that our dataset construction successfully mitigates trivial answer hints and information leakage.
>
> Separately, to address the query on human verification in Section 3.4, we rigorously evaluated annotation reliability using \\(Fleiss' \\kappa\\) [1]. Our annotation team consisted of a fixed group of 10 time-series domain experts. The overall \\(\\kappa\\) score of **0.73** was achieved, which falls into the **"Substantial Agreement"** category (0.61–0.80) according to Landis and Koch [2]. To ensure the validity of the final benchmark, the ground truth labels were determined by majority vote, and any ambiguous cases with low agreement underwent a second round of adjudication. This high level of agreement provides strong statistical evidence for both the reliability and validity of the dataset.
>
> ----
>
> [1] Fleiss, Joseph L. "Measuring nominal scale agreement among many raters." Psychological bulletin 76.5 (1971): 378.
>
> [2] Landis, J. Richard, and Gary G. Koch. "The measurement of observer agreement for categorical data." biometrics (1977): 159-174.

---

> ### Author Response · Authors · 2025-11-19
> **Part 3 of the rebuttal**
>
> ## 3. Analysis of COT performance enhancement
>
> > The paper includes a CoT ablation showing a 10–20% improvement, which is encouraging. It might be helpful if some qualitative examples or error analyses (e.g., distinguishing logical vs. perceptual errors) were added, as these could provide deeper insight into how models reason through the tasks.
>
> We thank the Reviewer for the constructive feedback. To further investigate the source of the performance gain from Chain-of-Thought (CoT), we conducted a systematic error analysis on both correct and incorrect predictions. Following the Reviewer's suggestion, we analyzed the mechanisms through which CoT helps mitigate both perceptual and logical errors.
>
>  - **Mitigating Perceptual Errors from Oversight**: When processing long time-series, the baseline model is prone to "attention drift," causing it to miss or mislocalize critical data points (e.g., an anomalous spike within a specified weekend). CoT addresses this by enforcing an explicit index-time mapping and a point-wise scanning over the given intervals. This shifts the model's approach from a vague, "browsing-style" processing to a more precise, "localization-style" inspection, thereby reducing such perceptual omissions.
>
>  - **Correcting Logical Errors from Prior Bias**: Many errors stem not from data ambiguity, but from the model's over-reliance on generic priors over the current evidence (e.g., assuming "peaks always occur in the morning" while ignoring actual "afternoon peaks" in the data). CoT compels the model to perform explicit, computational reasoning—such as precisely calculating time indices and comparing peak values across windows. This grounds the model's reasoning in the raw data and curbs unwarranted extrapolation from prior assumptions.
>
> In summary, our findings indicate that the benefits of CoT extend beyond “increased reasoning depth.” More importantly, CoT enhances the model’s ability to extract and utilize evidential information within complex long-context scenarios: it reduces both perceptual oversight of critical locations and unwarranted reliance on generic priors.

---

> > ### Comment · Reviewer_cxqH · 2025-11-25
> >
> > Thanks for the detailed response. It was helpful and resolved my initial confusion.

---

### Official Review · Reviewer_VshB · 2025-10-31

**Soundness:** 3
**Presentation:** 3
**Contribution:** 2
**Rating:** 4
**Confidence:** 4

**Summary:**

This paper introduces MMTS-Bench, a multimodal benchmark for evaluating Large Language Models (LLMs) on time series understanding and reasoning tasks. The benchmark comprises 2,424 question-answering (QA) pairs across four subsets (Base, InWild, Match, and Align), covering five orthogonal task dimensions: structural awareness, feature analysis, temporal reasoning, sequence matching, and cross-modal understanding. The authors propose a hierarchical task taxonomy and employ a three-stage framework for real-world QA generation combined with synthetic data construction. They conduct extensive evaluations on closed-source, open-source, and time-series-adapted LLMs, revealing that general-purpose LLMs outperform TS-LLMs in cross-domain generalization, and that CoT reasoning and multimodal fusion significantly improve performance.

While the work addresses an important problem in time series LLM evaluation, several concerns limit its contribution: the distinction from prior benchmarks is not sufficiently clear, the dataset size is relatively small compared to existing work, and the benchmark lacks coverage of high-dimensional, long-sequence industrial time series that are common in real-world applications.

**Strengths:**

1. **Hierarchical Task Taxonomy**: The multi-dimensional task classification framework with five orthogonal dimensions is well-structured and provides a systematic way to organize time series understanding tasks. The theoretical combination of 286 fine-grained task types demonstrates thoughtful design.
2. **Comprehensive Evaluation**: The authors evaluate a wide range of models (closed-source, open-source, and TS-LLMs) across different modalities (text-only, vision-only, and multimodal), providing valuable insights into current model capabilities and limitations.
3. **Quality Control**: The human-in-the-loop curation process with expert verification helps ensure dataset quality, as demonstrated by the experiments showing that expert review effectively prevents question errors and answer leakage.
4. **Practical Insights**: The finding that CoT reasoning provides greater improvements than parameter scaling, and that multimodal fusion enhances performance, offers actionable guidance for future development.
5. **Clear Methodology**: The three-stage progressive framework for real-world QA generation and the modular synthetic data construction pipeline are well-described and reproducible.

---

**Weaknesses:**

1. **Limited Distinction from Prior Work**: The paper does not clearly articulate how MMTS-Bench significantly differs from or improves upon existing benchmarks (ChatTime-TSQA, Time-MQA, ChatTS, EngineMT-QA, etc.). While Table 6 provides a horizontal comparison, it lacks a clear summary highlighting what makes MMTS-Bench uniquely valuable. The authors should provide a more explicit comparison table or section that directly contrasts:

   - Dataset sizes and scales
   - Task coverage and taxonomy depth
   - Evaluation dimensions
   - Construction methodologies
   - Unique contributions of MMTS-Bench
2. **Limited Dataset Scale**: With 2,424 QA pairs, MMTS-Bench is relatively small compared to existing datasets:

   - ChatTime-TSQA: 48K pairs
   - Time-MQA-TSQA: 200K pairs (1.4K for testing)
   - EngineMT-QA: 11K pairs

   This limited scale may constrain the benchmark's ability to evaluate model performance comprehensively, especially for fine-grained capability assessments. The authors should discuss whether this size is sufficient for robust evaluation or provide justification for the chosen scale.
3. **Insufficient Coverage of High-Dimensional, Long-Sequence Industrial Time Series**: Real-world industrial time series often exhibit:

   - High dimensionality (tens to hundreds of variables)
   - Long sequences (thousands to millions of time points)
   - Complex temporal dependencies across multiple scales

   However, MMTS-Bench primarily covers:

   - Low-dimensional cases: Most datasets have 1-2 variables (only Climate has 4 variables)
   - Moderate sequence lengths: Maximum length of 1,728 points (CloudOps), with most datasets having sequences of 100-700 points

   The benchmark's current design may not adequately assess LLMs' capabilities on the types of industrial-scale time series that practitioners encounter. The authors acknowledge extending to longer sequences in future work, but this limitation significantly affects the benchmark's applicability to industrial settings.
4. **Incomplete Comparison Analysis**: While Table 6 provides some comparison, it lacks quantitative metrics comparing:

   - Task diversity and granularity
   - Domain coverage
   - Difficulty distribution
   - Evaluation rigor (e.g., statistical robustness measures)
5. **Limited Discussion on Dataset Selection**: The paper does not sufficiently justify why certain domains were selected over others, or how the chosen domains represent the diversity of real-world time series applications. The absence of certain critical domains (e.g., manufacturing, finance with high-frequency data, IoT sensor networks) limits generalizability.
6. **Missing Evaluation of Scalability**: The benchmark does not systematically evaluate how model performance scales with sequence length or dimensionality, which would be valuable for understanding practical applicability.

**Questions:**

* **Comparative Analysis**: Could the authors provide a more detailed comparison table or section that explicitly highlights:
  - What specific capabilities MMTS-Bench evaluates that prior benchmarks do not?
  - How the hierarchical taxonomy differs from the "flat" taxonomies in prior work?
  - What quantitative advantages MMTS-Bench offers (e.g., task coverage, evaluation rigor)?
* **Dataset Scale Justification**:
  - What methodology was used to determine that 2,424 pairs is sufficient for comprehensive evaluation?
  - Have the authors conducted power analyses or similar statistical assessments?
  - What is the minimum dataset size required for reliable evaluation across different subtasks?
* **Industrial Applicability**:
  - What is the plan for extending the benchmark to high-dimensional (e.g., 50+ variables) and long-sequence (e.g., 10K+ points) cases?
  - Can the authors provide preliminary experiments or analysis on how current models perform on such challenging cases?
  - Would it be feasible to add an additional subset specifically targeting industrial-scale time series?
* **Task Taxonomy Validation**:
  - How was the five-dimensional orthogonal structure validated? Are there empirical studies demonstrating that these dimensions are indeed orthogonal?
  - What is the coverage of the 286 theoretical task combinations in the actual dataset?
* **Cross-Dataset Generalization**:
  - Have the authors evaluated whether models trained on MMTS-Bench generalize to other time series datasets?
  - What is the benchmark's utility for model development vs. evaluation?

---

> ### Author Response · Authors · 2025-11-17
> **Part 1 of the rebuttal**
>
> We sincerely thank you for the constructive feedback and insightful comments and appreciate the time and effort the you devoted to evaluating our work.
>
> Based on the comments, we carefully provide detailed responses below. We aim to address your questions clearly and thoroughly, and hope our clarifications help improve your understanding of our contributions.
>
> ## 1. Detailed Comparison with Previous Work
> During literature review, we conducted a thorough comparative analysis of related work. Due to space constraints, only the most critical attributes were presented in our paper. In response to your concerns, we have further expanded the comparison across different works, which can be found at: https://anonymous.4open.science/r/MMTS-BENCH-BEF7/comparison.md. This full table covers aspects that you specifically referred to, such as dataset sizes and scales, task coverage and taxonomy depth, etc. Via comparison, our unique contributions can be summarized as:
>
> **Hierarchical Multi-level Taxonomy:**
> As highlighted in the "Taxonomy Details" column, existing TSQA datasets often lack a hierarchical structure, which may hinder comprehensive and interpretable evaluation. In our work, we decompose TS understanding into 5 orthogonal core dimensions: structural awareness, feature analysis, temporal reasoning, sequence matching, and cross-modal understanding, with their relevant subtasks. This design represents the most detailed hierarchical multi-level taxonomy among existing works, facilitating balanced and multi-level evaluation.
>
> **Diverse Data Formats:**
> Guided by our proposed hierarchical taxonomy, we construct specialized data subsets emphasizing different capability dimensions, as detailed in Table 1 in the paper. As shown in the "Generation Details" column of the comparison table, we adopt dedicated synthesis methods for each subset, in contrast to the largely fixed and static approaches of previous work. This effectively increases data diversity and improves quality. Consequently, our benchmark is more fine-grained, with strong interpretability and high data quality.
>
> **High-Quality, with Balanced Attributes:**
> In terms of multiple dataset attributes, MMTS-Bench represents the most balanced, high-quality benchmark currently available. Regarding TS diversity, we include both real-world and synthetic data, covering multiple domains, TS lengths ranging from 13 to 2048 and covers 1–4 variables. In terms of human-verified data size, we ranks first with 2524 samples. For question types, we support True/False, binary/multiple-choice, and numerical questions. Our evaluation metrics include Accuracy, Accuracy@N%, Offset, and Relative Accuracy. Our taxonomy and multi-level capability design is the most hierarchical and systematic, ensuring balanced difficulty. Finally, our task-specific synthesis methods yield a dataset that is diverse, balanced, interpretable, and robust.
>
>
> ## 2. Quantitative Experiments on Statistical Robustness
> To quantitatively assess the statistical robustness of MMTS-Bench, we conducted two commonly used statistical experiments: **bootstrap confidence interval analysis** and **subsampling stability curve analysis**.
>
> The goal of the Bootstrap Confidence Interval Analysis is to examine the testing robustness of the benchmark. Specifically, given a dataset of size D, we perform sampling with replacement for D model predictions and compute the mean accuracy of the sampled set. Repeating this process N = 1000 times yields an empirical distribution of each model's mean accuracies on this dataset, from which we estimate the mean, standard deviation (std), and 95% confidence interval (CI).
>
> The goal of the Subsampling Stability Curve Analysis is to examine whether the scale (number of samples) of the benchmark is appropriate. Given a dataset of size D, we repeatedly sample S instances, conduct N = 50 trials, and compute the mean, standard deviation, and coefficient of variation (CV = std / mean). The sampling size S increases with step size T = 20. The CV value reflects the magnitude of accuracy fluctuations during the evaluation process.
>
> Our experimental results indicate that across both subsets and the full dataset, and across all evaluated models, the bootstrap std and CI widths remain consistently low (std ≈ 0.01–0.02; CI width ≈ 0.5–0.8). This demonstrates the strong statistical robustness of MMTS-Bench, showing that it supports stable evaluation across models and capability dimensions. The results also show that for each subset of MMTS-Bench, the actual dataset size is 1.42–4× larger than the minimum required for stable evaluation, and the minimum CV can reach the 0.001 range. This indicates the current benchmark scale provides stable evaluation while maintaining a good balance between robustness and evaluation cost (runtime). Due to length constraints, for detailed results: https://anonymous.4open.science/r/MMTS-BENCH-BEF7/Extra_experiment_results/robustness_experiments.pdf

---

> ### Author Response · Authors · 2025-11-17
> **Part 2 of the rebuttal**
>
> ## 3. Evaluation of Scalability
>
> A large body of research on shortcut learning and dataset artifacts has shown that when model performance on a benchmark can be predominantly predicted by a single, shallow, and explicit feature, models often “get the right answer for the wrong reason,” relying on spurious correlations rather than the reasoning or semantic understanding abilities that a benchmark is intended to assess [1,2,3].
>
> MMTS-Bench is fundamentally designed for time-series question answering (TSQA), with the goal of evaluating model understanding and reasoning across multiple dimensions. Therefore, in designing both the data and tasks, we intentionally avoid making TSQA performance trivially predictable from single explicit attributes such as sequence length or variable count; otherwise, MMTS-Bench would risk degenerating into a length/scale stress test instead of a genuine TS reasoning benchmark.
>
> In response to your suggestion, we conducted a systematic analysis of how model performance varies with sequence length and variable count on the representative real-world InWild subset (1084 TSQA samples, sequence lengths ranging from approximately 24 to 1728 with 1–4 variables). Specifically, for 3 representative models, GPT-4o, Qwen2.5-32B-Instruct, and ChatTS, we computed sample-level correlations between TSQA accuracy and (1) sequence length, (2) variable count, and (3) question text length, and compared accuracy differences between short/long sequences and uni-/multivariate samples. For clarity, \\(r_L\\) and \\(r_T\\) denote Pearson correlations between log sequence length / log question text length and accuracy (small \\(|r|\\) indicates weak dependence); \\(\\Delta_{\\text{long}}\\) denotes the difference in mean accuracy between the shortest 25% and longest 25% sequences (small \\(|\\Delta_{\\text{long}}|\\) indicates no substantial degradation on long sequences); and \\(\\Delta_{\\text{dim}}\\) denotes the mean accuracy difference between multivariate and univariate time series (small \\(|\\Delta_{\\text{dim}}|\\) indicates limited impact of variable count). Full formal definitions and implementation details of these metrics will be included in the appendix of the revised submission. The main findings are as follows:
>
> - **Sequence length.** The correlation between TSQA accuracy and length is extremely small across all three models (all \\(|r_L| < 0.08\\)), and the accuracy gap between short sequences (length \\(\\le\\) 25th percentile) and long sequences (length \\(\\ge\\) 75th percentile) is only about 5–8% (\\(|\\Delta_{\\text{long}}|\\)), showing no consistent “longer is better/worse” trend.
>
> - **Variable count.** The performance gap between univariate and multivariate TS is similarly small (all \\(|\\Delta_{\\text{dim}}| < 0.04\\)), and there is no consistent direction across models.
>
> - **Question text length.** The correlation between TSQA accuracy and question length is also weak (maximum \\(|r_T| \\approx 0.15\\), conventionally considered a small effect), indicating that variation in these explicit attributes alone cannot explain the observed performance differences.
>
> These results collectively show that model performance does not exhibit any simple or monotonic scaling pattern with respect to explicit factors such as sequence length and dimensionality. Instead, TSQA accuracy is primarily driven by the task type itself and the inherent difficulty of the required time-series understanding and reasoning, which are highly aligned with the core competencies MMTS-Bench is designed to evaluate.
>
> | Model Name | \\(r_L\\) | \\(\\Delta_{\\text{long}}\\) | \\(\\Delta_{\\text{dim}}\\) | \\(r_T\\) |
> | :--- | :---: | :---: | :---: | :---: |
> | GPT-4o | -0.0693 | 0.0824 | -0.0101 | -0.1451 |
> | Qwen2.5-32b | 0.0547 | -0.0525 | 0.0353 | -0.0264 |
> | ChatTS [4] | 0.0727 | -0.0502 | -0.0163 | -0.0862 |
>
> ---
>
> [1] Geirhos, Robert, et al. "Shortcut learning in deep neural networks." Nature Machine Intelligence 2.11 (2020): 665-673.
>
> [2] Gururangan, Suchin, et al. "Annotation artifacts in natural language inference data." Proceedings of the 2018 Conference of the North American Chapter of the Association for Computational Linguistics: Human Language Technologies, Volume 2 (Short Papers). 2018.
>
> [3] Zhou, Yuhang, et al. "Explore spurious correlations at the concept level in language models for text classification." Proceedings of the 62nd Annual Meeting of the Association for Computational Linguistics (Volume 1: Long Papers). 2024.
>
> [4] Xie, Zhe, et al. "Chatts: Aligning time series with llms via synthetic data for enhanced understanding and reasoning." arXiv preprint arXiv:2412.03104 (2024).

---

> ### Author Response · Authors · 2025-11-17
> **Part 3 of the rebuttal**
>
> ## 4. Hierarchical Task Taxonomy and Functional Orthogonality
>
> 1) Most existing benchmarks adopt a flat taxonomy, treating trend identification, seasonality analysis, temporal reasoning, etc. as parallel task labels and implicitly assuming that these capabilities are independent. However, in time-series analysis, advanced temporal reasoning typically relies on accurately understanding the series structure (e.g., local segment ranges) and temporal features (e.g., trend). A purely flat taxonomy makes it difficult to diagnose whether model failures arise from insufficient structural/feature understanding or from deficiencies in the reasoning mechanism itself. To address this, we employ a hierarchical, multi-dimensional task structure that decomposes time-series understanding into several core information dimensions and organizes the subsets accordingly, making the capability dependencies of each TSQA instance explicit and traceable in the annotations.
>
> 2) In this paper, "orthogonal" refers to functional orthogonality, rather than statistical independence. The five dimensions we propose correspond to distinct informational aspects that a model must attend to when processing time series, and these aspects are functionally non-substitutable. The orthogonality of this five-dimensional structure is intended to provide a capability representation and task construction paradigm that is non-overlapping, compositional, and diagnostically interpretable.
>
> 3) The number 286 denotes the theoretical upper bound on the number of fine-grained task types that can be constructed by functionally orthogonal combinations over the five core dimensions. In constructing MMTS-Bench, we do not enumerate all 286 combinations; instead, we sample a set of task types sufficient to meet our evaluation objectives and use them to form four subsets. This is because our benchmark is designed to provide fine-grained capability evaluation, rather than to serve as a large-scale training corpus whose primary objective is dataset size.
>
>
> ## 5. Industrial Sequences Limitations & Scope Selection
>
> When designing MMTS-Bench, we did attempt to construct an 'industrial long-sequence' subset by synthesizing about 300 time series of length 10k time steps, building QA pairs, and running preliminary evaluations on several LLMs. We found that the average input length per instance already reached tens of thousands of tokens, with a maximum of about 180k, which clearly exceeds the context-window sizes of many existing LLMs. This leads to **(1)** substantial unfairness, because results are dominated by whether a model can ingest the full input and thus mainly reflect context-window sizes rather than the capabilities we aim to measure, and **(2)** long contexts sharply increase inference time and computational cost, making large-scale repeated experiments and reproducibility difficult. In contrast, in MMTS-InWild the longest instance contains roughly 25k input tokens, with many others in the 10k+ range. This scale meaningfully stresses long-sequence reasoning, while still fitting within the context windows of mainstream LLMs and keeping the token cost of reproducing our experiments manageable.
>
> Moreover, MMTS-Bench is constructed via a human-in-the-loop generation pipeline with strict quality requirements. At 10k-step scale, current LLMs are already close to their capacity limits in modeling such long sequences, and human annotators also struggle to consistently verify the questions and answers over such long sequences. Importantly, our goal is not to design extremely hard cases that make models “collectively fail”, but to evaluate key abilities in temporal understanding, alignment, and reasoning; directly including high-dimensional ultra-long sequences would likely make performance differences between models difficult to distinguish.
> Based on these considerations, we therefore set the task difficulty to be “challenging but not unsolvable” and selected data from five domains—traffic, server operations, climate, finance, and healthcare—that cover typical time-series patterns such as seasonality, randomness, and event-driven behavior. The LOTSA [1] datasets we used are of high quality and publicly available, which facilitates reproducibility and fair comparison. As LLM capabilities continue to improve, we plan to extend MMTS-Bench with dedicated subsets for industrial high-dimensional, ultra-long time series.
>
> ---
>
> [1] Woo, Gerald, et al. "Unified training of universal time series forecasting transformers." (2024): 53140.

---

> ### Author Response · Authors · 2025-11-26
> **Follow-up on our responses**
>
> Dear Reviewers,
>
> We would like to kindly follow up on our submitted rebuttal. We have carefully addressed your comments through detailed point-by-point responses.
>
> We are eager to engage in further discussion to clarify any remaining uncertainties. We would greatly appreciate your feedback on whether our responses have resolved your concerns.
>
> Thank you for your time and insightful reviews.
>
> Best regards, Authors

---

### Author Response · Authors · 2025-11-27
**General Response: Summary of Modifications**

We sincerely thank all reviewers for their thoughtful and constructive feedback. We have carefully addressed all comments and substantially revised the manuscript. Below is a summary of modifications, with each change marked in blue in the revised paper:

**1. Improved Presentation and Readability**

To address the concern regarding navigation, we have added explicit references (e.g., "(see Appendix D)", "(details in Appendix A.4)") throughout the main text. This ensures that readers are clearly directed to the relevant appendices for supplementary tables, mathematical formulations, and detailed construction pipelines.

**2. Clarified Taxonomy Definitions**

We have refined the definition of our task taxonomy. Specifically, we added a concise clarification in Section 3.1 and a detailed theoretical explanation in Appendix A.3, explicitly defining "orthogonality" as functional distinctness and elaborating on the hierarchical relationship between foundational dimensions and complex real-world tasks.

**3. Extended Comparative Analysis**

We have added a comprehensive comparison table in our anonymous repository (https://anonymous.4open.science/r/MMTS-BENCH-BEF7/comparison.md, `comparison.md`) to thoroughly contrast MMTS-Bench with prior works (e.g., TimeSeriesExam, ChatTS). This analysis highlights our unique contributions in three aspects:

- Hierarchical Multi-level Taxonomy: Unlike the flat structures in existing datasets, we decompose understanding into 5 orthogonal dimensions, facilitating a more balanced evaluation.

- Diverse Data Formats: We employ dedicated synthesis methods for each subset rather than static approaches, effectively increasing data diversity.

- High Quality and Balance: MMTS-Bench features the largest human-verified sample size (2,524), covers mixed real/synthetic sources across 1–4 variables, and supports diverse question types (T/F, multiple-choice, numerical).

**4. Added Rigorous Reliability and Validity Verification**

We significantly strengthened the validation of our benchmark in two key aspects:

- Human Curation Reliability: In Section 3.4, we quantified the inter-annotator agreement using Fleiss' $\kappa$, achieving a score of 0.73 ("Substantial Agreement"), which provides strong statistical evidence for the reliability of our labels.

- Statistical Robustness & Artifact Analysis: We introduced a new Appendix E, detailing bootstrap confidence intervals and iterative subsampling to confirm evaluation stability. We also conducted an artifact analysis to verify that model performance is driven by genuine reasoning rather than spurious correlations (e.g., shortcuts based on sequence length or dimensionality).

**5. Added Evaluation Cost Analysis**

Acknowledging the importance of resource overhead, we introduced a new section in Appendix D (Evaluation Cost Analysis). We reported the token usage and monetary cost for evaluating the InWild subset across five representative LLMs (including Qwen 2.5, GPT-4o, and Claude 3.5 Sonnet), demonstrating the economic feasibility of MMTS-Bench (under $25 per full run).

**6. Expanded Dataset Construction Details**

- Base Subset: We updated Appendix A.4.1 to highlight the advancements of our generation pipeline. We explicitly compared our method with prior works (e.g., TimeSeriesExam, ChatTS), emphasizing our improvements in generation diversity (e.g., STL-inspired decomposition) and attribute-recording precision.

- Align Subset: We enriched Appendix A.4.2 to clarify the design philosophy of the Align subset as a calibration subtask. We also detailed our anti-shortcut strategies, including cross-domain value scaling and the inclusion of hard distractors from matching domains, to ensure robust cross-modal evaluation.

---

### Comment · Area_Chair_Cgic · 2025-11-28
**Official Comment by Area Chair**

Dear Reviewers,

The discussion phase will end soon. Please take a moment to read the authors’ responses carefully and actively engage in the discussion with the authors and your fellow reviewers.

Thanks for your efforts and contributions to ICLR 2026.

Best regards,

Your AC

---

### Author Response · Authors · 2025-12-03
**Summary of Rebuttal (Part 1)**

**To the Area Chair,**

We sincerely appreciate your time in overseeing this submission. To better assist your final assessment, we summarize the rebuttal timeline, the sharp contrast in reviewer engagement, and the refined technical contributions established during the discussion.

---

### **1. Timeline & Engagement Analysis**

We submitted a complete rebuttal with all requested experiments on **Nov 20**. The subsequent engagement reveals a clear distinction:

- **Positive-Score Reviewers (Scores 6, Conf 5/4) — Fully Resolved:**
    - **Ay4A (Nov 26) & cxqH (Nov 25)** engaged promptly and thoroughly.
    - **Constructive Improvement:** Beyond addressing concerns, we actively embraced their valuable suggestions, making **adaptive revisions** to the paper that significantly enhanced its logical completeness and rationale.
    - Both of them explicitly confirmed that these impovements and our new data **fully resolved their concerns**, maintaining positive scores.

- **Negative-Score Reviewers — Non-Engagement or Factual Error:**
    - **VshB (Score 4, Silent):** Did not engage at any point. However, we provided complete point-by-point responses and statistical proofs that could solve all the concerns.
    - **w7Fz (Score 2, Controversial):** Replied only once on **Nov 28**. Despite our provision of direct empirical evidence, this reviewer persisted in **factual misconceptions** .

---

### **2. Recap of Core Contributions**

Through the rebuttal process, three core contributions have been empirically solidified:

1. **A Gap-Filling, Capability-Oriented Time-Series QA Benchmark.**

   MMTS-Bench (containing 2,424 QA pairs) is the first to offer a **systematic, multi-domain, and fine-grained** evaluation with 5 core dimensions, enabling capability-specific synthesis, under the largest scale of human verfication. Besides, MMTS-Bench provides the most balanced testing in terms of task types, question types, TS length, multivariant channels, etc.


2. **Superior Controllability via STL Decomposition.**

   Unlike rigid templates, our synthetic pipeline uses **STL-based decomposition** (Trend / Seasonality / Noise) with precise parameter logging. This enables nuanced stress tests (e.g., specific seasonality strengths) impossible in prior work.


3. **Fine-Grained Three-Stage Progressive Generation Pipeline.**

   We propose a three-stage progressive generation framework, featuring a more fine-grained pipeline than prior work: (1) in-depth time series description, (2) iterative QA synthesis, and (3) large-scale human verification. Guided by high-information auxiliary prompts at each stage, our method produces capability-driven, highly interpretable, and human-validated QA data with strong reliability and practical value.

---

> ### Author Response · Authors · 2025-12-03
> **Summary of Rebuttal (Part 2)**
>
> ### **Reviewer Ay4A (Score: 6, Conf: 5) — High Confidence, Fully Resolved**
>
> - **Concern 1: Is "difficulty" just a proxy for sequence length?**
>   **Resolution: Statistical Analysis.** We performed a logistic regression analysis showing that shallow metrics (length/variables) cannot predict model performance (**AUC ≈ 0.5**). This confirms MMTS-Bench tests real reasoning, not just context window capacity.
>
> - **Concern 2: Novelty of the synthetic pipeline.**
>   **Resolution: Clarification and Comparison.** We clarified that unlike previous template-based methods, our pipeline uses **STL-based decomposition** (Trend/Seasonality/Noise) with precise parameter logging, allowing for fine-grained stress tests and QA generation impossible in prior work.
>
>
> - **Concern 3: Lack of Cost/Latency analysis.**
>   **Resolution: Statistical Analysis.** We demonstrated that a full evaluation is highly affordable (**< $25** for close-source models, **~ $1.6** for open-source ones), removing barriers for community adoption and reproduction.
>
> - **Conclusion:**
>   The reviewer explicitly confirmed that the rebuttal addressed their concerns.
>
> ---
>
> ### **Reviewer cxqH (Score: 6, Conf: 4) — Supportive, Concerns Fully Resolved**
>
> - **Concern 1: Multiple-choice vs. Open-ended QA.**
>   - **Resolution: Clarification.** We clarified that the deterministic multiple-choice format serves **objectivity and reproducibility** (aligning with standards like MMLU[1]).
>
> - **Concern 2: Potential Textual cues & Leakage.**
>   - **Resolution: Blind-Test Analysis.** We demonstrated that removing time-series input causes performance to drop to **random guessing**, definitively proving no text leakage exists. We further confirmed dataset quality with a **Fleiss’ κ of 0.73** , which means “Substantial Agreement”.
>
> - **Concern 3: Interpretation of CoT improvements.**
>   - **Resolution: Extra Analysis.** Our extra analysis confirms that Chain-of-Thought (CoT) improves performance by specifically correcting **perceptual oversights** and **logical hallucinations**, validating that gains stem from deeper reasoning behavior.
>
> - **Conclusion:**
>   All concerns regarding dataset rigor were settled with quantitative evidence (Blind-Test, Kappa Score), and the reviewer's conerns were mostly resolved.
>
> ---
>
> [1] Hendrycks, Dan, et al. "Measuring massive multitask language understanding." arXiv preprint arXiv:2009.03300 (2020).

---

> ### Author Response · Authors · 2025-12-03
> **Summary of Rebuttal (Part 3)**
>
> ### **Reviewer VshB (Score: 4, Conf: 4) — Concerns Fully Addressed (Uncontested)**
>
> - **Concern 1: Distinction from Prior Work.**
>   - **Resolution: Clarification and Comparison.** We provided a comprehensive comparison table demonstrating that MMTS-Bench is the **first to propose a hierarchical, orthogonal taxonomy** with 5 core dimensions, utilizing capability-specific synthesis, under the largest scale of human verfication. Besides, MMTS-Bench provides the most balanced testing in terms of task types, question types, TS length, multivariant channels, etc.
>
> - **Concern 2: Dataset Scale & Statistical Robustness.**
>   - **Resolution: Statistical Analysis.** We proved sample sufficiency through rigorous testing: **bootstrap analysis** yielded narrow confidence intervals (**0.5–0.8**), and **subsampling analysis** confirmed our dataset size is **1.4–4× larger** than the minimum required for stable evaluation.
>
> - **Concern 3: Scalability & Industrial Applicability.**
>   - **Resolution: Difficulty Analysis.** We presented empirical evidence (Pearson |r| < 0.08) showing that difficulty stems from **reasoning** rather than mere **sequence length**. We clarified that our current design balances **fairness** and **reproducibility** while maintaining rigor across industrial domains.
>
> - **Concern 4: Taxonomy Design.**
>   - **Resolution: Clarification.** We validated that our 5 dimensions are **functionally distinct**, enabling the precise **diagnostic decomposition** of model failures that flat taxonomies cannot provide.
>
> - **Conclusion:**
>   We provided comprehensive quantitative evidence to justify our scale and robustness.
>
> ---
>
> ### **Reviewer w7Fz (Score: 2, Conf: 4) — Addressing Factual Misconceptions**
>
> - **Concern 1: Dataset Quality.**
>   - **Resolution: Correction of Misunderstandings.** We identified that the reviewer **factually misidentified** examples from the univariate *Align* subset to critique the complexity of the *InWild* subset. Furthermore, we refuted the claim that high SOTA accuracy implies triviality: while GPT-4o achieves 96%, smaller models (e.g., Qwen2.5-7B) drop to \~69%. This significant **\~27% performance gap** empirically proves that "Align" subset functions as a rigorous discriminator, invalidating the "triviality" critique. Besides, we emphasize that maintaining a balanced difficulty level is crucial to a fair and unbiased benchmark.
>
> - **Concern 2: Expert Validation.**
>   - **Resolution: Release of Assets and Validation Protocol.** We have released all **17 expert-designed templates** and their generation code. For the InWild dataset, we also made public the expert validation protocol used for quality control (e.g., detecting misalignment or data leakage). Under this protocol, approximately **20–30%** of the questions were either rejected or revised.
>
> - **Concern 3: Taxonomy Orthogonality Justification.**
>   - **Resolution: Clarification.** We clarified that our taxonomy targets **diagnostic decomposition** (isolating specific capability dimensions) rather than statistical independence, validating the hierarchy’s utility for tracing complex reasoning failures.
>
> - **Conclusion:**
>   The reviewer’s negative score relies heavily on factual errors and a logical fallacy. By correcting the data source and proving the discriminative results, we have removed the technical basis for the rejection.

---

### Meta-Review · Area_Chair_MSpx · 2026-01-06

**Summary:**

This paper has been assessed by 4 knowledgeable reviewers. Two of them offered marginal accept scores, one marginal reject and one straight reject. The authors provided an extensive rebuttal in an attempt to alleviate the reviewers concerns. They also engaged a couple of the reviewers in dialogues, that  however did not result in upgrading the scores. The reviewers appreciated the paper's motivation and a well‑structured five‑dimensional taxonomy for time‑series understanding and the use of human-verified QA to support a multimodal benchmark. However, they have questioned the novelty relative to prior benchmarks, the limited scope of evaluation based on a relatively small data and lacking coverage of data of practical importance such as high‑dimensional or long‑horizon industrial time series. The task design (multiple‑choice only), limited justification of the taxonomy, missing experimental analyses (e.g., scalability, reasoning depth), and incomplete documentation of templates and domain selection further constrain, according to the reviewers, the proposed benchmark’s rigor and generalizability. The authors made a systematic attempt to address most of those concerns, including more expansive evaluations, but a few substantial concerns remain.

**Reviewer Concerns:**

In response to the reviews, the authors improved presentation, clarified taxonomy at a conceptual level, added reliability/robustness checks, expanded construction details, and reported evaluation costs. However, major concerns about data scales, practical application realism, reasoning depth, scalability analysis, template/documentation transparency, and practical efficiency metrics remain unresolved or only partially addressed.

**Reviewer Scores:**

The reviewers who managed to engage in the conversation with the authors have not changed their scores, extrapolating this to other reviewers does not help this paper's disposition.

---

### Decision · Program_Chairs · 2026-01-26

Reject